EMBO
Molecular Medicine

# DiPRO1 distinctly reprograms muscle and mesenchymal cancer cells

Jeremy Rich [1], Melanie Bennaroch[1], Laura Notel[1], Polina Patalakh [1], Julien Alberola[1], Fayez Issa [2], Paule Opolon[3], Olivia Bawa[3], Windy Rondof [4,5], Antonin Marchais [4,5], Philippe Dessen [4], Guillaume Meurice [4], Morgane Le-Gall [6], Melanie Polrot[7], Karine Ser-Le Roux[7], Kamel Mamchaoui [8], Nathalie Droin [9,10], Hana Raslova [10], Pascal Maire [2], Birgit Geoerger [5,11] & Iryna Pirozhkova [1,2,11✉]

## Abstract

**We have recently identified the uncharacterized ZNF555 protein as a component of a productive complex involved in the morbid function of the 4qA locus in facioscapulohumeral dystrophy. Subsequently named DiPRO1 (Death, Differentiation, and PROliferation related PROtein 1), our study provides substantial evidence of its role in the differentiation and proliferation of human myoblasts. DiPRO1 operates through the regulatory binding regions of SIX1, a master regulator of myogenesis. Its relevance extends to mesenchymal tumors, such as rhabdomyosarcoma (RMS) and Ewing sarcoma, where DiPRO1 acts as a repressor via the epigenetic regulators TIF1B and UHRF1, maintaining methylation of cis-regulatory elements and gene promoters. Loss of DiPRO1 mimics the host defense response to virus, awakening retrotransposable repeats and the ZNF/KZFP gene family. This enables the eradication of cancer cells, reprogramming the cellular decision balance towards inflammation and/or apoptosis by controlling TNF-α via NF-kappaB signaling. Finally, our results highlight the vulnerability of mesenchymal cancer tumors to si/shDiPRO1-based nanomedicines, positioning DiPRO1 as a potential therapeutic target.**

**Keywords** DiPRO1; Mesenchymal Cancer; Muscle; Retrotransposable Repeats; Methylation
**Subject Categories** Cancer; Musculoskeletal System

## Introduction

Scientific discoveries keep highlighting and clarifying the mechanisms controlling cell proliferation and differentiation. While the four transcription factors of the Yamanaka cocktail have been shown to be sufficient to induce pluripotency (Takahashi and Yamanaka 2006), further research has focused on the mechanisms of stem cell differentiation, their plasticity, and trans-differentiation potential (Eisenstein 2016). Investigations of the reprogramming of cancer cells, which possess characteristics similar to those of stem cells, is a main topic in cancer research (Schulze and Harris 2012). They focus on factors that can control the trajectory of differentiation and induce tumor cells to behave less aggressively.

The zinc finger (ZNF) family is composed of 1723 annotated genes in the human genome (Cassandri et al, 2017). The zinc finger domain is one of the most abundant DNA-binding motifs of eukaryotic transcriptional factors. The amino acid identity of the binding site defines the DNA targeting sequences of zinc fingers and their corresponding functions. Through their ability to regulate gene expression, ZNF proteins are involved in cell proliferation, differentiation, and cancer progression (Jen and Wang 2016; Cassandri et al, 2017). The C(2)H(2) zinc fingers (Hall 2005), CCCH zinc fingers (Hajikhezri et al, 2020), and the RanBP2 type zinc fingers (De Franco et al, 2019) may be involved in RNA binding promoting regulation of mRNA expression or processing. Depending on the presence of protein-binding domains, such as BTB/POZ, Krüppel-associated box (KRAB) and SCAN, ZNF may be involved in protein-protein interaction (Jen and Wang 2016). At least one-third of mammalian ZNF proteins include a KRAB domain. The family of KRAB-containing ZNF proteins (KZFP) is specific to tetrapod vertebrates and has been expanded extensively to hundreds of members along all mammalian evolution (Huntley et al, 2006; Imbeault et al, 2017). Many of the KZFP act in association with KRAB-associated protein 1 (protein: TIF1B/KAP1;

[1]UMR8126 CNRS, Gustave Roussy Cancer campus, Université Paris-Saclay, Villejuif, France. [2]INSERM U1016, CNRS UMR 8104, Institut Cochin, Paris, France. [3]Pathology and Cytology Section, UMS AMMICA, CNRS, INSERM, Gustave Roussy Cancer campus, Université Paris-Saclay, Villejuif, France. [4]Bioinformatics Platform, UMS AMMICA, CNRS, INSERM, Gustave Roussy Cancer campus, Université Paris-Saclay, Villejuif, France. [5]Department of Pediatric and Adolescent Oncology, Gustave Roussy Cancer campus, INSERM U1015, Université Paris-Saclay, Villejuif, France. [6]Proteom'IC facility, Université Paris Cité, CNRS, INSERM, Institut Cochin, F-75014 Paris, France. [7]Pre-clinical Evaluation Unit (PFEP), INSERM, Gustave Roussy Cancer campus, Université Paris-Saclay, Villejuif, France. [8]Sorbonne Université, Inserm, Institut de Myologie, Centre de Recherche en Myologie, F-75013 Paris, France. [9]Genomic Platform, UMS AMMICA US 23 INSERM UAR 3655 CNRS, Gustave Roussy Cancer campus, Université Paris-Saclay, Villejuif, France. [10]UMR1287 INSERM, Gustave Roussy Cancer campus, Université Paris-Saclay, Villejuif, France. [11]These authors contributed equally: Birgit Geoerger, Iryna Pirozhkova. ✉E-mail: iryna.pirozhkova@cnrs.fr

gene: TRIM28) to repress transcription by recruiting histone deacetylases, SETDB1 histone methyltransferase, HP1, and the NuRD (nucleosome remodeling deacetylase) complex containing histone deacetylases, and may contribute to methylation of CpG islands (Quenneville et al, 2012; Helleboid et al, 2019a, 2019b). One of the consequences of the KZFP repressive function is a silencing of repetitive DNA sequences, which constitute a large proportion of *cis*-regulatory elements, including enhancers, promoters, suppressors, insulators, and TF binding sites (Rich et al, 2014; Bertozzi et al, 2020). Epigenetic fine-tuning of repetitive DNA sequences is essential for appropriate development and cellular homeostasis, whereas any deregulation can lead to aberrant dedifferentiation and various pathological processes, including carcinogenesis (Sobocinska et al, 2021). Nevertheless, recent studies suggest that KZFP can associate with a variety of cofactors and play a dual role in transcription, extending the one previously believed to be limited to transcriptional repression (Lim and Park 2016; Helleboid et al, 2019a, 2019b). Despite the high abundance of KZFP in the human proteome, the biological roles played by most of them remain unknown so far. Thus, the experimental demonstration of predicted ZNF gene products, as well as the study of their functions is an important area of biomedical research.

Recently, we identified the ZNF555 (Q8NEP9) protein as part of the productive transcriptional hub in myoblasts from patients with facioscapulohumeral muscular dystrophy (FSHD). This protein binds to a *cis*-regulatory element of the beta-satellite repeat (BSR) that controls the transcription of muscle-specific and pro-apoptotic gene ANT1/SLC25A4 (Kim et al, 2015). Related to the above findings, the present study reports novel myogenic functions for ZNF555. We demonstrate that ZNF555 shares regulatory binding sites with SIX1, an upstream regulator of myogenesis (Maire et al, 2020) and epithelial-mesenchymal transition (Min and Wei 2021). ZNF555 drives muscle differentiation inhibition through SIX1 targets. We therefore applied ZNF555 targeting first to rhabdomyosarcoma (RMS) and then to Ewing sarcoma cells, two mesenchymal cancers, which showed increased levels of ZNF555 expression. Excitingly, the loss of ZNF555 results in cancer cell death. This was associated with global demethylation of CpG islands within the gene bodies and retrotransposable repeat elements (RE). Loss of methylation, in turn, may contribute to increased expression of pro-apoptotic genes as well as genes involved in inflammation and defense against viral infection. The cell death effect was further validated by targeting ZNF555 in the human cell-derived xenograft (CDX) mouse cancer model. Finally, the pan-cancer analysis of primary pediatric tumors illustrates the potential clinical relevance of ZNF555 targeting mesenchymal cancers and SOX2-related cancers. On the basis of morphological changes and transcriptional signature, ZNF555 was designated as death, differentiation, and PROliferation-related PROtein 1 (DiPRO1).

## Results

### DiPRO1 is an evolutionarily conserved gene in placental mammals

The DiPRO1/ZNF555 protein contains 628 amino acids (73.1 kDa, pI 8.9). It is encoded by NM_152791 and BC022022 [N107D

H194N K515T] (Mammalian Gene Collection) with some amino acid variations. The DiPRO1 protein includes one KRAB A-box and fifteen C2H2 zinc finger domains. In silico prediction describes three isoforms: Q8NEP9 (628 AA), Q8NEP9-2 (543 AA), and Q8NEP9-3 (627 AA) produced by alternative splicing (Appendix Fig. S1A). Isoform 2 lacks three zinc finger domains compared with isoforms 1 and 3. Isoform 3 has 99.8% similarity to isoform 1 with one missed amino acid difference. Protein blast analysis identified related proteins (BlastP, threshold 0.001) in vertebrates, with maximum homology in primates (98.7% in great apes and 89.5% in NW monkeys). However, a low homology of 50% does not allow identification of the DiPRO1 homologous protein in *Mus musculus*. Analysis of the phylogenetic history of all ancestral species showed the conservation of the DiPRO1 gene within placental mammals (Fig. 1A), indicating that DiPRO1 is an evolutionarily newly emerged gene found only in higher organisms (Mackeh et al, 2018). Significant ($p < 0.01$) expansion of the DiPRO1 gene was observed in pigs, American black bears, and ferrets. During evolution, these species acquired two DiPRO1 genes. Consistent with the results of the protein blast analysis, the DiPRO1 gene is absent in rodents and rabbits, and also with the exception of the naked mole rat, indicating that these animal models are not suitable for the DiPRO1 study. The hypothesis posits that the absence of the DiPRO1 gene in the mouse genome might have a substantial influence on the unique properties of human and mouse myogenic stem cells, affecting their self-renewal capabilities or their potential to give rise to RMS, as elaborated below.

### DiPRO1 binds the direct targets of SIX1

Recently, we have identified the DiPRO1 protein, as part of the productive transcriptional complex in the myoblasts of FSHD patients (17). In order to further explore the potential muscle-related functions and transcriptional co-regulators, we established the DiPRO1 direct physical targets by analyzing published ChIP-seq data (Imbeault et al, 2017, Data ref: Imbeault et al, 2017). The genomic distribution of DiPRO1-associated regions showed that 54% of the interactions occurred within gene-related regions, from 10 kb upstream of the transcription start site (TSS) to 5 kb downstream of the transcription end site (TES) (Fig. 1B). A considerable fraction of all detected regions (69.5%) was observed in RE, including short dispersed nuclear elements (SINE), long dispersed nuclear elements (LINE), and long terminal repeats (LTR) (Fig. 1C). In addition, the proportion of DiPRO1 binding to satellite and retroposon (SVA) repeat classes was multiplied by 6.8 and 8.7 compared to the human genome, suggesting the enrichment of DiPRO1 binding within these repeats. The chromatin state analysis based on histone marks (Ernst and Kellis 2010; Ernst et al, 2011) in the binding regions of DiPRO1 demonstrated their association with both repressed (64.4%) and active chromatin (30.7%) states, including promoter, enhancer and transcribed regions (Fig. 1D; Appendix Fig. S1B; Dataset EV1A). The Gene ontology (GO) analysis of direct transcriptional targets suggested that DiPRO1 functions may be coupled to cell cycle, transport, transcriptional co-regulator activity, and differentiation (Appendix Fig. S1C). Among them, MYH2 and MYH3, classical genes of mature and embryonic skeletal muscle, FABP3 (Pritt et al, 2008), NCOA2/TIF2 (Duteil et al, 2010) and SIRT1 (Vinciguerra et al, 2010) are directly related to muscle functions. Interestingly,

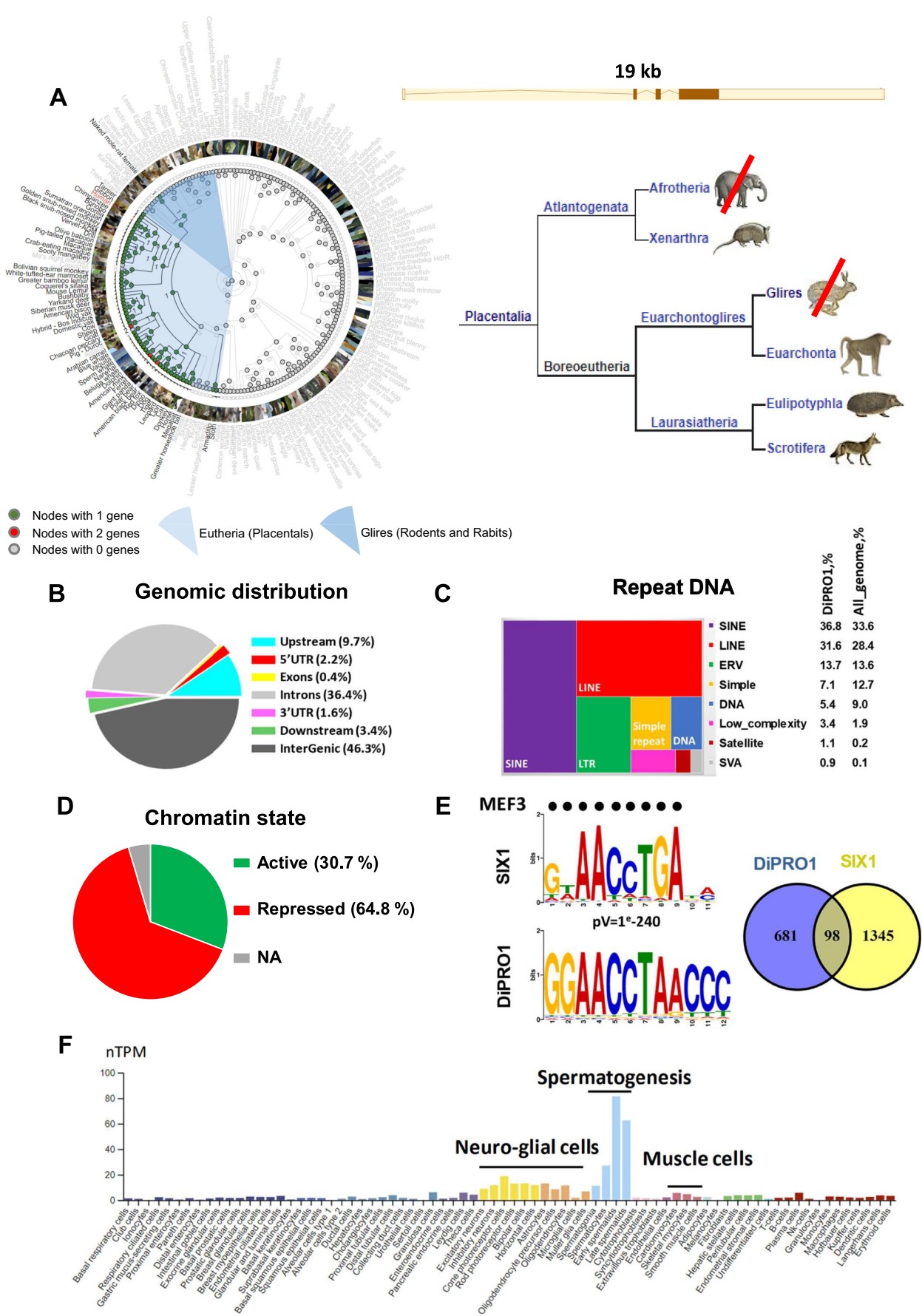

◀ **Figure 1.  DiPRO1 direct targets, de novo motifs, and associated chromatin state.**

(**A**) Phylogenetic tree of an Ensembl gene family representing the evolutionary history of gain/loss events of the DiPRO1/ZNF555 gene (chr19:2.841.435–2.860.471 region of GRCh38). A branch was marked as an expansion or contraction if the *p* value determined by the CAFE tool is <0.01. The numbers at each node refer to the number of DiPRO1 genes in the ancestral species. The color of each node reflects the number of genes colored according to the legend below. Species names are colored; red (Homo sapiens for the ZNF555 gene), black (species with the ZNF555 gene in Ensembl in this tree), and gray (species with no ZNF555 gene in this tree). (**B–E**) ChIP-seq data for DiPRO1 (GSM2466593) performed in HEK 293 T cells. (**B**) Pie chart of the distribution of DiPRO1 ChIP peaks relative to the gene body regions. The graph shows the percentage for each genomic location category. The upstream region was defined at −10 Kb from TSS and the downstream region at 5Kb from TES. Distribution of DNA repeats (**C**) and chromatin status (**D**) in DiPRO1 binding regions. Transcription factor binding regions and chromatin state were annotated using the ENCODE and BroadChromHMM tracks. The percentage of repeat class distribution in the whole genome and in DiPRO1 binding regions was shown. (**E**) The most specific DiPRO1 consensus motif 1 and its best match SIX1 (top left). The motif alignment was shown for the SIX1 human motif MA1118.1_Jaspar using the Tomtom tool. The black dots match the MEF3 motif (Santolini et al, 2016). Gene overlap between the *cis*-regulatory regions of DiPRO1 and SIX1. (**F**) Single cell type specificity of DiPRO1. Enhanced cell types are indicated. Data were taken from The Human Protein Atlas (www.proteinatlas.org).

NCOA2 fusion with PAX3 has been shown to inhibit myogenic differentiation in muscle cancer (Yoshida et al, 2014).

De novo motif identification for DiPRO1 revealed the motif with the highest enrichment rate (motif 1, GGGTTAGGTTCC), which accounted for 16.0% of the target DiPRO1 sequences (Fig. 1E; Appendix Fig. S1D). The best match to this motif corresponds to the transcription factor SIX1 (Liu et al, 2010; Santolini et al, 2016; Cheneby et al, 2018). In addition, the most abundant motif (motif 8, CAGAGGCGCACA) was found in 37.8% of the DiPRO1 target sequences, showing the highest homology to the ZSCAN4/GM397 motif (Appendix Fig. S1E). The most proximal genes associated with the *cis*-regulatory regions of DIPRO1 binding were shared between those of SIX1 (12.6%, jaspar.genereg.net) (Spitz et al, 1998; Cheneby et al, 2018) and ZSCAN4 (77.7%, GSE105311) (Fig. 1E; Dataset EV1B), thereby supporting the motif discovery analysis. Remarkably, the SIX1 homeoprotein serves as an upstream regulator of myogenesis (Maire et al, 2020), targeting the MEF3 motifs that are notably abundant among myogenic genes (Hidaka et al, 1993; Blais et al, 2005; Santolini et al, 2016). Additionally, it acts as a suppressor of differentiation and promotes metastasis in RMS muscle cancer (Reichenberger et al, 2005; Yu et al, 2006; Hsu et al, 2022).

The functions of ZSCAN4 are not specifically related to muscle, but rather to the epigenetic regulation of pluripotency in normal and cancer stem cells, affecting their growth (Hirata et al, 2012; Portney et al, 2020). Interestingly, ZSCAN4 has been found to be associated with microsatellite DNA regions (Srinivasan et al, 2020) and upregulated in muscles expressing a long FSHD-related macrosatellite transcript (DUX4-FL) (Ferreboeuf et al, 2014). In relation to RE, we have previously demonstrated that DiPRO1 can physically interact with a β-satellite DNA repeat (4qAe) in FSHD muscle cells (Kim et al, 2015). We show that the DiPRO1 binding motif 8, homologous to ZSCAN4, matches the repeat sequence. Consistently, the motif prediction was in good agreement with the experimental data (Appendix Fig. S1F). To further investigate transcriptional co-regulation of DiPRO1, the ENCODE collection of ChIP-seq data was used (Consortium 2012; Davis et al, 2018). The analysis revealed several regions of joint binding of DiPRO1 and TIF1B/KAP1, a mediator of transcriptional repression by KZFP (Dataset EV1A) (Turelli et al, 2014; Helleboid et al, 2019a, 2019b).

Collectively, this analysis predicts that DiPRO1 is a transcriptional regulator acting through gene promoters and *cis*-regulatory RE. DiPRO1 shows overlapping binding with SIX1, further arguing that it may play a role in myogenesis.

## DiPRO1 is a regulator of myogenic program in human myoblasts

Previously, we have shown a robust expression of DiPRO1 in myoblast cells in comparison to fibroblasts and its relevance to the muscular dystrophy FSHD (Kim et al, 2015). We have also reported that the most significant DiPRO1 expression fluctuations are related to musculoskeletal and connective tissue diseases according to the Disease Atlas (nextbio.com). We then queried a single-cell RNA-seq database (www.proteinatlas.org) to assess DiPRO1 expression in normal cells. The DiPRO1 expression is enhanced in three major cell types, namely, germ cells, neuro-glial cells, and muscle cells (Fig. 1F), supporting the prediction of DiPRO1 role in muscle. Therefore, we studied the role of DiPRO1 in muscle cells.

First, we performed DiPRO1 overexpression experiments. The full-length ORF (1887 bp) of DiPRO1 was fused to the C-terminus of the triple-tagged Flag, 6xHis, HA (FHH) YFP protein (pDiPRO1) within the pOZ retroviral backbone (Appendix Fig. S2A). In addition, the DiPRO1 ORF was codon-optimized, which enabled us to distinguish between endogenous and recombinant DiPRO1 expression (Appendix Fig. S2B). Using stable transduction of the pDiPRO1 retroviral vector, immortalized human myoblasts overexpressing the DiPRO1 mRNA and protein were generated and used for further study (Fig. 2A,B). The pDiPRO1 myoblasts were maintained in the myoblast proliferation medium. They had a rounded or short spindle shape with no alignment upon aggregation, in contrast to the long spindle or polygonal aligned myogenic parental cells (Wakelam 1985; Ostrovidov et al, 2014) (Fig. 2C). May-Grünwald–Giemsa (MGG) staining revealed that these cells had reduced size, hyperchromatic nuclei, and increased nucleus-to-cytoplasm (N/C) ratio (Fig. 2D), which are characteristics of undifferentiated and immature cells (Su Lim et al, 2015; Zhou et al, 2016). In addition, the pDiPRO1 myoblasts showed a significant increase in proliferation rate, comparable to that of RMS cells (Fig. 2E), and PAX3/6/9 gene upregulation (Fig. EV1A–C). In the differentiation medium, we did not observe multinucleated and elongated myotubes in the myoblasts overexpressing DiPRO1. The cells retained their disorganized shape and did not express muscle tropomyosin protein, suggesting a block in differentiation and fusion (Fig. 2F). To compare the effect of upregulation of DiPRO1 with that of downregulation of DiPRO1, myoblasts were then stably transduced with a lentiviral vector expressing a shRNA targeting DiPRO1 (shDiPRO1) (Fig. 2G). The resulting depletion of DiPRO1 induced the onset of cell cycle suppression at G1/S, with no difference in

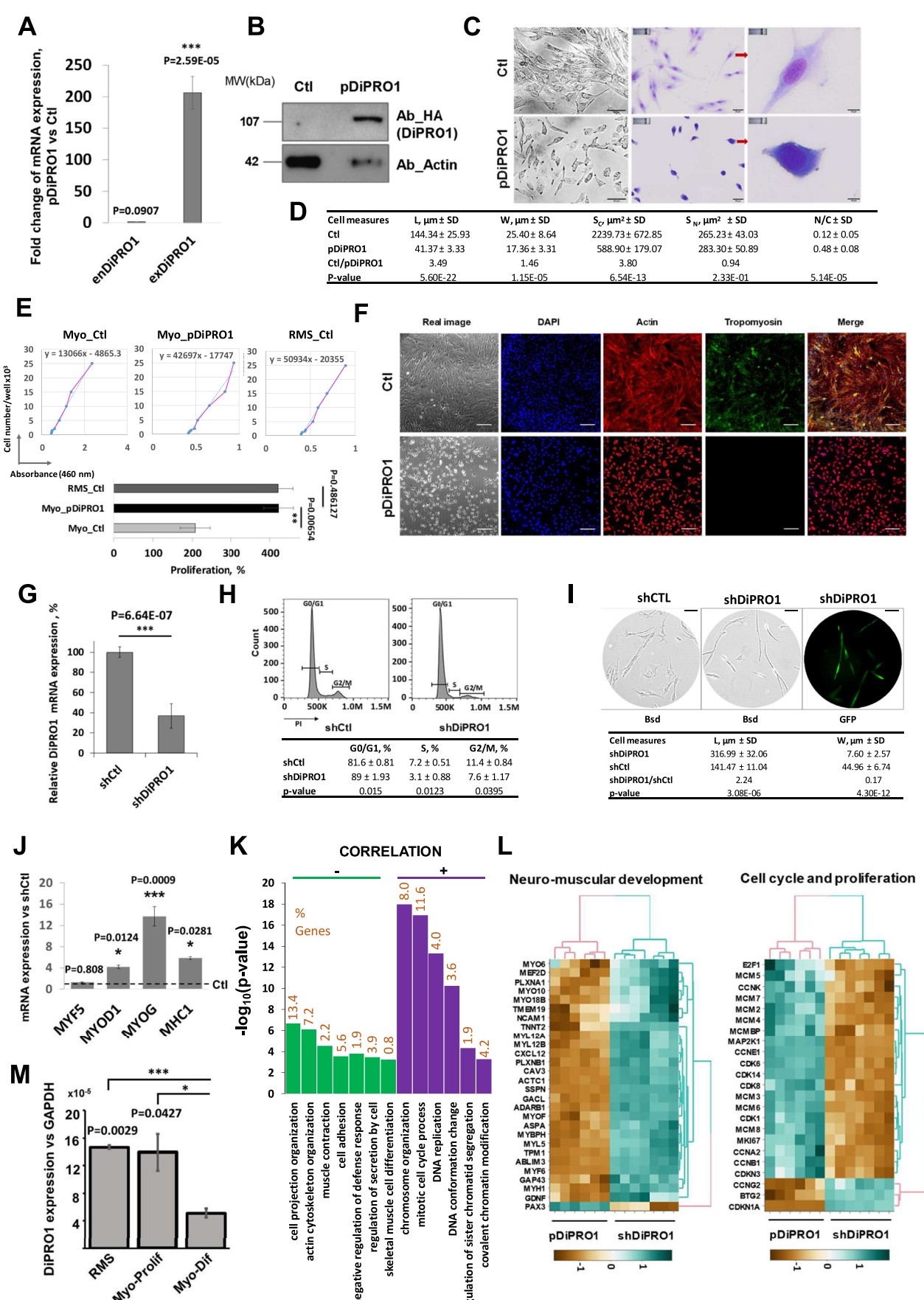

◀ **Figure 2. Genetic manipulation targeting DiPRO1 modulates the differentiation state of myoblasts.**

(A) The efficiency of exogenous DiPRO1 overexpression was verified by mRNA expression analysis. The qRT-PCR results were normalized to the GAPDH gene and represent three independent total RNA extractions measured in triplicate, FC ± SD. Ct = 45 was taken for exDiPRO1 ORF expression in control cells. The P value indicates the difference between endogenous (enDiPRO1) and exogenous (enDiPRO1) DiPRO1 expression, n = 3. (B) Western blotting revealed exogenous expression of DiPRO1 protein. Immunoblotting assays were performed on whole-cell extracts, with DiPRO1 protein labeled with the HA-tag. Actin was used as an internal control. (C) Morphological changes in myoblasts stably expressing pDiPRO1. Real images (left panel) and cells stained in MGG reagent (middle and right panels) are shown. The red arrows point to a magnified view of an individual cell. Scale bar: 50 µm (left, middle) and 10 µm (right). (D) DiPRO1 overexpression induces changes in cell dimension. L: cell length, W: cell width, Sc: cytoplasmic surface, Sn: nuclear surface, N/C: nuclear-cytoplasmic ratio. Unit of length = µm. Results represent m ± SD, n = 20. T-test was used for statistical analysis. (E) DiPRO1 induces myoblast proliferation. The proliferation rate was estimated for DiPRO1-overexpressing myoblasts (Myo_pDiPRO1) and their control (Myo_Ctl) and compared with RMS cells (RMS_Ctl) by Counting Kit 8 (Sigma-Aldrich). Cell number was determined using a titration curve performed at different cell dilutions from 0 to 25,000 cells/well. Cells were then seeded at $1.0 \times 10^4$, $1.5 \times 10^4$, and $2.5 \times 10^4$ cells/well and absorbance was measured at 460 nm after 72 h of proliferation. Viable cells were compared to the initial number of cells. Data were expressed as mean % relative to initial cell number ± SD and represent three independent experiments, t-test, n = 3. (F) Overexpression of DiPRO1 results in the block of myoblast differentiation. Cells were stained with anti-tropomyosin (green), anti-actin (red) and DAPI (blue) antibodies. The merge contains the combined images of three different stainings. Scale bar 120 µm. (G) DiPRO1 mRNA inhibition was verified by RT-qPCR analysis. Data were normalized to GAPDH expression and DiPRO1 expression in control cells was set to 100%, proportions (%) ± SD, n = 3 corresponding to three independent experiments measured twice, t-test. (H) Cell cycle analysis of DiPRO1-depleted myoblasts was performed one week after transduction. Transduced myoblasts were stained with propidium iodide (PI). The percentage of dead and viable cells in each phase, according to DNA content, was determined by flow cytometry and compared with control cells, proportions (%) ± SD, n = 3 corresponding to three independent experiments. (I) Morphological changes in myoblasts resulting from DiPRO1 knockdown (shDiPRO1) one week after transduction (top). Transduction efficiency was verified by confocal microscopy using the shDiPRO1 vector expressing GFP (top left). Scale bar 100 µm. Length (L) and width (W) of cells were measured (bottom). Length unit = µm. Results are presented as m ± SD, number of fields n = 20. (J) DiPRO1 KD contributes to myogenic gene expression. Results of RT-qPCR analysis were normalized to GAPDH expression and presented as ratios to shCtl. The t-test was applied for statistical difference with the appropriate control, FC ± SD, n = 3, indicating three independent experiments performed in duplicate. (K) GO terms of significantly regulated biological processes are summarized using the genetic interaction network. Cytoscape software with ClueGO plug-in was used, P < 0.01, kappa score threshold 0.4, n = 6–7. (L) Gene signatures of neuromuscular development, and cell cycle and proliferation pathways correlated with DiPRO1 expression versus control, n = 6–7. (M) Endogenous DiPRO1 gene expression in RMS, proliferating (Myo-prolif) and differentiating (Myo-dif) myoblasts. The qRT-PCR results were normalized to the GAPDH gene and represent three independent total RNA extractions measured twice, m ± SD, n = 3, t-test. Data information: In (A–G), two myoblast cell lines were transduced with a retroviral vector expressing the DiPRO1 ORF (pDiPRO1) and compared with parental control counterparts (Ctl). In (G–J), two human myoblast cell lines were transduced with a lentivirus vector expressing a shRNA targeting the DiPRO1 gene (shDiPRO1) and compared with myoblasts transduced with the equivalent vector expressing a nontargeting shRNA (shCTL). In (K, L), global transcriptome analysis by microarray was implemented using total mRNA extracted from myoblast cell lines: with DiPRO1 knockdown (shDiPRO1) or overexpression (pDiPRO1) and corresponding controls. Three or four separate extractions were performed from two cell lines (n = 6–7 per condition). Differentially expressed genes in pDiPRO1- and shDiPRO1-myoblasts compared with corresponding controls (P < 0.05). The Limma R package was used for statistical analysis. In (A, D, E, G–J, M), *, **, and *** indicate significant differences from the corresponding control, P < 0.05, 0.01, and 0.001, respectively. Source data are available online for this figure.

mortality compared to control cells (Figs. 2H and EV1D). One week later, the DiPRO1-depleted myoblasts underwent a transformation into elongated, thin cells (Fig. 2I), showing increased MYOD, MYH1, and MYOG mRNA levels (Fig. 2J).

Next, genomic transcriptomic analysis was performed to identify differentially expressed genes (DEGs) focusing on common responders in the DiPRO1 overexpression and knockdown models (Fig. EV1A–C, E,F; Dataset EV2). Negatively correlated genes with DiPRO1 expression in loss-of-function and gain-of-function have been involved in muscle tissue development, actin cytoskeleton organization, and muscle contraction (Fig. 2K). Among them, MYH1, MYF6, MYO18B, TNNT2, and TPM1, key genes for skeletal muscle fiber development and muscle differentiation (41,42) were identified (Figs. 2L and EV1G), which showed inversed expression with DiPRO1 gene levels. In contrast, PAX3, a marker of myogenic progenitor cells (43), showed a parallel correlation with DiPRO1 expression. The genes positively correlated with the DiPRO1 expression level were enriched in the cell cycle and replication pathways. The potential function of DiPRO1 in proliferation and cell cycle was reflected by transcriptional changes in genes encoding MCM and Ki-67 proteins, conventional markers of cell proliferation (Fig. 2L) (Barton and Levine 2008; Jurikova et al, 2016). The expression level of cell cycle regulatory genes was mainly distinguished by a positive correlation with cyclins A, B, E, cyclin-dependent kinases (CDK4, CDK6, and CDK1), E2F1, and by a negative correlation with p21/CDKN1A, predominantly interfering in the G1/S phase transition, in agreement with the cell cycle analysis results (Fig. 2H).

Interestingly, the global transcriptome of shDiPRO1 myoblasts differed less from controls (PC2 = 10.1%) than that of pDiPRO1 (PC1 = 34.1%), suggesting less severe rearrangement during DiPRO1 downregulation (Fig. EV1H).

Finally, analysis of endogenous DIPRO1 gene expression demonstrated a 2.7-fold decrease during myogenic differentiation, consistent with our observations above (Fig. 2M). Taken as a whole, the functional models provide evidence that DiPRO1 maintains myogenic stem cells in a proliferative program and antagonizes their differentiation.

## DiPRO1 depletion induces mesenchymal cancer cell death in vitro

The RNA-seq analysis of 1375 human cancer cell lines (Data ref: DepMap portal of Broad Institute) demonstrated particularly high expression of DiPRO1 in RMS (muscle cancer) and Ewing sarcoma (bone or soft tissue cancer), both of mesenchymal origin (Fig. EV1I; Dataset EV2) (Xiao et al, 2013; Louis et al, 2016). It has been previously reported that mesenchymal stem cells (MSCs) exhibit a high potential for transformation into mesenchymal tumors (Vishnubalaji et al, 2019) and could be a possible source of trans-reprogramming of RMS into Ewing sarcoma by introducing the EWS-FLI1 gene (Hu-Lieskovan et al, 2005). In addition, the analysis of DiPRO1 gene expression revealed comparable levels in RMS and proliferative myoblasts (Fig. 2M). We, therefore, investigated whether the phenomenon of cell cycle restriction and proliferation inhibition induced by DiPRO1 KD in myoblasts could

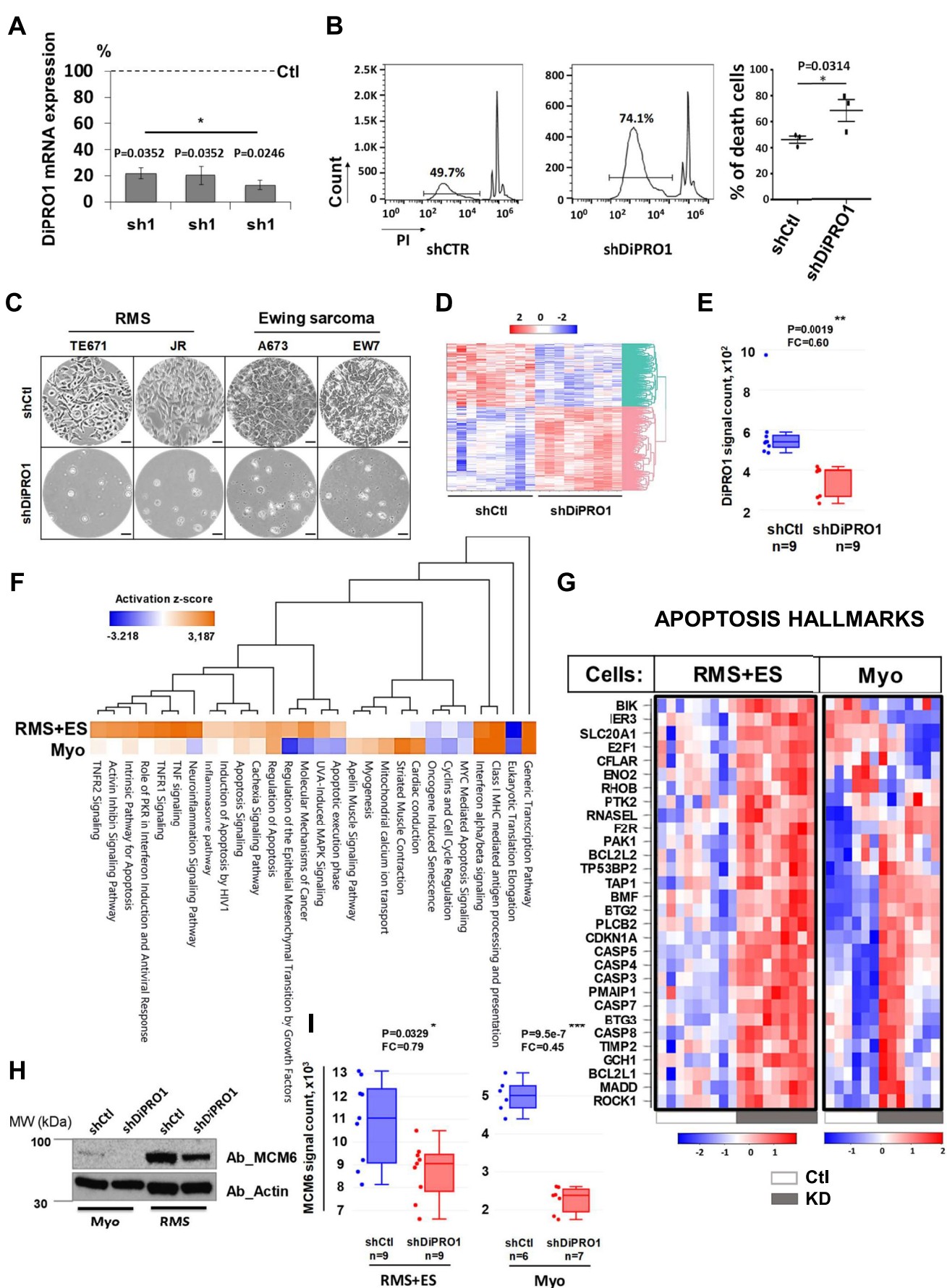

**Figure 3. Functional depletion of DiPRO1 gene expression compromises cell fate and modulates the transcriptome of RMS and Ewing sarcoma cells.**

(A–I) DiPRO1 knockdown was achieved by lentiviral transduction of vectors producing a non-targeted shRNA (shCtl) and a shRNA targeting the DiPRO1 gene (shDiPRO1). (A) The efficiency of DiPRO1 inhibition was verified by mRNA expression analysis in TE671 cells. Three different shRNAs (shDiPRO1-1/2/3) were tested individually and the inhibition effect was analyzed by RT-qPCR. Total mRNA was extracted 48 h after transduction. The results represent three independent assays and were compared to shCtl, referenced as 100%, using the t-test, *P < 0.05, n = 3, proportions (%) ± SD. (B) Induction of cell death in TE671 cells lacking DiPRO1 48 h after transduction. Three independent experiments correspond to three individual shDiPRO1. Transduced RMS cells were stained with propidium iodide (PI). The percentage of dead cells in non-gated areas was determined by flow cytometry. The pairwise t-test was applied for statistical analysis, *P < 0.05, n = 3, proportions (%) ± SD. (C) Dramatic cell death was observed 5–7 days after transduction of TE671 and JR RMS, and A673 and EW7 Ewing sarcoma cell lines. Scale bar: 50 μm. The cell images are also represented in Appendix Fig. S3. (D) Heatmap showing genes that were affected in RMS and Ewing sarcoma cells under DiPRO1 knockdown versus control. Clustering analysis of Euclidean distribution, full linkage. (E) Boxplots of DiPRO1 downregulation in both RMS and Ewing sarcoma cells with DiPRO1 knockdown (red) and in control (blue). Data were presented as median (Q2) ± Q3-Q1, whiskers extend to 1.5x IQR, **P < 0.01. (F) Canonical pathways differentially regulated in cancer (RMS + ES, n = 9) and normal (Myo, n = 6-7) cells under DiPRO1 knockdown. Analysis was performed using Ingenuity software, Fisher's Exact test p < 0.05. (G) Common apoptosis hallmarks in RMS and Ewing sarcoma (RMS + ES, n = 9), myoblasts (Myo, n = 6-7) and appropriate controls. (H, I) MCM6 protein (H) and gene (I) expression in myoblasts (Myo, n = 6-7) and cancer (RMS + ES, n = 9) cells with DiPRO1 KD and control counterparts was determined 48 h after transduction. The mRNA expression data are presented as median (Q2) ± Q3-Q1, whiskers extend to 1.5x IQR. *P < 0.05, ***P < 0.001. Western blot was performed using anti-MCM6 and anti-actin antibodies (internal control). Data information: In (D–G, I), global transcriptome analysis by microarray was implemented using total mRNA extracted from RMS and Ewing sarcoma cell lines: with DiPRO1 knockdown (shDiPRO1, n = 9) and corresponding controls (shCtl, n = 9). Differentially expressed genes versus controls (P < 0.05) were analyzed using the Limma R package. Source data are available online for this figure.

be extrapolated to mesenchymal cancer cells. Initially, TE671 RMS cells were subjected to DiPRO1 KD by transduction with lentiviral vectors each expressing one of the three shRNAs (shDiPRO1-1/2/3) or a non-targeted shCtl (Fig. 3A). Propidium iodide (PI) staining showed a 24% increase in the TE671-sub-G1 fraction compared with control 48 h after transduction, indicating the onset of cell death (Fig. 3B). This led to a progressive deregulation of the cell cycle as seen in the diploid phases G0/G1 and S (Fig. EV1J) (Hinson et al, 2013), resulting in 95-100% cell death by day 5–7. The observed effect was similar for three individually transduced shDiPRO1 (Appendix Fig. S3A). RMS in children occurs as two main subtypes, embryonal (ERMS) and PAX-FOXO fusion-positive alveolar (ARMS) (89). The TE671 cell line belongs to the ERMS subtype. Therefore, we additionally screened the ARMS JR cell line as well as the Ewing sarcoma cell lines (A673, EW7). As shown in Fig. 3C; Appendix Fig. S3B–D, the cell death phenomenon in sarcoma cells was comparable to that observed in the ERMS cells and was independent of the sarcoma type. Given the reproducibility of the effect induced by each shDiPRO1, one of the three shDiPRO1s (shDiPRO1-1) was used in subsequent experiments. Thus, the effect of DiPRO1 KD differed from that observed in myoblasts, suggesting the induction of an aberrant death program in malignant cells.

To identify distinct anti-tumorigenic mechanisms of DiPRO1 depletion in mesenchymal cancer cells, the global gene expression changes in RMS TE671 and Ewing sarcoma A673 cells were compared to those in myoblasts. The early molecular events (48 h) of shDiPRO1-introduced cells were compared to those of shCtl. DiPRO1 depletion altered the expression of 4202 unique genes across these cells, corresponding to 57% up- and 43% down-regulated genes (Limma, padj >0.05) (Fig. 3D,E; Dataset EV3). There is a 20.5% overlap in upregulated DEGs and a 14.8% overlap in downregulated DEGs between mesenchymal cancer cells and myoblasts in response to DiPRO1 knockdown, suggesting a distinct whole transcriptome signature in a cell-state-dependent manner (Appendix Fig. S4). Ingenuity canonical pathway (Qiagen) analysis showed differential activation of several pathways involved in myogenesis, striated muscle contraction and cardiac conduction in shDiPRO1 myoblast cells, consistent with the observed cellular phenotype (Fig. F). In contrast, depletion of DiPRO1 in cancer cells

induced activation of signaling pathways for inflammation (TNF/TNFR1/2, inflammasome, neuroinflammation), stress response (MAPK signaling), and apoptosis (intrinsic pathway, apoptotic execution, and HIV1 induction of apoptosis). In contrast to myoblasts, the DiPRO1 KD led to the upregulation of genes associated with apoptosis (Liberzon et al, 2015, Data ref: Liberzon et al, 2015), including ROCK1 (Shi and Wei 2007), RHOB (Liu et al, 2001), E2F1 (Lazzerini Denchi and Helin 2005), TP53BP2 (Samuels-Lev et al, 2001) and BIK (Chinnadurai et al, 2008) (Fig. 3G). The increased gene expression of initiator caspase CASP8, along with effector caspases CASP3 and CASP7, exhibited a correlation with caspase-3 protein activation in RMS cells (Fig. EV1K). This observation suggests that the depletion of DiPRO1 has the potential to induce caspase-dependent cancer cell death (Fig. 3I) (Tang et al, 2019). Inflammatory caspases CASP4 and CASP5 were upregulated in both cell types, correlating with partial activation of inflammation pathways in myoblasts and a robust activation in cancer cells. These shared effects were also associated with activated IFN-signaling and the suppression of senescence and cell cycle-related pathways. MCM6 emerged as a common responder to DiPRO1 downregulation in both myoblasts and mesenchymal cancer cells (Fig. 3H,I), providing further evidence of DiPRO1 shared role in cell proliferation (Schrader et al, 2005). It's worth noting that the overexpression of DiPRO1 led to an enhanced proliferation rate of RMS cells without apparent morphological changes (Figs. EV1L and 2E).

Collectively, these findings illustrate that inhibiting DiPRO1 not only impairs cell proliferation, as observed in myoblasts but also selectively triggers cell death by activating inflammation, stress response, and apoptosis in mesenchymal cancer cells. This underscores an aberrant transcriptional program and its associated functional impact in normal and malignant cells.

## Proof of concept: in vivo antitumor activity evaluation

Tailored nanocarriers have gained huge research focus for tumor drug delivery (Din et al, 2017). The branched and linear polyethylenimines (PEIs) have become prominent gene carriers for cancer and are explored in clinical trials (ClinicalTrials.gov NCT00595088 and NCT02806687). To confirm our in vitro findings and to address the

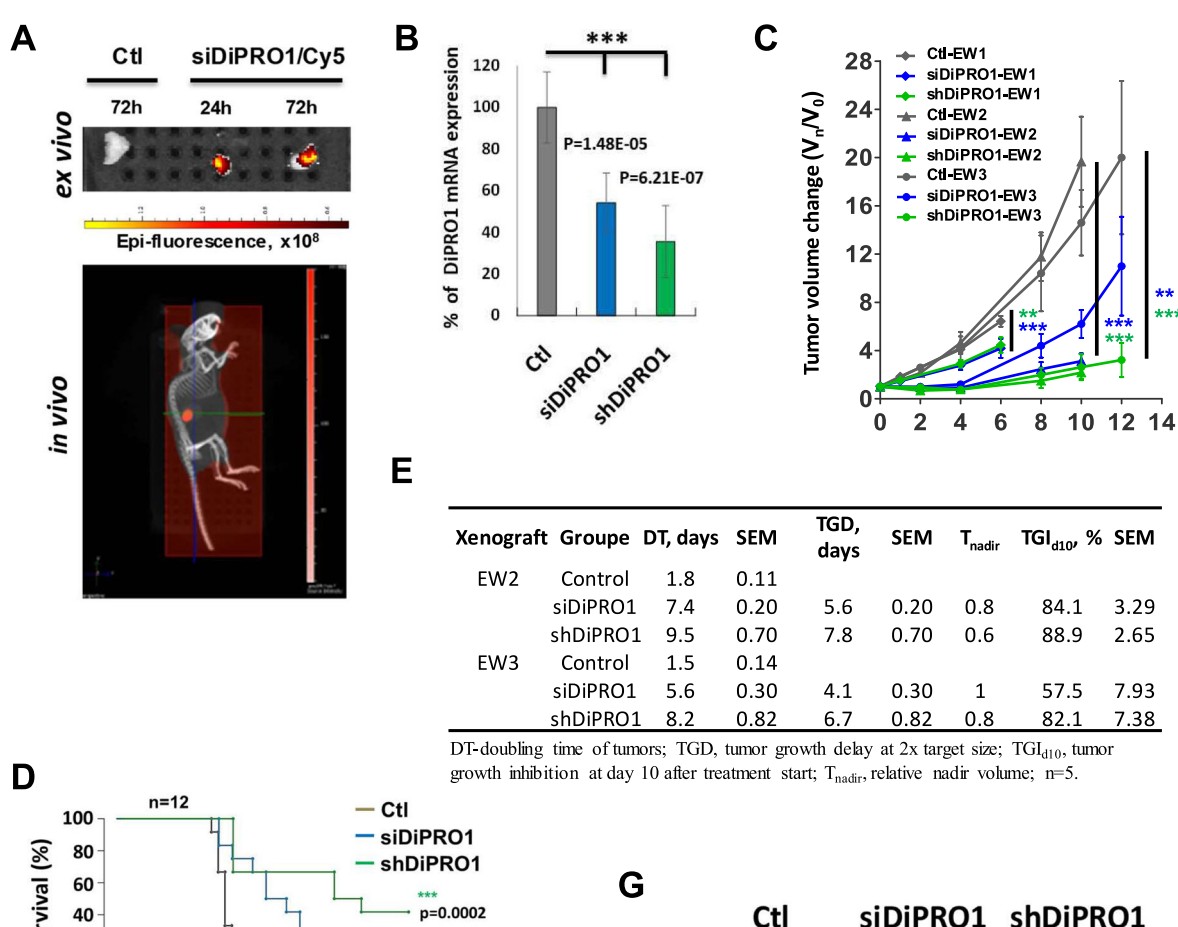

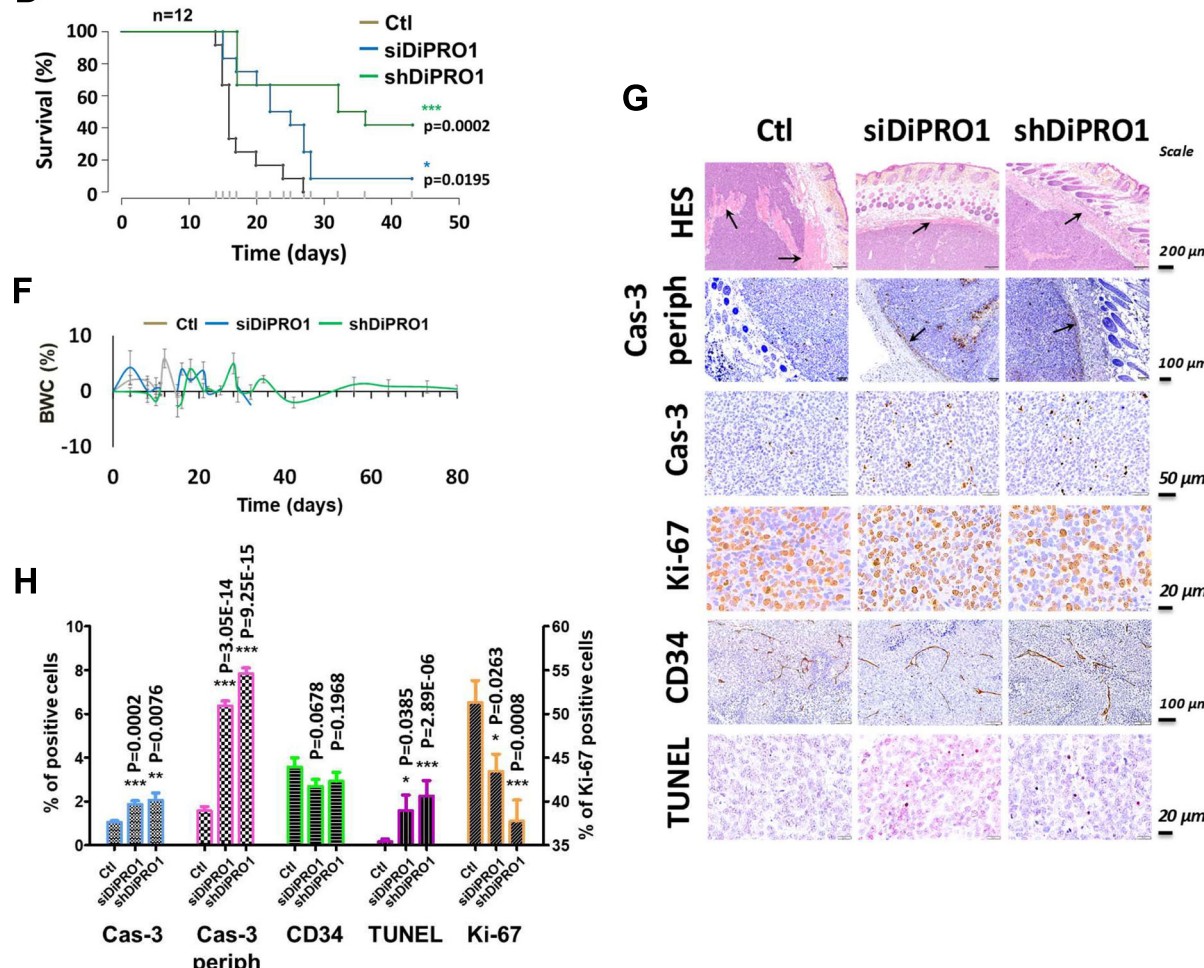

| Xenograft | Groupe | DT, days | SEM | TGD, days | SEM | $T_{nadir}$ | $TGI_{d10}$, % | SEM |
|-----------|--------|----------|-----|-----------|-----|-------------|----------------|-----|
| EW2 | Control | 1.8 | 0.11 | | | | | |
| | siDiPRO1 | 7.4 | 0.20 | 5.6 | 0.20 | 0.8 | 84.1 | 3.29 |
| | shDiPRO1 | 9.5 | 0.70 | 7.8 | 0.70 | 0.6 | 88.9 | 2.65 |
| EW3 | Control | 1.5 | 0.14 | | | | | |
| | siDiPRO1 | 5.6 | 0.30 | 4.1 | 0.30 | 1 | 57.5 | 7.93 |
| | shDiPRO1 | 8.2 | 0.82 | 6.7 | 0.82 | 0.8 | 82.1 | 7.38 |

DT-doubling time of tumors; TGD, tumor growth delay at 2x target size; $TGI_{d10}$, tumor growth inhibition at day 10 after treatment start; $T_{nadir}$, relative nadir volume; n=5.

◄ **Figure 4.   Antitumor activity of DiPRO1 inhibitors in the Ewing sarcoma (EW) subcutaneous tumor xenograft model.**

(A) Uptake by tumor cells of siDiPRO1/jetPEI®/Cy5 nanocomposites. Internalization was followed in live mice (lower panel) treated with a single dose (0.5 mg/kg) of Cy5-coupled anti-DiPRO1 siRNA complexes and in extracted tumors (upper panel) 24 and 72 h after treatment. Control mice were treated with an equimolar dose of non-targeted siCtl/jetPEI® free of Cy5. (B) DiPRO1 depletion efficiency in tumors was verified by RT-qPCR. DiPRO1 mRNA expression levels in three independent experiments performed in triplicate were normalized to GAPDH, $n = 9$. DiPRO1 expression in control tumors was referenced to 100%, proportions (%) ± SD. Statistical analysis was performed using the $t$-test. (C) Tumor progression was compared between the indicated groups ($n = 5$). Results from three independent experiments (EW1, EW2, and EW3) are presented. A two-way ANOVA test was applied for statistical analysis at the indicated time points. The data were presented as $V_n/V_0$ ratio ± SEM. (D) Effect of DiPRO1 inhibition on survival of tumor-bearing mice. Kaplan–Meier curves of overall survival for 43 days ($n = 12$) from the day of xenograft implantation are shown. Log-rank test (Mantel-Cox) for each pairwise comparison (treatment vs. control) was applied for statistical analysis. Results from two independent experiments (EW1 and EW2) are combined. The $p$ value was adjusted by the Holm method, proportions (%) ± SEM. (E) Analysis of main endpoints of antitumor activity of the si/hDiPRO1 and control nanomedicines against human Ewing sarcoma xenografts. (F) Assessment of body weight change (BWC) in mice bearing tumor xenografts. The results of two independent experiments (EW1 and EW2) are combined, $n = 12$, proportions (%) ± SEM. (G) Immunohistochemical staining of excised tumor tissues at day 12. Representative images of tumor sections stained with hematoxylin/eosin/safranin (HES) (arrows indicate striated muscle), and cells positive for internal and peripheral caspase-3 (Cas-3 periph, indicated by arrows), Ki-67, CD34, and TUNEL are shown. (H) Quantitative analysis of positively immunostained cells (%) relative to the total number of tumor cells. Six to thirteen fields were selected for counting. $n = 6$–13, proportions (%) ± SEM, $t$-test was used. Necrotic fields were excluded. Data information: In (B–H), nude mice received s.c. inoculations of A673 Ewing sarcoma cells and were treated with siDiPRO1/jetPEI® and shDiPRO1/jetPEI® nanocomposites or siCtl/jetPEI® scramble (Ctl) at a dose of 0.5 or 1 mg/kg/d by intratumoral administration. The initial tumor volume on the day of treatment initiation was V0 = 168 ± 76 mm³ in the first (EW1; $n = 7$ mice per group) and V0 = 71 ± 24 mm³ in the second (EW2) and third (EW3) independent experiments ($n = 5$ mice per group). In (B, C, H), *$P < 0.05$, **$P < 0.01$, ***$P < 0.001$, $P$ values are shown between indicated groups and appropriate controls. Source data are available online for this figure.

significance of DiPRO1 KD in tumor development in vivo, we evaluated the tumor growth when DiPRO1 was locally attenuated using sh/siRNA-based nanomedicines. Antitumor activity of the si- or shDiPRO1/jetPEI® at equimolar doses was investigated against Ewing sarcoma cell xenograft compared to scramble siRNA/jetPEI® (Ctl) in three independent experiments. The tumor cell internalization of siDiPRO1/jetPEI® coupled with Cy5 in living mice and extracted tumors was confirmed 24 and 72 h after a single injection of 0.5 mg/kg (Figs. 4A and EV2A; Movie EV1) and DiPRO1 gene inhibition was verified (Fig. 4B). In the first experiment (EW1), treatment was initiated when tumors had reached tumor volumes of 168 ± 76 mm³ and the nanocomposites were administered according to a twice-weekly schedule at a dose of 0.5 mg/kg/injection during 1 week followed by 1 mg/kg/injection (Fig. EV2B). The treatment of both siDiPRO1 and shDiPRO1 nanomedicines showed moderate tumor growth stabilization as compared to controls in this rapidly growing model, however tumor regression and survival advantage was evident in smaller tumors (Fig. EV2C–E). Two additional independent experiments (EW2 and EW3) were performed starting treatment at tumor volumes of 81 ± 21 mm³ with an intensified three times-per-week regimen during the first 2 weeks followed by twice-weekly 0.5 mg/kg/injection doses (Fig. EV2F). Visible ulcerations were observed in the EW3 model independent of treatment, therefore all experimental animals were sacrificed at day 12 according to animal protection endpoints and the survival analysis was not taken into account. In three independent experiments, the treatment with si- or shDiPRO1/jetPEI®-nanomedicines reduced tumor growth starting from day 4 of treatment (Figs. 4C and EV2C,D, G,H, J,K). Both treatments resulted in the survival advantage ($p$adj <0.05, Holm method) (Figs. 4D and  EV2E,I). Tumor growth delay (TGD) and tumor growth inhibition at day 10 (TGId10) was more prominent in the group treated with shDiPRO1/jetPEI® relative to siDiPRO1/jetPEI® (Fig. 4E). This was correlated with a stronger inhibiting effect on DiPRO1 gene expression (Fig. 4B). Of note, the weight maintenance of treated mice suggests an absence of significant toxicity of the DiPRO1-inhibiting nanomedecines (Fig. 4F). The antitumor activity of DiPRO1 depletion was also confirmed by morphological and immunohistochemical analyses of tumor biopsy sections at day 12

after treatment initiation (Fig. 4G). In controls, specimens were composed of malignant proliferative Ewing sarcoma consisting of large atypical polygonal eosinophilic cells associated with focal necrotic areas. At its periphery, the tumor was ill-defined with irregular nests of tumor cells invading the adipose tissue of the mouse hypoderma, indicating aggressive behavior of the tumor. Concerning the siDiPRO1/jetPEI® treated group, a similar aspect of the proliferation was observed, including again focal necrotic areas. However, the peripheral limit of the tumor exhibited a different pattern. The tumor cells were well limited by a thin regular layer of normal striated muscle cells tending to encapsulate the tumor. The same peripheral limitation of the proliferation was also observed in the shDiPRO1/jetPEI® group. The major change here was the presence of vast amounts of pink necrotic tissue suggestive of tumor destruction. These results were confirmed by TUNEL staining of the tumor biopsy sections used to assess the induction of apoptosis. The mean proportion of TUNEL positive cells/field (Fig. 4H) in the controls was 0.08%, whereas the si- and shDiPRO1/jetPEI® treated tumors had a mean proportion of 3.1 and 2.3%, respectively, corresponding to a 39.4- and 28.2-fold increase. Immunostaining of the active form of caspase-3 protease revealed caspase-3 activation inside tumor cells in both DiPRO1-depleted tumors, which was more pronounced at its periphery. Tumors from mice receiving the si- and shDiPRO1/jetPEI®-PEI nanoparticles revealed 4.0- and 5.0-fold increases in peripheral caspase-3 expression and 1.2- and 1.4-fold decrease, respectively, of Ki-67 expression, a marker of proliferating cells. No significant difference in CD34, a marker of early hematopoietic and vascular-associated tissue, was observed at this stage of treatment. It is noteworthy that shDiPRO1/jetPEI®-PEI treatment in EW2 resulted in the survival of 60% of animals for 84 days after xenograft implantation. Histological analysis of their tumors revealed very small specimens without viable malignant cells or with a weak presence of viable tissue among a large area of necrotic tumor (Fig. EV2L).

In conclusion, nanocomposites inhibiting DiPRO1 show effective regression of tumor growth rate of Ewing sarcoma and absence of weight loss in animal models. Apoptosis-mediated tumor cell death verifies the feasibility of targeting DiPRO1 and is consistent with the results of the in vitro study.

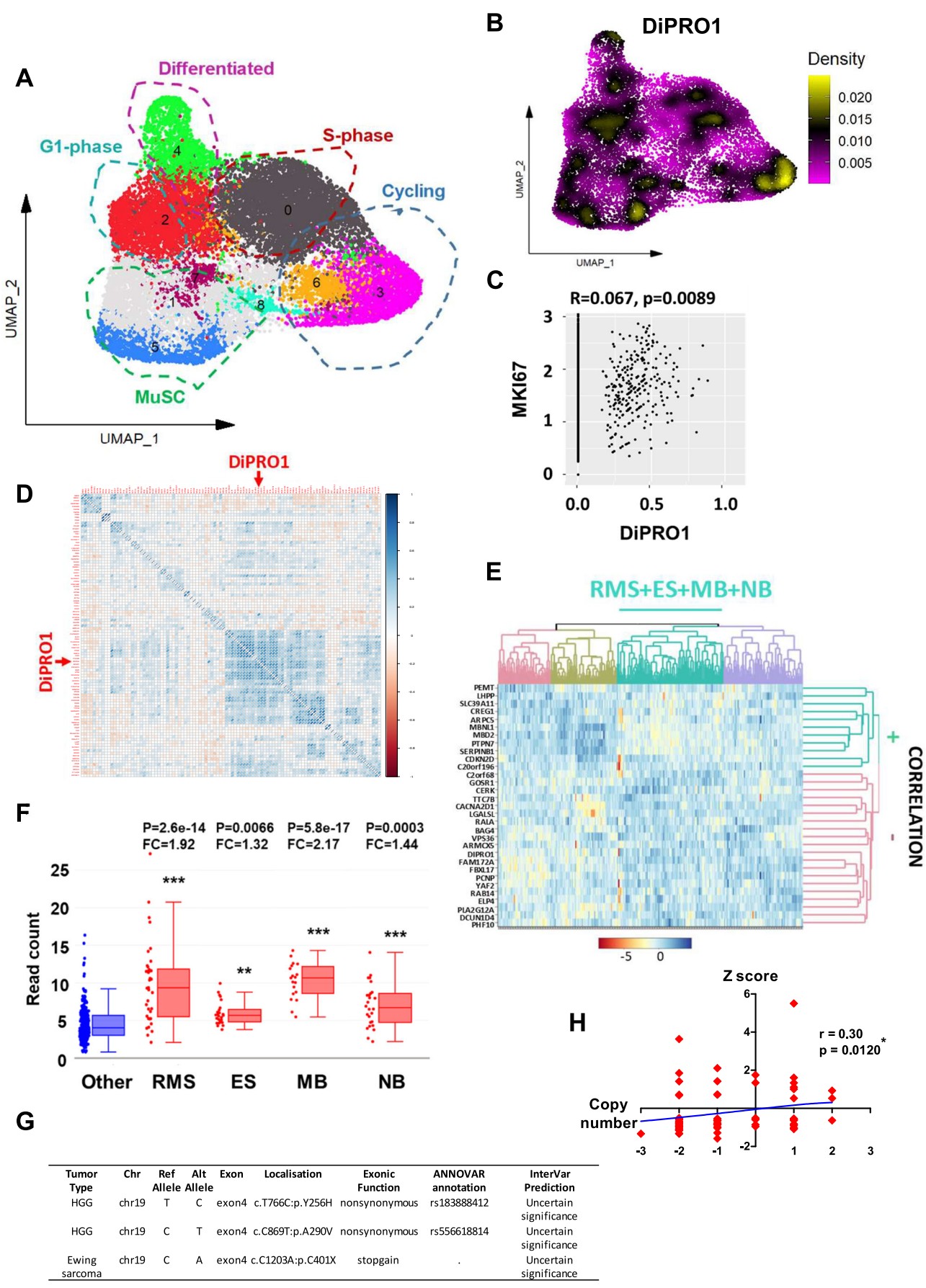

◄ **Figure 5. Clinical relevance of the DiPRO1 gene and its downstream targets in primary pediatric tumors.**

(A–C) scRNAseq datasets (GSE218974) derived from PDX primary RMS cultures ($n = 3$ eRMS and $n = 3$ aRMS) were reanalyzed using Seurat R toolkit. The individual data were normalized, feature-selected, and integrated into the individual data samples using the SCTransform function. (A) UMAP plot showing RMS cell populations after integration. (B) DiPRO1 expression density in RMS cell populations. Analysis was performed using the R package scCustomize. (C) Coexpression of DiPRO1 and MKI67 genes in the cycling cell population. The stat_cor(method = "pearson") R function was used. Pearson correlation coefficient (R) and p value represents the statistical significance of the linear relationship between two gene expression, $n = 6$. (D) The correlation plot shows a group of genes that coexpress with DiPRO1 in pediatric tumors ($n = 327$). Corrplot R function (M, method = "shade", order = "AOE"). (E) The heatmap of DiPRO1 expression and its positively and negatively correlated genes distinguishes a cluster that includes RMS, Ewing sarcoma (ES), medulloblastoma (MB), and neuroblastoma (NB). Heatmap parameters: Euclidean distribution, complete clustering linkage, $n = 327$. (F) Boxplots of DiPRO1 gene overexpression in RMS ($n = 37$), Ewing sarcoma (ES, $n = 26$), medulloblastoma (MB, $n = 19$), and neuroblastoma (NB, $n = 26$) tumors colored in red versus other pediatric tumors (Other, $n = 220$, Appendix Table S2) colored in blue. Data were presented as median (Q2) ± Q3-Q1, whiskers extend to 1.5x IQR. *$P < 0.05$; **$P < 0.01$; ***$P < 0.001$. P values (DESeq2 R package) are shown between the indicated tumor type and other samples. (G) Single nucleotide variations (SNVs) of DiPRO1/ZNF555 in pediatric tumors. The analysis was performed by exploring whole-exome sequencing data on pediatric tumor specimens. HGG high-grade glioma. (H) Integrative analysis of the association between DiPRO1 copy nimbler variation (CNV) and differential gene expression. XY scatter plot shows the relationship between Z score (y-axis) and copy number value (x-axis) in 101 pediatric tumor samples. The correlation curve is shown in blue. Spearman's correlation coefficient (r) of the nonlinear regression fit and p value (*$P < 0.05$) with Gaussian approximation between copy number and mRNA expression are indicated. Data information: In (D–F), RNA-seq data of primary tumor biopsies were processed using DEBrowser, DESeq2 (TMM normalization, parametric fit, LRT test) and ggplot2 R package, FC >1.4, pV < 0.05. Source data are available online for this figure.

## DiPRO1 upregulation in recurrent pediatric tumors. Clinical relevance

To understand the clinical relevance of DiPRO1 in RMS tumors, we examined its expression in six primary cultures (3 eRMS and 3 aRMS) derived from patient-derived xenografts (PDXs). The data from single-cell analysis were extracted from GSE174376 (Danielli et al, 2023, Data ref: Danielli et al, 2023) and reanalyzed. According to the cell markers, the clustering analysis separated the cells into different subpopulations (Figs. 5A and EV3A), including mesenchymal/muscle stem cells (MuSC) expressing PAX7 and mesenchymal marker IGFBP7 (Barruet et al, 2023), cycling cells expressing CDC20/PAX3, differentiated cells expressing MYOG, and populations of cells in the S-phase expressing MCM2, and G1 lacking cell type-specific gene expression. This aligns with the analysis published by Schäfer' team (Danielli et al, 2023). The DiPRO1 expression analysis revealed a striking expression enrichment in the cycling cell subpopulation (Fig. 5B), which was highly correlated with MKI67 gene expression ($p = 0.0089$) (Fig. 5C). Similarly, expression correlation was observed between DiPRO1 and PAX3 expression (Fig. EV3B), in accordance with the results observed in in vitro models with modulated DiPRO1 expression (Fig. 2L). Together with cycling cell subpopulations, DiPRO1 expression was enriched in mesenchymal/muscle progenitors and differentiated tumor cells (Fig. EV3C). Cycling cells have been reported to correlate with the worst patient prognosis (Repo et al, 2020), and mesenchymal muscle-like stem cells driving relapse were reported among new insights into therapeutic targeting, suggesting that DiPRO1 could be a potential biomarker for RMS prognosis (Patel et al, 2022; Wei et al, 2022). It is important to note that current single-cell technology does not fully cover transcription factors with low expression levels. The expression of DiPRO1 was detected in only 10% of cells (Fig. EV3C). Therefore, a more sensitive technology is required for a comprehensive profiling of DiPRO1 and similar transcription factors.

Therefore, to gain further insight into the role of DiPRO1 in driving mesenchymal cancer relapse and to extend its role to other pediatric cancers, we explored whole-exome and bulk RNA sequencing data from the international precision medicine MAPPYACTS trial (NCT02613962) (Berlanga et al, 2022). The analysis cohort included 327 patients (median age 13 years),

including 37 patients with RMS and 25 patients with Ewing sarcoma, who underwent tumor biopsy/surgery for their recurrent/refractory malignancy (Appendix Tables S1, 2).

First, we performed a cross-sectional analysis of DiPRO1 expression and its downstream gene network that showed the highest fold change (FC >1.4, pV < 0.05) in in vitro DiPRO1 KD experiments performed in RMS and Ewing sarcoma cells (Fig. EV4A). Compared with other genes, DiPRO1 showed modest expression level and expression fold change in cancers (Fig. EV4B), as the majority of KZFP (Wolf et al, 2020). To elucidate the transcriptional network of DiPRO1 in patient tumor samples, we performed correlation analysis. We found a specific cluster of genes that were positively co-regulated with DiPRO1 across all tumors (Fig. 5D). Interestingly, the DiPRO1-gene cluster explicitly recognizes not only the expected tumor types, namely RMS and Ewing sarcomas, but also medulloblastomas and neuroblastomas (Fig. 5E), showing the most significant upregulation of DiPRO1 in these particular tumors (Fig. 5F). Of note, the expression level of DiPRO1 was comparable between ERMS and ARMS clinical subtypes (Fig. EV4C), suggesting that its expression is not linked to the PAX3/7-FOXO1 gene fusion, a hallmark of ARMS (Marshall and Grosveld 2012). DiPRO1 and its network were then analyzed on patient-derived cancer cell lines from the Broad Institute collection (Fig. EV4D–F; Dataset EV4A). The expression pattern was consistent with that observed in primary tumors.

We next addressed whether abnormal DiPRO1 expression could be caused by somatic mutations or copy number variations. According to our study, no somatic DiPRO1 mutations were detected in RMS patients, whereas three single nucleotide variants (SNVs) were found in exon 4 of two patients, an exonic stop-gain mutation (exon 4: c.1203 C → A) in an Ewing sarcoma and two synonymous SNVs (exon 4: c.766 T → C and c.869 C → T) in high-grade glioma (Fig. 5G; Dataset EV4B). These variations of uncertain significance (InterVar prediction) did not alter DiPRO1 gene expression compared with other samples within the same cancer type and can, therefore, be considered non-functional. We additionally explored external COSMIC (Catalog of Somatic Mutations In Cancer) data (Tate et al, 2019) for fusions, translocations, and mutations of the DiPRO1 gene and found no alterations in its coding sequence in human cancer, which was consistent with our results. Integrative analysis of copy number

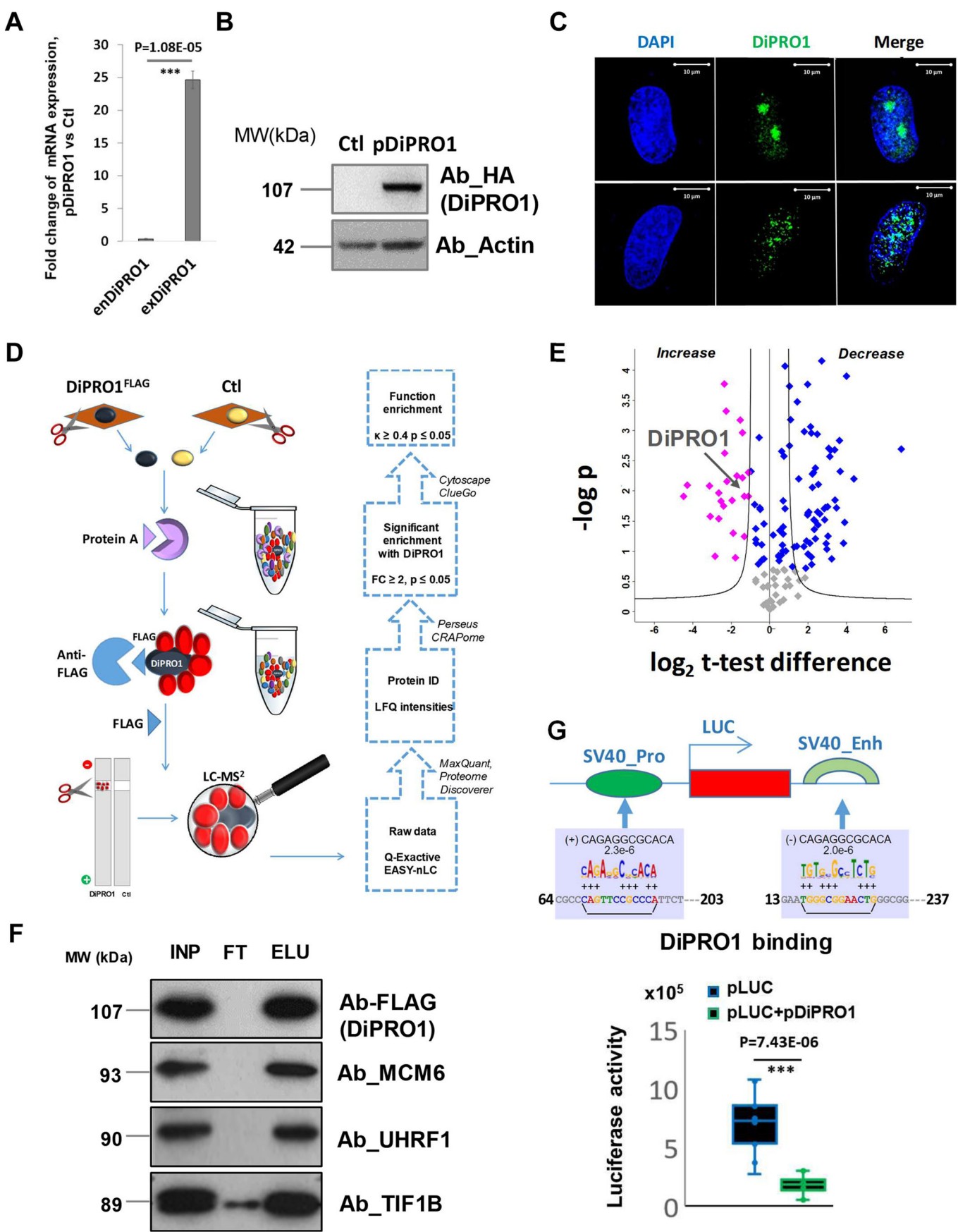

**Figure 6. DiPRO1 transcriptional activity and interaction partners.**

(A–G) The RMS (TE671) were transiently transfected with a retrovirus vector expressing an ORF of DiPRO1 and FLAG, 6-His, and HA tags (pDiPRO1$^{FLAG, 6-His, HA}$). (A) The efficiency of exogenous DiPRO1 overexpression was verified by mRNA expression analysis 24 h post-transfection. The RT-qPCR analysis was performed using the primers for exogenous (exDiPRO1) and endogenous DiPRO1 (enDiPRO1) cDNA. The qRT-PCR results were normalized to the mRNA expression of the GAPDH gene, with the control (Ctl) set to 1. The data represent the FC ± SD of three independent total RNA extractions. Ct = 45 was set up for exDiPRO1 ORF expression in control cells. t-test was used for statistical analyses. ***$P < 0.001$ indicates the difference between endogenous and exogenous DiPRO1 expression. (B) Immunoblotting assays were performed with whole-cell extracts from RMS cells overexpressing DiPRO1 (pDiPRO1) or vector control (Ctl). The DiPRO1 protein level was analyzed using an anti-HA antibody. The actin was served as an internal control. (C) Confocal microscopy analysis revealed the expression of DiPRO1-fused YFP. Scale bar: 10 μm. (D) Sample overview and methodological workflow. Nuclei extracted from RMS cells stably expressing pDiPRO1$^{FLAG}$ and their isogenic counterparts, lacking expression of the bait protein (Ctl), were pre-cleared with protein A beads, immunopurified with an anti-FLAG affinity column, and eluted by adding FLAG-competitor. Eluates were resolved by SDS-PAGE and the corresponding protein bands were excised, processed, and analyzed by mass spectrometry (LC-MS2). IDs—identifications; LFQ—label-free quantification. (E) The volcano plot represents proteins that were differentially enriched (blue dots) between groups of RMS cells expressing DiPRO1 (pDiPRO1) and parental cells (Ctl). Pink dots correspond to proteins enriched in the DiPRO1 complex. A two-sample t-test was applied with threshold s0 = 2 and FDR = 0.05. Data represent independent biological replicates (n = 2–3). (F) Identification of MCM6, UHRF1, and TIF1B proteins in DiPRO1-precipitates by Western blot. The nuclear extracts from pDiPRO1$^{FLAG}$ RMS cells were immunoprecipitated with antibodies against FLAG followed by immunoblotting with antibodies against FLAG, MCM6, UHRF1, and TIF1B. INP input, FT flow-through, and ELU DiPRO1-immunoprecipitated elution. (G) Overexpression of DiPRO1 inhibits the activity of the gene enhancer and promoter. The pDiPRO1 was cotransfected with a luciferase reporter construct (pLUC) containing the SV40 promoter (SV40pro) and SV40 enhancer (SV40e) sequences. Data were scaled and the luciferase signal from the pGL3 base vector was subtracted. Results represent four experiments performed in triplicate, median (Q2) ± Q3-Q1, whiskers extend to 1.5x IQR. The t-test for differences between transfections with and without pDiPRO1 was performed, n = 10–12. Source data are available online for this figure.

variations (CNVs) and differential gene expression in pediatric cancers showed that DiPRO1 DNA copy number was altered in 29.4% (101 of 343) of tumor samples, showing a weak correlation ($r = 0.30$) (Ohshima et al, 2017) with DiPRO1 gene expression level (Fig. 5H; Dataset EV4C). The frequency of high copy numbers of the DiPRO1 gene with high gene expression was remarkably low (2.3%) compared with conventional oncogenes (Ohshima et al, 2017; Shao et al, 2019), albeit among these eight samples, three were from RMS and one from medulloblastoma.

Overall, the single-cell expression in PDX-derived RMS primary cultures showed a specific expression density of the DiPRO1 gene in cycling cells, mesenchymal MuSC, and MYOG-expressing differentiated tumor cell subpopulations. Furthermore, a pan-cancer analysis of recurrent tumor biopsies supports the consideration of DiPRO1 targeting in mesenchymal tumors, indicating elevated expression levels of DiPRO1. This also suggests its potential role as a prognostic marker for driving relapse. Notably, based on mutational analysis, DiPRO1 does not exhibit the characteristic properties of classical oncogenes.

## DiPRO1 physically interacts with TIF1B and UHRF1

In previous sections, we showed that DiPRO1 can bind promoter, gene body, and intergenic regions enriched with retroviral repeat elements. DiPRO1 regulates large-scale transcriptional networks and its downregulations leads to cancer cell death. To elucidate the molecular mechanism by which DiPRO1 regulates its transcriptional network, we sought to determine the partners with which DiPRO1 interacts. To this aim, the pDiPRO1 vector expressing the DiPRO1-YFP protein with FLAG and HA tags was transfected into RMS TE671 cells and the expression of exogenous DiPRO1 mRNA and protein was verified (Fig. 6A,B and Methods section). Consistent with its predicted role in transcriptional regulation, DiPRO1 is strictly localized in the nucleus (Fig. 6C). Accordingly, the affinity purification coupled with mass spectroscopy (MS) was performed using nuclear fraction (Fig. 6D). Pulldown analysis revealed 24 proteins (cutoff s0 = 2 and FDR = 0.05) enriched by the FLAG-tagged DiPRO1 bait protein in the nuclei of RMS cells expressing DiPRO1 (Fig. 6E). The putative DiPRO1 interacting

candidates were then filtered on the basis of (1) at least two identified peptides, (2) significantly higher intensity of the identified peptides in DiPRO1-expressing cells compared with controls, (3) low CRAPome score, and (4) nuclear localization. Among ten selected proteins (Dataset EV5), the KRAB co-repressor TIF1B/KAP1 and UHRF1/NP95 appeared in the precipitated complex with DiPRO1. It should be noted that, in addition to gene expression regulation, TIF1B and UHRF1 play a critical role in the epigenetic silencing of retroviral elements (Rowe et al, 2013; Nakamura et al, 2016; Dong et al, 2019). In addition, four proteins of the hexameric mini-chromosome maintenance complex (MCM) were found enriched in the DiPRO1 protein complex, confirming the involvement of DiPRO1 in cell cycle control that we previously demonstrated. The physical interaction of DiPRO1 with TIF1B, UHRF1 and MCM6 was validated by anti-FLAG immunoprecipitation (IP) and Western blot analysis in RMS cells (Fig. 6F).

We previously reported that the inhibition of DiPRO1 led to the partial loss of β-satellite enhancer activity (4qAe), indicating a transcriptional activator characteristic of DiPRO1 (Kim et al, 2015). Based on the properties of the interacting module proteins, TIF1B (Iyengar and Farnham 2011) and UHRF1 (Alhosin et al, 2011), we hypothesized that DiPRO1 might be a transcriptional repressor. According to the motif prediction, DiPRO1 may have binding sites in the SV40 promoter and enhancer and thus may regulate gene transcription through viral cis-elements. To verify this, we applied luciferase reporter assays using the pGL3 reporter vector containing the SV40 promoter (Pro) and enhancer (Enh). The base vector without promoter and enhancer was used for normalization. The vectors were transfected to RMS cells alone or cotransfected with pDiPRO. The results of the chemiluminescence signal quantification showed that DiPRO1 strongly reduced reporter gene expression (Fig. 6G), indicating that DiPRO1 could repress enhancer/promoter-mediated gene expression.

## Distinct DiPRO1-mediated CGI methylation regulation in cancer and normal muscle cells

Our results in RMS cells established the physical interaction of DiPRO1 with TIF1B and UHRF1. Both proteins can participate in the regulation of CpG islands (CGI) methylation (Liu et al, 2013;

Rowe et al, 2013). While TIF1B plays the role of a scaffold to recruit DNA methyltransferases (DNMTs) (Turelli et al, 2014), UHRF can bind directly to methylated DNA via its SRA domain (Stirzaker et al, 2017). Thus, we hypothesized that DiPRO1 may contribute to gene and regulatory element (RE) regulation via CGI methylation. To characterize this relationship, we applied the methylated CpG island recovery assay sequencing (MIRA-seq) to RMS (TE671) and myoblast cells transfected with shDiPRO1- and shCtl-expressing vectors 48 h following infection. Consistent with previous studies, the upstream CpG regions of the PAX3 (Kurmasheva et al, 2005) and MYOD1 (Chen et al, 1998) genes were methylated, while those of the fibroblast growth factor receptor 1 (FGFR1) (Goldstein et al, 2007) and JUP (γ-catenin) (Gastaldi et al, 2006) genes were unmethylated in control RMS cells (Fig. EV5A). The myoblasts under control conditions (shCtl) exhibited methylation at the CpG island (CGI) promoter regions of MYOG and PAX3, and demethylation at PAX7, consistent with published findings (81,82) (Fig. EV5B). Subsequent analysis demonstrated an enrichment of 49.8% (ENR = 27,425) and 55.4% (ENR = 30,494) of CGI methylation regions in control RMS and myoblasts, respectively, compared to the corresponding inputs (INP) (Fig. EV5C). DiPRO1 knockdown (shDiPRO1) resulted in a reduction of methylated regions by 22.2 and 36.9% in RMS and myoblasts, respectively, with a more pronounced effect observed in myoblasts. This led to the identification of 3751 and 772 differentially methylated unique CGIs (DMCs) in shDiPRO cells, denoted as hypermethylated (hyper-DMCs), and 10,982 and 18,441 DMCs in control RMS and myoblast cells, respectively, denoted as hypomethylated DMCs (hypo-DMC) versus shDiPRO1 (Fig. 7A; Dataset EV6). These findings collectively signify a genome-wide involvement of DiPRO1 in CGI methylation, with its downregulation resulting in CGI hypomethylation in both cancerous and normal proliferating muscle cells.

Genomic distribution analysis demonstrated that ~60% of differentially methylated events in both shDiPRO1 and shCtl RMS cells were situated within core gene regions, comprising a population of long housekeeping-CGIs (hk-DMC) with an average length of 777 bp and a population of short repeat-associated CGIs (rep-DMC) with an average length of 309 bp (Fig. EV5C–E). The remaining 40% originated from intergenic regions, primarily represented by rep-DMCs (Fig. 7B). Previous investigations revealed that DiPRO1 can directly interact with diverse classes of repetitive elements (REs). Our exploration into which classes of repeats are influenced by DiPRO1 KD showed that hypermethylation was predominantly associated with SINE repeats (92%) (Fig. EV5G), whereas hypomethylation was discerned in various repeat classes (SINE, LINE, LTR/ERV) (Fig. 7B), aligning with the DiPRO1 binding analysis. A parallel pattern emerged in myoblast cells, with a slightly heightened proportion of hypomethylation in intergenic regions (47%) and a substantial impact on ERV repeats due to DiPRO1 KD (Fig. EV5F,H). This implies that DiPRO1 binding to REs may contribute to their transcriptional silencing through methylation. Functional analysis of hypo-DMCs based on KEGG pathways indicated enrichment in pathways associated with cancer, virus infection, axon guidance, neurodegeneration, and endocytosis in RMS cells (Fig. EV5I). Notably, within the viral infection pathway, an extensive array of Zinc Finger Proteins (ZFPs) and Krüppel-associated box domain-containing Zinc Finger Proteins (KZFPs) was overrepresented on chromosome 19,

according to Positional Gene Enrichment (PGE) analysis (De Preter et al, 2008) (Fig. 7C). This phenomenon was similarly observed in DiPRO1 KD myoblasts, emphasizing a distinctive impact of DiPRO1-related demethylation on chromosome 19.

To delineate the functional distinctions arising from methylation alterations due to DiPRO1 KD in RMS and myoblast cells, we systematically examined the cell-specific methylation patterns and identified shared events. Moreover, we investigated the correlation between methylation status and gene expression changes, with a specific focus on hypomethylation, given its prevalence following DiPRO1 KD. CpG methylation is traditionally thought to be linked to gene repression (Tamaru and Selker 2001). Consequently, we juxtaposed hypomethylated DMCs with upregulated DEGs in both DiPRO1 KD and control cells. Approximately 12% of differentially methylated regions (DMCs) exhibited a concurrent correlation with changes in gene transcription in both RMS and myoblast cells (Appendix Fig. S5A). Intriguingly, these DMCs overlapped with both hk-DMCs and rep-DMCs, implying a potential contribution of the rep-CGI population to gene expression. This observation aligns with prior research on the epigenetic regulation of DNA repeat silencing by KZFP (Imbeault et al, 2017; Turelli et al, 2020) and the role of DNA repeats in modulating gene expression (Garrick et al, 1998). As depicted in Fig. 7A and Dataset EV6, 49.9% of the hypo-DMC regions in RMS cells overlapped with those in myoblasts. The remaining 51.1% were RMS-specific, giving rise to 578 differentially methylated genes (DMGs) exclusively upregulated in RMS cells. Notably, these DMGs were prominently associated with TNF-α signaling via NFkB (FDR = 1.15E-08) based on gene set enrichment analysis (Liberzon et al, 2015, Data ref: Liberzon et al, 2015) (Fig. 7D). Within this pathway, the adapter protein TRAF1 and the TNFSF9, a ligand for TNF-receptor TNFRSF9, emerged as key components (Fig. 7E) (Zapata et al, 2018). Their interaction with TNFRSF9 receptor is crucial for transmitting signals that activate NF-kappaB, an anti-apoptotic factor. Simultaneously, TRAF1 acts as a specific target of activated caspases, contributing to apoptosis induction in cancer cells and serving as a link between caspases and TNF receptors (Leo et al, 2001). Studies have reported that TNFRSF9 suppresses tumor progression in breast cancer (Liu et al, 2022) and that TNFSF9 reverse signaling induces apoptosis in non-small cell lung cancer cells (Qian et al, 2015), suggesting a potential role in the antitumor activity of DiPRO1. Transcriptome analysis reveals a distinct shift in the expression pattern of these genes, transitioning from low expression in RMS control cells to elevated expression in DiPRO1 KD of RMS and ES cells (Fig. 7F) and a partial upregulation in myoblasts. This partial effect in myoblasts correlates with common DMGs (n = 321) that were hypomethylated and upregulated in both cell types (Fig. 7A,D). Principal component analysis (PCA) illustrates that the gene signature related to TNF-α signaling via NFkB distinctly separates control samples from DiPRO1 knockdown (KD) samples in RMS and ES cells, whereas no clear segregation is observed in myoblasts (Fig. 7G). This observation implies the specific activation of this pathway in mesenchymal cancer cells. Remarkably, the myoblast-specific hypo-DMGs (n = 945) exhibited significant enrichment in myogenesis (FDR = 3.11E-20), consistent with the results of the transcriptome functional analysis discussed in the previous section (Fig. 2K). This set includes genes such as CHRNA1, SPEG, MYOG, and skeletal muscle troponins TNNI1, TNNC2, and TNNT3. The upregulation of the inflammatory CASP5 gene (55) was associated with hypomethylation of proximal CGIs in both RMS and myoblasts.

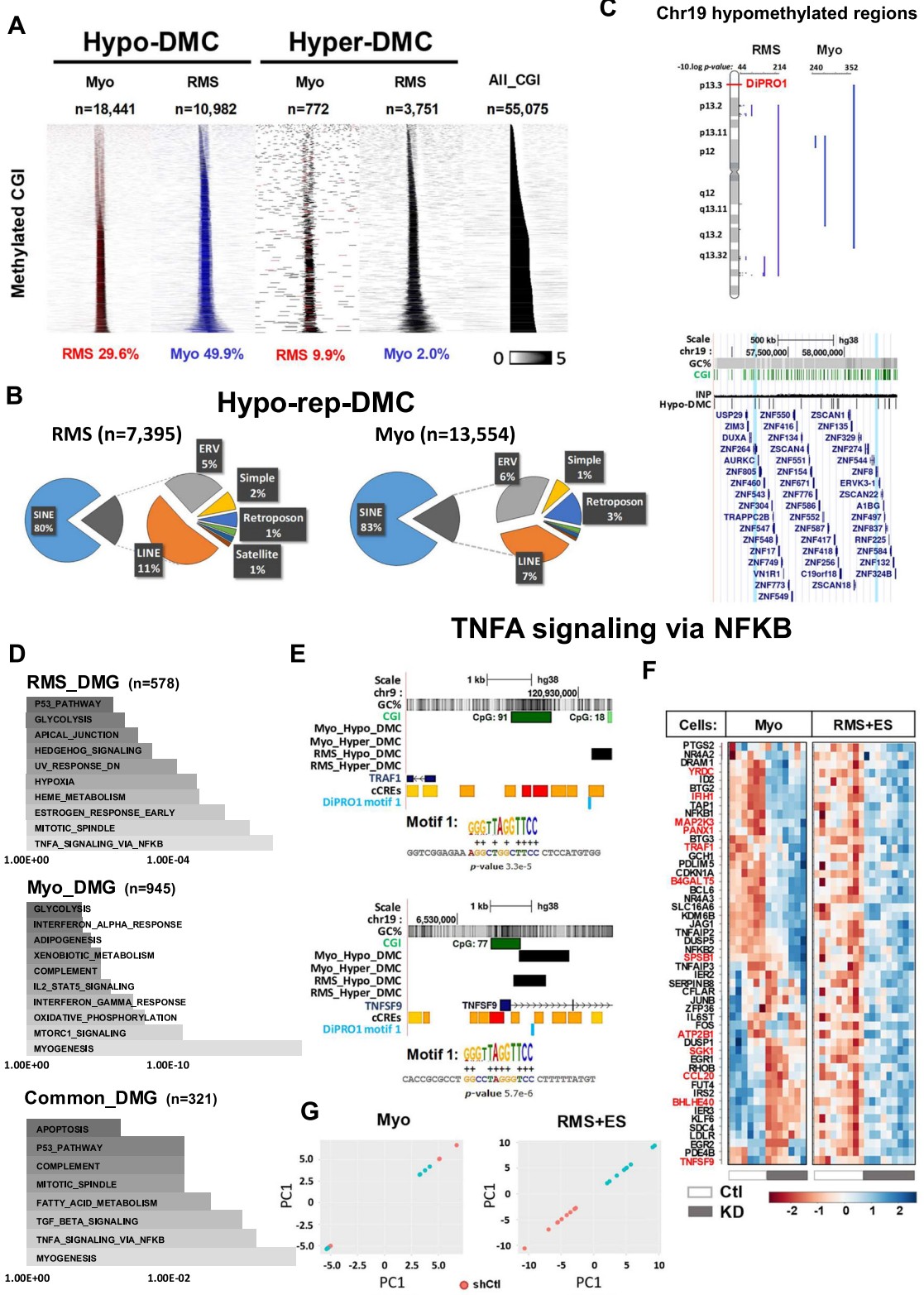

**Figure 7.  Methylation alterations in different CpG island populations associated with DiPRO1 inhibition in RMS and myoblast cells.**

(A) Heatmaps displaying size-sorted hypo- and hypermethylated DMC in shDiPRO1 versus shCtl. Overlapped myoblast (Myo) regions in RMS CGIs are colored blue and overlapped RMS regions in myoblast CGIs are colored red. The percentage of overlap is indicated. The bar reflects signal intensity. Y-axis: DNA fragments per 1 M reads per 1 K. X-axis: surrounding area corresponding to 500% of each CGI region, segmented into 200 bins. (B) DNA repeat class distribution within DiPRO1-linked hypomethylated DMC regions in RMS cells. (C) Hypomethylated DMC signals are overrepresented in chromosome 19. The analysis was performed using the positional gene enrichment (PGE) tool. (D) Hallmark enrichment of genes linked to differentially methylated genes (DMGs) was identified in myoblast and RMS cells following DiPRO1 knockdown. (E–G) Enrichment of TNFA Signaling via NFKB pathway in hypomethylated DMGs of RMS cells. (E) TNFSF9 and TRAF1 genes were hypomethylated in TSS regions and upregulated in RMS cells. The prediction analysis using the FIMO tool Version 5.5.4 identified the presence of potential DiPRO1 binding motifs within the GGI, aligning with the proximal enhancers. CGI, GC (%), cCRE (cis-regulatory elements), and DMC tracks over the TNFSF9 and TRAF1 loci. (F) Heatmap expression of the TNFA signaling via NFKB pathway gene set across control and DiPRO1 KD samples of RMS and Ewing sarcoma cells ($n = 9$) and myoblasts ($n = 6$-7). Clustering analysis of Euclidean distribution, complete linkage. (G) PCA analysis of DEGs of the TNFA signaling via NFKB pathway of DiPRO1 KD in RMS and Ewing sarcoma cells ($n = 9$) and myoblasts ($n = 6$-7). Data Information: In (A–C, E), methylation profiles were assessed using MIRA-seq in DNA samples from RMS TE671 and immortalized human myoblast (Myo) cells. The signals from methylation-enriched DNA (ENR) were normalized to unenriched input DNA (INP). CpG islands from cells expressing a DiPRO1-targeting shRNA (shDiPRO1) were compared with control cells expressing a nontargeting shRNA (shCtl), and differentially methylated CpG islands (DMC) were analyzed between normal and cancer muscle cells. In (D, F, G), global transcriptome analysis by microarray was implemented using total mRNA extracted from RMS and Ewing sarcoma cell lines: with DiPRO1 knockdown (shDiPRO1) and corresponding controls (shCtl). Three to six separate extractions from two independent experiments were performed to collect the replicates ($n = 9$ per condition). Differentially expressed genes in shDiPRO1- cells versus corresponding controls ($P < 0.05$) were analyzed using the Limma R package. CGI, CpG islands according to Gardiner-Garden and Frommer criteria. Source data are available online for this figure.

Among the pro- and anti-apoptotic hallmarks, 48 genes in RMS and 65 genes in myoblasts showed proximal hypomethylation, leading to altered expression of some of them. ROCK1 and PMAIP1 were identified as unique RMS DMGs, while DNAJA1, BMP2, and DCN showed exclusive DMG profiles in myobalsts. Notably, these myoblast-exclusive DMG may play an additional role in the development and differentiation of normal muscle cells, as reported (Katagiri et al, 1994; Barruet et al, 2020; Liu et al, 2021).

Our results, therefore, support the hypothesis that DiPRO1 may act as a regulator of CGI methylation. Depletion of DiPRO1 leads to widespread demethylation of CGIs in both normal and cancerous muscle cells, affecting gene core regions and DNA repeats. This phenomenon is associated with the activation of viral infection pathways, myogenesis, and apoptosis. DiPRO1 plays a pivotal role in the methylation of the inflammation pathway in RMS and myogenesis in myoblasts, underscoring a distinct methylation pattern between cancerous and normal muscle cells.

## DiPRO1 and SIX1 crosstalk in mesenchymal cancer

Because DiPRO1 shares common targets with SIX1 and regulates the myogenic transcriptional program in normal muscle cells, we wondered whether the DiPRO1 and SIX1 gene regulatory network is common to mesenchymal cancer cells. Using SIX1 targets from harmonized ChIP-seq, ChIP-exo, DNase-seq, MNase-seq, ATAC-seq and RNA-seq data (Kolmykov et al, 2021, Data ref: Kolmykov et al, 2021), we showed that 12.4% of promoter regions bound by SIX1 overlap with DMCs in DiPRO1 KD with predominance in hypomethylated regions (10.0%) (Fig. 8A). This supports the possibility of DiPRO1-linked methylation in SIX1 binding regions. Using the recently published dataset of SIX1 KD in human RMS (Hsu et al, 2022, Data ref: Hsu et al, 2022), we compared its transcriptional network with that of DiPRO1 KD. The overlapping pattern of upregulated genes in DiPRO1 KD corresponds to 8.2% in RMS cells and 10.8% in myoblasts. As expected, these genes were enriched in the main hallmarks of myogenesis (Fig. 8B). As illustrated in the Venn diagram (Fig. 8C), the myogenic signature of SIX1 KD cells closely resembled that of DiPRO1 KD myoblasts, including upregulation of MYOG, ACTA1, and the troponin subunits TNNC1/2, TNNT3, and TNNI1. In RMS cells with SIX1

KD and DiPRO1 KD, we found commonly increased levels of the embryonic MYH3 gene, early myogenic genes MEF2a/c, and contractile muscle genes MYOZ1, MYOM1, MYL1, and TNNT2, showing that both proteins in RMS slow down the differentiation program. Furthermore, the depletion of DiPRO1 induced upregulation of additional myogenic genes DMD, TCAP, and MYF6/MRF4, which are important players in muscle differentiation and contraction. A very limited number of myogenic genes were upregulated in a pooled pattern across ES and RMS cells upon DiPRO1 KD. While MYOM2 is a striated muscle-specific gene, encoding a sarcomeric protein, the MAPRE3/EB3, AK1, ABLIM1, and BHLHE40/DEC1 genes share neuronal and muscle functions (Fig. EV3D). RMS cells include the subpopulations of neural progenitors, along with cells expressing muscle developmental features (Wei et al, 2022).

We have previously reported that murine SIX1 controls Myogenin activity via the evolutionarily conserved MEF3 motif (Spitz et al, 1998). The MEF3 consensus (TCAGGTTTC) (Santolini et al, 2016) (Fig. 1F) is highly prevalent among target genes of myogenic regulatory factors (MRF) (Liu et al, 2010), playing a crucial role in myogenesis. It has been demonstrated to play a role in the transcriptional regulation of the cardiac troponin C gene (Parmacek et al, 1994) and the muscle-specific promoter of aldolase A (Hidaka et al, 1993; Spitz et al, 1997). We investigated whether DiPRO1 is able to bind to human MEF3 motifs. The MEF3 motif of skeletal muscle hAldolase A promoter is closely located to the DiPRO1 binding site and the CGI region differentially methylated by DiPRO1 repression in RMS cells (Fig. 8D). Electrophoretic mobility shift assay (EMSA) performed with nuclear extracts from RMS cells overexpressing DiPRO1-HA, using either the hAldolase A, the Myogenin MEF3 sites, or the motif 1 of DiPRO1 (Fig. 1F), revealed four slowly migrating complexes (Fig. 8E). Three of them (BS1, 2, and 4) appeared to be specific for both MEF3 and DiPRO1 motifs, while BS3 appeared not specific. Their formation was inhibited by excess copies of the unlabeled MEF3 fragment and by DiPRO1 motif 1 or by the DiPRO1-binding motif 4qAe (Kim et al, 2015). These complexes were suppressed by anti-HA antibodies targeting the DiPRO1 protein with an HA-tag on the N-terminal. These data demonstrate that human muscle-specific MEF3 complexes are composed of the DiPRO1 proteins in nuclear

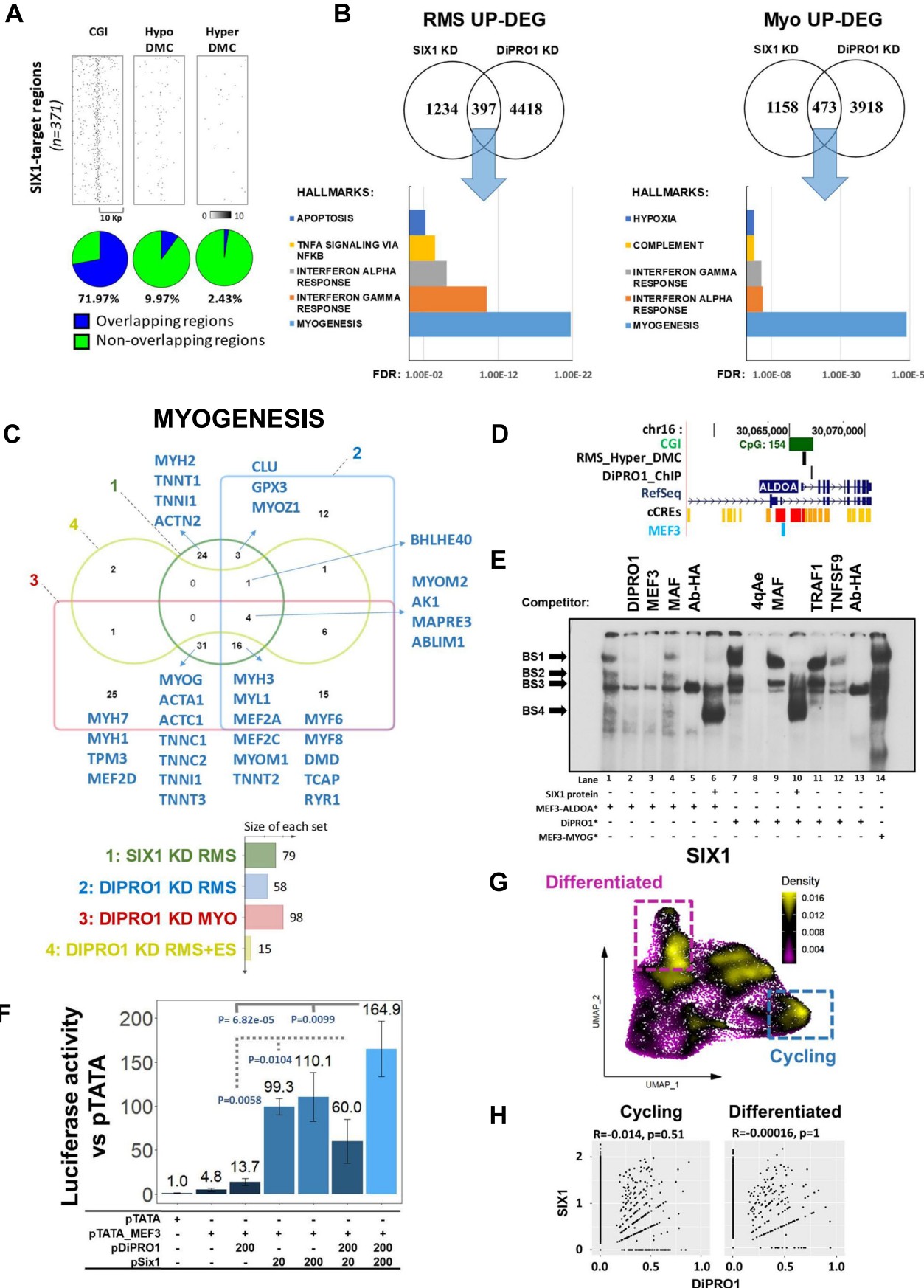

**Figure 8. DiPRO1 and SIX1 KD common signature.**

(A) Heatmaps (top) displaying SIX1 target signals from the GSEA data collection within CGIs of hg38, hypo-, and hypermethylated DMC in DiPRO1 KD RMS. The bar reflects the signal intensity. Y-axis: DNA fragments per 1 M reads per 1 K. X-axis: 10 Kb surrounding each region. Pie chart (bottom) displays overlapping regions. The ChIP-seq dataset of SIX1 KD in RMS cells were retrieved from GSE173155 and uploaded by the IGV browser. (B, C) The SIX1 KD RNA-seq (GSE173155) data of RMS cells were processed using DEBrowser, DESeq2 (TMM normalization, local fit, LRT test). Global transcriptome analysis by microarray was implemented using total mRNA extracted from myoblasts (Myo), RMS, and Ewing sarcoma (ES) cell lines with DiPRO1 knockdown (shDiPRO1) and corresponding controls (shCtl). Three to four separate extractions from two independent experiments were performed to collect the replicates. Differentially expressed genes in shDiPRO1- cells versus corresponding controls ($P < 0.05$) were analyzed using the Limma R package. (B) Hallmark enrichment analysis of shared upregulated DEGs in SIX1 KD ($n = 4$) and DiPRO1 KD in RMS (left, $n = 3$) and myoblasts (right, $n = 6$–7) using MSigDB database. (C) Jvenn diagram of the myogenesis gene set shows a joint signature of SIX1 KD and DiPRO1 KD, $n = 3$–7. (D) The skeletal muscle Aldolase A (ALDOA) gene containing an MEF3 motif in the promoter region was hypermethylated in DiPRO1 KD RMS cells at the DiPRO1 binding motif. CGI, DiPRO1 ChIP-seq, cCRE (candidate *cis*-regulatory elements) and hypermethylated DMC tracks over the ALDOA locus. (E) DiPRO1 binds directly to MEF3 motifs. Competition EMSA evaluation of the specific binding of DiPRO1 motif 1 (lanes 7–13) and MEF3 motifs from ALDOA (lanes 1–6) and MYOG (lane 14) with DiPRO1 protein from nuclear extract of RMS cells overexpressing pDiPRO1-HA. Binding specificity was examined by competition in the binding reaction between nuclear proteins and labeled oligoprobes by adding a 300-fold molar excess of homologous unlabeled probes (MEF3) or heterologous probes DiPRO1, MAF, 4qAe, TRAF1, and TNFSF9. In vitro-translated SIX1 protein was added to pDiPRO1-RMS nuclear extracts and incubated with DiPRO1 and ALDOA-MEF3 probes (lanes 6, 10). EMSA Supershift was performed by pre-incubating pDIPRO_HA RMS nuclear extract with anti-HA Ab. Band shifts 1 and 2 (BS 1/2) were specific for DiPRO1 and band shift 4 (BS4) was specific for SIX1. (F) The trans-acting properties of DiPRO1/SIX1/MEF3 complexes were investigated. Multimerized MEF3 motifs, positioned upstream of a minimal TATA box to drive a firefly luciferase reporter, were co-expressed with pDiPRO1 alone (200 ng) and/or with increasing amounts of the transcriptional activator pSix1VP16 (20 ng and 200 ng) in E18.5 Six1/4/5 KO primary mouse myoblasts. Luciferase expression was analyzed 48 h post-transfection. The data represent results from triplicates in two experiments ($n = 6$), normalized against the renilla luciferase reporter vector, and the firefly luciferase expression with pTATA-box was set to 1. The data were shown as m±SD. Welch's two-sample *t*-test was applied for statistical difference. Boxplots are colored according to the average mean (indicated at the top). (G, H), scRNAseq datasets (GSE218974) derived from PDX primary RMS cultures ($n = 3$ eRMS and $n = 3$ aRMS) were reanalysed using the Seurat R toolkit. The individual data were normalized, feature-selected, and integrated into the individual data samples using the SCTransform function. (G) Integrated UMAP plot showing SIX1 expression density across RMS cell populations. Analysis was performed using the R package scCustomize ($n = 6$). (H) Coexpression curves for DiPRO1 and SIX1 genes in differentiated and cycling RMS cell populations. The stat_cor(method = "pearson") R function was used. Pearson correlation coefficient (R) and *p* value represents the statistical significance of the linear relationship between two gene expression, $n = 6$. Source data are available online for this figure.

extracts from RMS cells overexpressing DiPRO1-HA. The presence of several complexes could be explained by the presence of different DiPRO1 complexes with additional proteins from RMS nuclear extracts, or it could reflect the processed or degraded DiPRO1 protein, as punctually observed by Western blot. The addition of SIX1 protein extract resulted in a supplementary fast-migrating complex (migrating like the BS4 DiPRO1 complex) and weakened the formation of the BS1/2 complexes, suggesting a higher affinity of SIX1 for the MEF3 motif (lanes 6 and 10). The prediction analysis identified the presence of a potential DiPRO1 motif 1 within the GGI, corresponding to the proximal enhancers of TRAF1 and TNFSF9 genes (Fig. 7E). These genes exhibited CGI hypomethylation and upregulation in the context of DiPRO1 knockdown. Competition by the TNFSF9 motif, but not by the TRAF1 motif, considerably reduced the DiPRO1 complexes (lane 12), suggesting that DiPRO1 could have a strong affinity for the TNFSF9 motif.

We then assessed whether DiPRO1 influences the MEF3 cis-regulatory elements in vitro and whether the combined contributions with SIX1 operate in an additive, cooperative, or repressive manner. Primary myoblasts from mouse E18.5 Six1/4/5 KO fetuses which express neither endogenous Dipro1 nor Six1, Six4 or Six5, were used for this study. Multimerized MEF3 motifs (6xMEF3) positioned upstream of a minimal TATA box to drive a firefly Luciferase reporter (Grifone et al, 2004) were co-expressed with expression vectors pDiPRO1 and/or the transcriptional activator pSix1VP16 in increasing amounts (Brugmann et al, 2004). Co-transfection of either 200 ng of DiPRO1 or 20 and 200 ng of Six1VP16 alone was sufficient to activate the MEF3 motifs, as shown in Fig. 8F. However, although significant, the activation by DiPRO1 remains weak (2.9-fold increase) compared to the 20.7- and 22.8-fold increases induced by different doses of the Six1VP16 activator, respectively. Coexpression of both factors in equal doses resulted in a 34.4-fold increase in MEF3 activity, suggesting their

additive combined action (Cunha et al, 2010). Conversely, the MEF3 motifs, strongly activated by Six1VP16 alone, were partially blocked (1.7-fold decrease) by DiPRO1 when the dosage of Six1VP16 was reduced by a factor of 10, suggesting that DiPRO1 may attenuate the effect of Six1VP16 transcriptional activation in a dose-dependent manner. Therefore, these findings, consistent with the EMSA experiments, support the ability of DiPRO1 to independently regulate MEF3 motifs in vitro and to result in dose-dependent reporter activity in combination with SIX1, confirming the in silico prediction.

The analysis of SIX1 and DiPRO1 gene coexpression in patient-derived primary RMS cultures (Danielli et al, 2023), revealed the highest expression density of both genes in clusters corresponding to cycling (KI-67-marker) and differentiated (MYOG-marker) tumor cells (Figs. 8G and EV3C). Surprisingly, the coexpression analysis did not reveal an expression correlation ($p > 0.51$) (Fig. 8H), suggesting that SIX1 and DiPRO1 genes are expressed in distinct individual RMS cells.

With regard to clinical relevance, MEF3-related genes TRAF1 and TNFSF9, as well as direct *cis*-targets of DiPRO1 (Dataset EV1A,B) that transcriptionally respond to DiPRO1 KD in RMS and Ewing sarcoma cells in vitro (Dataset EV3), were analyzed in biopsies of recurrent tumors. The downregulation of genes associated with TNF-α signaling via NFkB, apical junctions, and responses to interferons and interleukins was observed, in agreement with methylation and transcriptional analysis (Fig. EV3E; Dataset EV4D) (Liberzon et al, 2015, Data ref: Liberzon). This downregulation was enriched in pediatric RMS tumors, and nearly half (48%) of these genes were also common with Ewing sarcoma tumors. Moreover, the upregulation of myogenesis hallmarks, such as the embryonic muscle gene MYH3, PCEN2, COL4A2, and DOK7 in RMS tumors, coupled with the downregulation of CHRNA4 (Niemeyer et al, 2001; Faravelli et al, 2014), N-Cadherin (CDH2) (Marthiens et al, 2002),

LRRC4 (Deng and Wu 2023), and STXBP5 (Carr and Munson 2007) genes in both tumor types, which are implicated in synapse organization (GO:0045202) and neuromuscular junctions, was evidenced. This further corroborates the plausibility of a shared mechanism of neuromuscular regulation by DiPRO1 in mesenchymal tumors. Notably, within this DEG-set, 35 genes (16%) are associated with cis-regulatory regions of SIX1 binding (Fig. 1E; Dataset EV1B).

## Discussion

We provide evidence that DiPRO1 is a transcriptional and epigenetic regulator that compromises normal and malignant cell fate in a selective manner. Functional modulation of gene expression demonstrates that DiPRO1 is a positive regulator of proliferation and a negative regulator of differentiation of human myoblasts whereas it is a positive regulator of proliferation and prevents apoptosis in cancer cells. Besides discrete expression alterations, a feature of KRAB zinc finger proteins (Wolf et al, 2020), DiPRO1 inhibition is toxic to RMS and Ewing sarcoma cancer cells, suggesting that cancer cells require DiPRO1 for their survival. DiPRO1 is not a classical oncogene, its mutations are rarely observed in cancer. Targeting rarely mutated genes that are conserved by cancer cells during clonal evolution has been reported as an alternative strategy for cancer therapy (Gatenby et al, 2014), supporting our hypothesis of defeating cancer cells by inhibiting DiPRO1.

Our findings indicate that DiPRO1 directly interacts with TIF1B, UHRF1, and MCM complex proteins. The KAP1/TIF1B protein is present alongside MyoD and Mef2 at numerous muscle genes, where it acts as a scaffold for recruiting coactivators and corepressors (Singh et al, 2015). UHRF1 undergoes activation in satellite cells, regulating their proliferation and differentiation (Sakai et al, 2022). In myoblast cells, the MCM complex is involved in rendering chromatin operational for DNA replication, a crucial function for myoblasts emerging from quiescence and undergoing proliferation. MyoD directly targets the promoters of MCM complex members 2 and 7, inducing transcription from MCM (Zhang et al, 2010). Upregulation of MCM gene expression has also been demonstrated to be triggered by MEF2A, inducing myoblast proliferation and inhibiting the myogenic differentiation process (Wang et al, 2018). This study presents the first experimental evidence that, similarly to its interacting partners, DiPRO1 is transcriptionally activated in human proliferating myoblasts and acts as a negative regulator in the differentiation. We also demonstrated that DiPRO1 shares direct targets with SIX1, preserving their methylation and interfering with SIX1 binding. SIX1, an upstream regulator of myogenesis (Maire et al, 2020), drives Myogenin activity via the evolutionarily conserved MEF3 consensus (Spitz et al, 1998), which is crucial for MRF target genes (Liu et al, 2010; Santolini et al, 2016). DiPRO1 can independently regulate MEF3 motifs in vitro, exhibiting dose-dependent reporter activity. In combination with SIX1, DiPRO1 can operate repressively or in an additive activation manner. This further underscores the possible role of DiPRO1 in the regulation of myogenesis.

Several studies have reported the role of SIX1 in oncogenesis (Hua et al, 2014; Wu et al, 2015; Hsu et al, 2022) and in the regulation of epithelial-mesenchymal transition (Min and Wei 2021). It can promote tumor growth, invasion, and metastasis. In RMS, SIX1 preserves the undifferentiating state of tumor cells and SIX1 KD leads to myogenic differentiation and impaired tumor growth (Hsu et al, 2022). In our study, DiPRO1 maintains a suppressed muscle differentiation program in myoblasts and RMS cells. Loss of SIX1 or DiPRO1 leads to common myogenic consequences. During mouse hypaxial myogenesis, murine Six1 shows a feedback regulatory loop with Pax3 and is likely to be an upstream positive regulator of Pax3 (Grifone et al, 2005; Wu et al, 2014; Maire et al, 2020). The myogenic alterations due to the mutation of Six1 resemble those of Pax3 mutants (Tremblay et al, 1998; Laclef et al, 2003). In myoblasts and RMS, DiPRO1 positively controls the PAX3 gene expression, and PAX3 is required for RMS cell viability (Bernasconi et al, 1996). In addition to SIX1, previous works dealing with the effects of myogenic regulatory factors on muscle cancer have discovered their capacity to stimulate cancer cell differentiation and deregulation of the cell cycle leading to tumor growth regression (Tenente et al, 2017). However, DiPRO1 seems to play a more prominent role than myogenic regulators in RMS switching them to cell death. The contribution of DiPRO1's interaction partners could advance DiPRO1 roles in mesenchymal tumors. The expression of UHRF1 increases in mesenchymal sarcoma tumors compared to non-sarcomas (Liu et al, 2020), promoting cell invasion and correlating with recurrence and overall survival (Sannino et al, 2017). TRIM28/TIF1B gene expression exhibits strong positive associations with sarcoma tumors (Shang et al, 2023) and serves as a prognostic predictor of immunotherapy resistance, regulating stemness, proliferation, and migration of cancer cells (Yang et al, 2023). TIF1B plays a crucial role in herpes virus-associated tumors, including EBV, Kaposi's sarcoma-associated herpes virus, and human cytomegalovirus(Randolph et al, 2022). The MCM core hexamer complex (MCM2-7) plays an essential role in DNA replication and cell cycle progression, being ubiquitously expressed in proliferating normal and cancer cells (Maiorano et al, 2006). According to our findings, MCM 2/3/4/6 members interact with the DiPRO1 protein. These MCM members have been shown to be highly expressed in sarcoma tumors, proposing them as potential biomarkers for the survival prognosis of sarcoma patients (Zhou et al, 2021).

One theory of carcinogenesis (Holliday 1979) explaining cancer development through epigenetic cell reprogramming has important implications for finding new strategies for cancer therapy (Brown et al, 2020). The DiPRO1 interaction partners UHRF1 and TIF1B have been proposed as targets for cancer therapy (Wang et al, 2005; Sidhu and Capalash 2017; Kong et al, 2019). Both of them can contribute to DNA methylation recruiting, mainly DNMT1 (Liu et al, 2013; Cheng et al, 2014). UHRF1 is a 5mC reader and essential to maintain CGI methylation and associated gene silencing in mammalian cells (Bostick et al, 2007). While UHRF1 can recognize hemi-methylated CGI via its SRA domain, TIF1B does not have a DNA-binding domain and thus requires another protein partner to be recruited to the genome. TIF1B functions as a scaffold coordinating specific gene silencing, targeting specific promoters and cis-regulatory elements via the KZFP (Schultz et al, 2002). UHRF2, a paralog of UHRF1, directly binds to 5hmC. ZNF618 binds DNA adjacent to 5hmC and facilitates UHRF2 deposition, thus regulating UHRF2 chromatin localization (Liu et al, 2016). Both UHRF2 and TIF1B can contribute to DNA

methylation recruiting mainly DNMT1 (Liu et al, 2013; Cheng et al, 2014). According to this cooperation, one of the mechanisms, by which DiPRO1 can attenuate gene expression, is associated with methylation. In our experiments, DiPRO1 depletion led to CGI demethylation in specific regions, suggesting that DiPRO1 might recruit TIF1B and contribute to UHRF1 5mC binding in DiPRO1-targeted loci. This, in turn, could lead to DNMT1 loading and chromatin silencing. Importantly, the interaction between KAP1 and TIF1B has been reported in several studies (Kim et al, 2018; Verdikt et al, 2022), supporting our hypothesis.

The disruption of DiPRO1 cooperation with CGI methylation regulators lead to massive CGI demethylation including epigenetic awakening different classes of RE, including endogenous retrovirus (ERV), in cancer and normal muscle cells. Considered for decades as "junk", DNA repeat *cis*-elements could influence the oncogene and tumor suppressor gene transcription (Zhu et al, 2011). Some of them contribute to the heterochromatin integrity protecting the DNA from damage (Rich et al, 2014). Hence, it is tempting to assume that DNA repeats could constitute an "epigenetic barrier" for an anticancer response. Human ERV activation in tumors drives the synchronized elevation of KZFP expression, presumably leading to tumor suppression (Ito et al, 2020). Increasingly attractive as therapeutic targets in cancer, they stimulate interferon signaling, acting as a viral mimicry response (Jansz and Faulkner 2021). HERV-mediated inflammation is being explored as a way to sensitize tumors to immunotherapy (Gonzalez-Cao et al, 2016). It has been also proposed to explore KAP1 silencing, wherein the expression of lineage-specific KZFPs facilitates the tissue-specific control of HERV (Jansz and Faulkner 2021). In agreement with this, DiPRO1 loss induces a massive activation of ZFP, including a large amount of KZFP, mimicking a virus infection response via the interferon signaling pathway. ZFP of vertebrates evolutionarily developed a host defense to virus infection mediated by viral ZFP (Esposito et al, 2022). In addition to ZFP, the early events of DiPRO1 KD are associated with CGI demethylation-linked gene activation, which are involved in inflammation processes, mainly via TNF-α signaling. In relation to virus infection, TNF-α signaling is an important driver of host antiviral defense leading to cell death and tissue destruction (Benedict 2003; Kash et al, 2006; Oyler-Yaniv et al, 2021). Among them, the TRAF1 adapter protein and the TNFSF9 ligand of the TNF-R receptor TNFRSF9, potential direct targets of DiPRO1, were demethylated and overexpressed. Their interaction is important to transduce the non-apoptotic signals leading to activation of NF-kappaB. At the same time, TRAF1 is a specific target of activated caspases for the apoptosis induction in cancer cells playing a role as a link between caspases and the TNF receptors (76). Together with the upregulation of pro-inflammatory NFKB1 and NFKB2 genes and inflammatory caspases CASP4 and CASP5, DiPRO1 KD induces the expression of the initiator caspase CASP8, leading to the activation of CASP3, a final "executioner" of apoptosis (48). To summarize, early events of disruption of the DiPRO1-repressive complex in cancer cells lead to mimicking the defense against viral infection and tuning the cell decision from inflammation to death.

Remarkably, the observed demethylation of REs in myoblast cells exhibited a correlation with the activation of the interferon signaling pathway alongside the myogenesis pathways. As previously demonstrated, DiPRO1 has the capacity to regulate DUX4 and BSR satellite DNA, implicated as pathological loci in the

muscular dystrophy FSHD (Kim et al, 2015). Consistent with the role of REs in muscle biology, elevated levels of HERV transcripts were detected in biopsies obtained from affected muscles of patients with motor neuron disease (Oluwole et al, 2007). Moreover, ERV activation can play a role in myoblast fusion and increased muscle volume (Frese et al, 2015). The precise mechanisms through which REs influence muscle development are so far not well understood and further studies are needed, but their involvement highlights the complexity of the genomic landscape and the utilization of retroviral remnants in the regulation of essential biological processes like myogenesis. It is important to note that stringent conditions were selected for the analysis, with a focus on the conventional association of CGI methylation with gene repression. The consideration of methylation in relation to gene upregulation, an alternative scenario, was not undertaken in this study.

Previously, we reported that DiPRO1 binds the 4q35e enhancer of beta-satellite repeat region and its inhibition results in gene repression, suggesting the activator property of DiPRO1 (Kim et al, 2015). Here, we demonstrate that DiPRO1 presents binding sites in the SV40 enhancer and promoter and can inhibit gene expression. DiPRO1 binds also to DNA regions where it may contribute to CpG methylation, associated with heterochromatin. At the same time, it could be associated with the transcription at promoter and enhancer regions in active chromatin. We note that 42.6% of the genes associated with DiPRO1 binding sites in their vicinity are epigenetically activated and 22.5% are repressed by DiPRO1 KD, suggesting that DiPRO1 may exhibit a dual transcriptional role (Appendix Fig. S5B). This could probably related to a combinatorial recruitment of corepressors or coactivators, involved in the gene transcriptional control (Sobocinska et al, 2021). Similar to the vast majority of KZFP (Imbeault et al, 2017; Bertozzi et al, 2020), DiPRO1 interacts with the co-repressor TIF1B, also known as KAP1. Additionally, UHRF1 ubiquitin ligase, also known as NP95, was found in the DiPRO1 complex, similar to KZFP ZFP57 system (82). TIF1B and UHRF1 are classically considered to promote histone- and DNA-dependent epigenetic gene silencing (Groner et al, 2010; Niinuma et al, 2019), and this may contribute to the repressive property of DiPRO1. Therefore, it remains to be determined how DiPRO1 may activate gene expression, and the positive cofactors recruited to mediate it. It has been reported a bivalent nature of KAP1 in gene regulation (Singh et al, 2015). The mechanisms by which KAP1 can activate transcription are nevertheless not fully understood. A model involves a differential phosphorylation of KAP1 that leads to the release of corepressors and activation of MyoD/Mef2D promoters in myogenic cells (Singh et al, 2015). Whether this mechanism participates in the functions of DiPRO1 observed in RMS remains to be established.

The analysis of scRNA-seq of PDX tumors demonstrated that, in addition to cycling cell subpopulations, DiPRO1 expression exhibited enrichment in mesenchymal/muscle progenitors and in differentiated tumor cells. Previous studies have associated cycling cells with a worse patient prognosis (Repo et al, 2020), and newly discovered insights suggest that mesenchymal muscle-like stem cells play a role in driving relapse (Patel et al, 2022; Wei et al, 2022), opening avenues for therapeutic targeting. Differentiated MYOG-expressing cells exhibited lower expression levels of DiPRO1 compared to mesenchymal/muscle progenitors. The expression level of endogenous DiPRO1 in RMS cells was comparable to that in proliferating myoblasts. This implies that DiPRO1 could

potentially serve as a biomarker to predict the prognosis of RMS. Consequently, it was rational to examine DiPRO1 expression specifically in relapsed tumors that did not exhibit a complete response to therapy. As expected, high levels of DiPRO1 were detected in biopsies of relapsing tumors from patients with RMS and Ewing's sarcoma. RMS is the most common soft tissue sarcoma in children and adolescents, considered an aggressive and highly malignant cancer (Sun et al, 2015). Ewing sarcoma is a small round cell tumor, highly malignant and poorly differentiated that is currently the second most common malignant bone tumor in children (Gurria and Dasgupta 2018). It can also develop in the extraskeletal soft tissues, including muscles (Abboud et al, 2021). The shared molecular and functional changes of early events of DiPRO1 inhibition in these cancer cells led to cell cycle deregulation, differentiation, and caspase-mediated cell death. Despite heterogeneity, RMS and Ewing sarcoma are considered as being derived from primitive mesenchymal stem cells (Sun et al, 2015; Gurria and Dasgupta 2018) and have been currently classified as mesenchymal tumors in the 2016 World Health Organization classification (Louis et al, 2016), supporting a possibility of a common mechanism of DiPRO1 action in these cancer cells. Cell fate mapping of RMS tumors shows the presence of subpopulations of neural and muscle mesenchymal progenitors that can make both muscle and osteogenic cells, along with cells expressing muscle developmental features (Wei et al, 2022). RMS can be trans-reprogrammed to the neuronal phenotype resembling Ewing tumors (Hu-Lieskovan et al, 2005) by introducing the EWS-FLI1 fusion gene. Therefore, both tumors could be commonly targetable. Although the effect of DiPRO1 targeting in vivo has been demonstrated in a mouse tumor model of human Ewing sarcoma, we recognize the importance of investigating DiPRO1 targeting in RMS and other mesenchymal tumor types through future preclinical studies to broaden our findings.

Along with myogenic differentiation regulators, DiPRO1 can control neuronal development in RMS and Ewing sarcoma through SIX1 targets (Fig. EV3D,C). Among them, ephrin receptor EPHA4 (Gatto et al, 2014), actin binding LIM-proteins ABLIM1 and ABLIM2 (Barrientos et al, 2007), semaphorin SEMA7A (Korner et al, 2021) and glial neurotrophic factor GFRA2 (Wanigasekara et al, 2004) genes, which are involved in axon guidance, are commonly repressed in cancer cells by DiPRO1 and SIX1. This suggests that DiPRO1 may regulate neuronal plasticity in cancer through SIX1 targets. The DiPRO1-linked gene network expectedly distinguishes RMS and Ewing sarcoma, but also medulloblastomas and neuroblastomas, suggesting that targeting DiPRO1 might have clinical relevance for additional neuronal tumors.

Pediatric tumors overall are characterized by few specific genetic alterations (Grobner et al, 2018; Megiorni 2020), therefore epigenetic therapy would be particularly appreciable. One recent study reported a potential efficacy of epigenetic drug combination, inhibitors of histone deacetylases (HDAC) and lysine-specific demethylase 1 (LSD1), in Ewing sarcoma xenograft models (Garcia-Dominguez et al, 2018). Current US FDA-approved epigenetic drugs such as HDAC- and DNMT-inhibitors constitute a standard treatment in patients with leukemia, lymphoma, and multiple myeloma. Despite the ever-growing progress, epigenetic medicines still need to be improved due to a lack of locus specificity inducing off-target effects. The progress made in nanotechnology has allowed the development of nanomedicines based on inorganic nanoparticles for the transport and delivery of small RNA and DNA molecules, which will act with high precision and efficacy to silence target genes (Roberti et al, 2019). Thus, DiPRO1-inhibiting nucleic acid-based nanomedicines could be an attractive target for at least mesenchymal cancer treatment, contributing to the development of precision medicine. Moreover, taking into account that DiPRO1 depletion of normal cells does not induce cell death, the selective action represents a potentially less-toxic approach for treating patients. Indeed, the particular consequences of pediatric cancer treatment are related to the exposure to chemotherapy and radiation during the rapid physiologic changes that can result in specific tissue or organ damage, or alteration of growth and development. Two-thirds of survivors suffer from late effects, which could be severe or life-threatening (2003). Therefore, there is a high medical need for new less-toxic therapeutic approaches, particularly for pediatric cancer treatment.

## Methods

### Antibodies

Polyclonal rabbit anti-MCM6 (Proteintech, 13347-2-AP, 1:8000) and polyclonal rabbit anti-TIF1B/KAP1 (Merck Millipore, Sigma-Aldrich, ABE1859, 1:2000), mouse monoclonal anti-ICBP90/UHRF1 (Sigma-Aldrich, MABE308, 1:5000), rat monoclonal anti-HA-Peroxidase high affinity (Roche, Sigma-Aldrich, 12013819001, 1/1000), rabbit polyclonal anti-Actin (Sigma-Aldrich, A2103, 1/3000), mouse monoclonal anti-FLAG® M2-Peroxidase (Sigma-Aldrich, A8592, 1/5000), mouse monoclonal anti-Tropomyosin Ab (Sigma-Aldrich, T2780, 1/50), goat polyclonal anti-rabbit IgG–Peroxidase antibody (Sigma-Aldrich, A6154, 1/1000), Alexa Fluor® 488 Goat Anti-Mouse IgG antibody (Molecular Probes, Invitrogen, A-11001, A11029, 1/100), Alexa Fluor® 594 Goat Anti-Rabbit IgG antibody (Molecular Probes, Invitrogen, A-11012, 1/100), Fluoroshield™ with DAPI (Sigma-Aldrich, F6057, 1/3), Dynabeads™ CD25 (Invitrogen, Thermo Fisher Scientific, 11157D), rat monoclonal anti-CD34 antibody (HycultBiotech, HM1015, 1/20), Polink-2 Plus HRP Rat-NM Bulk kit for DAB (GBI Labs, D46-110), rabbit monoclonal anti-Ki-67 (Neomarkers; LabVision, Thermo Fisher Scientific, RM-9106, 1/200), rabbit polyclonal cleaved Caspase-3 (Asp175) (Cell Signaling, 9661, 1/400), anti-FLAG M2 affinity gel (Sigma-Aldrich, A 2220), FLAG® Peptide (Sigma-Aldrich, F3290), Protein A agarose (Pierce, Thermo Fisher Scientific, 20333).

### Cell Culture

Immortalized human myoblasts were generated at the Institute of Myology (Mamchaoui et al, 2011) and provided by MyoBank-AFM (Paris, France). RMS JR cells (COG, RRID: CVCL_RT33) and Ewing sarcoma A673 (RRID: CVCL_0080), EW7 (RRID: CVCL_1217) cells provided by Dr C. Lanvers (University of Muenster, Germany), TE671 (RRID: CVCL_1756) provided by Dr. K. Mamchaoui (Institute of Myology, Paris, France) were STR genotyped (Mycrosynth, Switzerland) and tested for mycoplasma contamination. The immortalized myoblasts (Myo) were cultured in 199 medium and Dulbecco's modified Eagle's medium (DMEM) (Invitrogen) in a 1:4 ratio, supplemented with 20% Fetal Calf serum (FCS) (Invitrogen), 2.5 ng/ml hepatocyte growth factor (Invitrogen), 0.1 µmol/l dexamethasone (Sigma-Aldrich) and 50 µg/ml gentamycin (Invitrogen) at

37 °C with 5% $CO_2$. They were differentiated into myotubes in DMEM medium with 2% FCS and 1% penicillin/streptomycin, at 37 °C with 5% $CO_2$ for 5–8 days. The formation of multinucleated myotubes was controlled by hematoxylin staining (Sigma-Aldrich). Cancer cells were maintained in DMEM medium (Gibco, Life Technologies)/10% FBS (PAA Laboratories GmbH) with 100 UI/ml penicillin/streptomycin (Invitrogen). Primary mouse *Pax7-nGFP::Six1*−/−*Six4*−/−*Six5*−/− E18.5 fetal myogenic stem cells were isolated and cultured as already described (Wurmser et al, 2020). Cell size was measured using ImageJ software.

## Patient consent and sample collection

The cohort of this study included patients <18 years old at the time of initial diagnosis who underwent tumor biopsy/surgery of their recurrent/refractory malignancy for WES and RNA sequencing within the MAPPYACTS (clinicaltrial.gov NCT02613962) (Berlanga et al, 2022). All patients and parents/legal representatives had signed informed consent for further research studies (Appendix Tables S1, 2). The experiments conformed to the principles set out in the WMA Declaration of Helsinki and the Department of Health and Human Services Belmont Report.

## Plasmids and constructs

### Vectors expressing the DiPRO1 ORF
The sequence of the full-length open reading frame (ORF) of human DiPRO1wt was optimized according to the degeneracy of the genetic code so as not to be recognized by shDiPRO1. The created cDNA (Genewiz) contained sites for the restriction enzymes *Xho*I and *Not*I for subsequent cloning into the appropriately digested retroviral vector pOZ_FHHY (Nakatani and Ogryzko 2003) used as a control. The resulting recombinant protein (FHHY_DiPRO1wt, pDiPRO1) contained several epitope tags and a fusion protein at the amino-terminal position: Flag polypeptide, polyhistidine motif (6xHis), hemagglutinin (HA), and the yellow fluorescent protein (YFP) (Appendix Fig. S2A). Cell lines were infected with virus expressing pDIPRO1, and IL2Rα+/YFP+ cells were sorted using magnetic beads conjugated with antibodies against IL2Rα (Dynabeads CD25, Invitrogen) and then on a FACS AriaIII cell sorter (BD).

### Knockdown vectors targeting DiPRO1
A lentivirus-based short hairpin (sh)RNA strategy was used to repress DiPRO1 expression. The shRNAs expressed by the pKLO.1-puro vector were purchased from Sigma-Aldrich. The sequences for human shDiPRO1 were as follows:

 shDiPRO1-1/sh1/shDiPRO1: CCATCATCTTTACCAATACAT
 shDiPRO1-2/sh2: GCCACGTTCACCAGAAATGTT
 shDiPRO1-3/sh3: CCTGAAGACAAATCCTATGAA

Myoblast cell lines were previously immortalized using a vector containing a puromycin resistance gene (Mamchaoui et al, 2011). Therefore, a second batch of shRNA constructs was created in which puromycin was replaced with blasticidin (pKLO.1-bsd). For this purpose, the cDNA of the blasticidin gene was amplified using the primers: 5′CACACAGGATCCACCGGAGCTTACCATGGC-CAAGCCTTTGTCTC3′ and 5′CACACAGGTACCGATGCATG GGGTCGTGCGCTCCTTTCGGTCGGGCGCTGCGGGTCGTGG GGCGGG CGTTAGCCCTCCCACACATAACCAGA3′, containing the enzymatic restriction sites *Bam*HI and *Kpn*I and then

inserted into the puromycin-excised-pKLO.1 vectors. Myoblasts and cancer cell lines were maintained under selection with a predetermined concentration of blasticidin S (60 µg/ml, Sigma-Aldrich). TE671 cells were also selected with puromycin (10 µg/ml). A lentivirus plasmid vector containing an shRNA insert not targeting human and mouse genes (SHC002, Sigma-Aldrich) was used as a non-target control (shCTL).

All new clones were verified by restriction enzyme digestion (*Xho*I and *Not*I) and sequencing to confirm identity (GATC Biotech).

### Luciferase reporter vectors
The pGL3 vector-based constructs were used in these experiments. pGL3-Control Vector (Promega, E1741) vector was used to evaluate promoter and enhancer activity. It contains the SV40 promoter and SV40 enhancer, unlike the pGL3-Basic vector (Promega, E1751) without a promoter or enhancer, which was used for normalization. pTATA-Luc, pMEF3-Luc, and pSix1VP16 plasmids have been described (Grifone et al, 2004).

## Luciferase assay

The transfection procedure and preparation of the complexes were performed using Lipofectamine 2000 reagent and 1 µg of DNA according to the manufacturer's recommended protocol (Invitrogen). The transfection experiments were performed with pGL3 luciferase reporter plasmids separately and in co-transfection with the pDiPRO1 vector using a 1:2 ratio of DNA (µg) to Lipofectamine™ 2000 transfection reagent (µl) (Invitrogen, Life Technologies). Myoblasts and RMS cells were seeded on 24-well culture vessels at $3.8 \times 10^4$ cells per well the day before transfection. After a 48-h incubation period, gene activity was assessed according to the supplied Luciferase Assay System Kit protocol (Promega). Luminescence detection was performed using the Microlumat LB96p device (EGG Berthold, Bad Wildbad, Germany). Protein concentration in the extracts was measured using the QuantiPro BSA Assay Kit (Sigma-Aldrich). Total protein extracts were prepared from transfected Six-mutant fetal myogenic stem cells for the measurement of Firefly and Renilla luciferase activities according to Promega's Luciferase assay protocol (Dual-Luciferase reporter assay system, Promega). The luciferase assay was analysed using R-script https://github.com/ScienceParkStudyGroup/dual-luciferase#relative-expressions. Welch Two-Sample t-test was applied for statistical difference.

## Transient transfection

Cultured human TE671 cells were transfected with 1 µg of DNA (pDiPRO1 or shDiPRO1) using Lipofectamine™ 2000 transfection reagent according to the manufacturer's recommended protocol (Invitrogen, Life Technologies). Briefly, 24 h before transfection, cells were trypsinized, harvested, diluted with fresh medium, and seeded into six-well plates at $4 \times 10^5$ cells per well. Each sample was assayed in triplicate 48 h post-transfection.

## Viral transduction

Stable overexpression or downregulation of DiPRO1 transcripts was achieved by creating stable cell lines using lentiviral or

retroviral vectors. Viral particles were obtained as described previously (Nakatani and Ogryzko 2003) with some modifications. To obtain a transduction-competent virus, the viral vector was transfected together with pCMV and pMD2G expression plasmids for gag-pol and env. The HEK293T (RRID: CVCL_0063) cells were transfected using the calcium phosphate transfection method (CAPHOS, Sigma-Aldrich). Selection was started 4–5 days post-infection using either magnetic sorting (pOZ), antibiotic selection (pOZ, pLKO.1-puro/blast), or fluorescence sorting (pOZ). For antibiotic selection, the lethal concentration of the antibiotic was first tested on parental nontransduced cells and used as a reference to select stable cell lines (10 μg/mL puromycin or 60 μg/mL blasticidin).

## Assessment of cell differentiation and cell dimensions

Differentiation was assessed morphologically based on May-Grünwald–Giemsa staining. To this aim, cells were cytospun onto slides, fixed with absolute methanol for 15 min at room temperature, and then transferred to a staining tank containing May-Grünwald dye diluted in an equal amount of water for 5 min and then to the Giemsa solution (Unité de cytopathologie, Gustave-Roussy). The extent of myogenic differentiation was determined by the number of multinucleated myotubes formed and the expression level of myogenic regulatory factors (tropomyosin). Cells containing more than three nuclei were considered as multinucleated myotubes. and then at least 20 cells for each experimental condition were analyzed on an Olympus IX83 inverted microscope equipped with an ORCA-Flash 4.0 digital CMOS camera (HAMAMATSU). Alternatively, the glass slides were scanned and real images were taken using an EVOS M7000 Imaging System (Thermo Fisher Scientific). The images were visualized using Olympus OliVIA software. Cell size and nuclear-cytoplasmic ratio (N/C) (Su Lim et al, 2015) was scaled and evaluated using ImageJ.

## Cell viability and proliferation assays

Cells were subcultured in multi-well coated cell culture plates and allowed to reach ~95% confluence within 72 h. Cell viability was assessed using Cell Counting Kit 8 (Sigma-Aldrich), according to the supplier's recommendations. Briefly, WST-8, a tetrazolium salt, is reduced by cellular $NAD+$- and $NADP+$-dependent dehydrogenases to an orange formazan product soluble in tissue culture medium. The amount of formazan formed is directly proportional to the number of living cells. The number of cells was determined using cells at different dilutions: from 0 to 25,000 cells/well/100 μL. Absorbance was measured at 460 nm using a plate reader 1 h after seeding. The resulting titration curve was used to estimate the number of viable cells for proliferation assay. Cells were then seeded at $1.0 \times 10^4$, $1.5 \times 10^4$, and $2.5 \times 10^4$ cells/well and absorbance was measured after 72 h of proliferation. Data were expressed as the average percentage of three dilutions of cells after 72 h relative to the initial number of cells (set at 100%).

## Active caspase-3 activity assay

RMS cells were plated at $2 \times 10^5$ cells/well and transfected with the pLKO vectors expressing shCTL or shDiPRO1. Forty-eight hours post-transfection, the cell lysates were assayed using the Caspase-3 Colorimetric Activity Assay Kit, DEVD (APT165, Sigma-Aldrich, Merck), according to the manufacturer's instructions. The protein quantity was measured at 562 nm absorbance for normalization using the BSA Protein Assay Kit (Pierce). The chromophore p-nitroaniline (pNA), after cleavage from the labeled substrate DEVD-pNA, was measured at 405 nm absorbance using a microplate spectrophotometer (EPOCH, BioTek). Significant differences between conditions were analyzed using the Welch two-sample t-test.

## Immunofluorescence microscopy

Cells were grown on two-well chamber slides (Nunc, Lab-Tek) and fixed in 4% formaldehyde, permeabilized with 0.2% Triton, and blocked in PBS containing 3% bovine serum albumin (Sigma) for 1 h at room temperature. The cells were then incubated with anti-tropomyosin monoclonal Ab (1/50, Sigma-Aldrich) and anti-actin Ab (1/100, Sigma-Aldrich) for 1 h. Signals were detected with the appropriate Alexa 488 or Alexa 594-conjugated secondary antibodies (1/100, Molecular Probes Invitrogen) for 1 h at room temperature. For nuclear counterstaining, cells were mounted with Fluoroshield containing DAPI (1/3, Sigma-Aldrich). Images were obtained using an Olympus IX83 inverted microscope equipped with an ORCA-Flash 4.0 Digital CMOS camera (HAMAMATSU).

## Electrophoresis mobility shift assay (EMSA)

Nuclear extracts from pDiPRO1-RMS were prepared as in (Klochkov et al, 2006). Cells were lysed by incubation for 2 min on ice in cell lysis buffer (10 mM NaCl, 20 mM HEPES [pH 7.6], 1.5 mM $MgCl_2$, 1 mM $ZnSO_4$, 20% glycerol, 0.1% Triton X-100, 1 mM dithiothreitol, and 1 mM phenylmethylsulfonyl fluoride) supplemented with Protease Inhibitor Cocktail (Merck). After centrifugation, the pelleted nuclei were collected and shaken for 30 to 60 min on ice in nuclear extraction buffer (0.5 M NaCl, 20 mM HEPES [pH 7.6], 1.5 mM $MgCl_2$, 1 mM $ZnSO_4$, 20% glycerol, 0.1% Triton X-100, 1 mM dithiothreitol, 1 mM phenylmethylsulfonyl fluoride) supplemented with Protease Inhibitor Cocktail (Merck). After centrifugation ($14,000 \times g$; 5 min), nuclear extracts were stored at −80 °C. EMSA and Six1 transcription-coupled translation were performed as described (Salminen et al, 1995). Binding assays were carried out on ice for 30 min in a buffer composed of 20 mM HEPES (pH 7.6), 40 mM NaCl, 5 mM $MgCl_2$, 0.5 mM $ZnSO_4$, 0.1 mM EDTA, and 4% Ficoll, with 100 to 200 fmol Klenow-end-labeled DNA, 1 μg poly(dI-dC) used as a nonspecific competitor, and 5 to 6 μg nuclear protein extract in a final volume of 20 μl. When necessary, specific competitor DNA was added simultaneously with the labeled DNA at a 300-fold molar excess. The sequences of oligonucleotides used in these experiments were as follows:

Aldolase A MEF3 site: 5′GAATGTCAGGGGCTTCAGGTTTCCCTAAATA

Myogenin MEF3 site: 5′GCTTAGAGGGGGGCTCAGGTTTCTGTGGCG

DiPRO1 motif 1: 5′AGGGGGCGGGTTAGGTTCCTGTGGCG

TNFSF9 site: 5′GAGAAAGGCTGGCTTCCCTCCATG

TRAF1 site: 5′GCGCCTGGCCTAGGGTCCCTTTT

MAF site: 5′AGTTCTGCTGACTCAGCAGTTTGGA and 4qAe enhancer (Kim et al, 2015).

## Gene expression analysis

### RNA isolation and RT-qPCR

Total RNA was extracted using the RNeasy Mini Kit (Qiagen) for cell culture samples. cDNA was synthesized from 500 ng of total RNA using the RevertAid H Minus First Strand cDNA Synthesis Kit (Fermentas, Thermo Fisher Scientific) for RT-PCR analysis according to the manufacturer's instructions. n the StepOnePlus Real-Time PCR Detection System (Applied Biosystems) using FastStart Universal SYBR Green Master Mix (Rox) (Roche Life Science). Relative expression was determined by the $2^{-\Delta\Delta Ct}$ method (Livak and Schmittgen 2001) using GAPDH as an internal control. The oligonucleotide sequences used in this study are listed in Appendix Table S3.

### Microarray assay

For microarray analysis, RNA quality was assessed using a Bioanalyzer (Agilent) and then quantified using a Biospecnano instrument (Shimadzu, Kyoto, Japan). Gene expression analysis was performed using an Agilent® SurePrint G3 Human GE 8x60K Microarray (Agilent Technologies, Santa Clara, CA, USA) using an Agilent Single Color Labeling Kit (Low Input Quick Amp Labeling Kit 034949) adapted for small amounts of total RNA (100 ng total RNA per reaction). Hybridization was performed according to the manufacturer's instructions. Microarray images were analyzed using Feature Extraction software version (10.7.3.1) from Agilent Technologies. Default settings were used. Raw data were processed using R with LIMMA, the R package from the Bioconductor project, and processed as follows: the gMedianSignal and rMedianSignal data were imported, the control probes were systematically removed, and flagged probes (gIsSaturated, gIsFeatpopnOL, gIsFeatNonUnifOL, rIsSaturated, rIsFeatpopnOL, and rIsFeatNonUnifOL) were set to NA. To obtain a single value for each transcript, the average value of the total data of each replicated probe was taken. The missing values were calculated using the KNN algorithm from the 'impute' package from the R Bioconductor. The normalized data were then analyzed. To estimate the differentially expressed genes between the two groups, we began by fitting a linear model to the data. We then used an empirical Bayesian method to moderate the standard errors of the estimated logarithmic changes. The top-ranked genes were selected according to the following criteria: absolute fold change (FC) ≥1.4 and ≤-1.4 and $p$ value <0.05.

### RNA-seq

RNA-seq libraries from the MAPPYACTS study sequencing data were aligned with the GRCh38 version of the human transcriptome from the GENCODE project using Salmon v0.9 (Patro et al, 2017) and converted from transcripts to gene expression using the Tximport R package. RNA-seq analysis data of 1375 cancer cell lines were extracted from the DepMap Broad Institute portal. The data were then computed using DEBrowser (Kucukural et al, 2019) and the DESeq2 R package (TMM normalization, parametric fit, LRT test). Coexpression analysis was performed using the network_plot R function.

## Flow cytometry analysis

Cells were washed with cold PBS and fixed in cold 70% ethanol for at least 30 min. The supernatants were washed twice in PBS. The cells were then treated with RNase (Macherey-Nagel, Thermo Fisher Scientific), and stained with propidium iodide (PI) (Sigma-Aldrich), DNA content and percentage of viable and dead cells in each phase of the cell cycle were assessed using a C6 Accuri flow cytometer (BD Biosciences) and FlowJo-V10 software (FlowJo, LLC).

## Immunoaffinity purification

DiPRO1 ORF was cloned into the pFHHY-expression vector, and a stable cell line overexpressing the DiPRO1wt protein (pDiPRO1) was generated. Immunoaffinity purification was based on the protocol proposed by (Nakatani and Ogryzko 2003) (Pankow et al, 2016) and optimized by our team. Approximately $10 \times 10^6$ RMS cells were lysed in CSK buffer (1:5 package V) containing 10 mM Tris-HCl, pH 7.5 100 mM NaCl, 300 mM sucrose, 3 mM MgCl₂, 1 mM EGTA, 1 mM NaF, 1 mM Na₃VO₄, 0.5% Triton X-100 and 2 mM PMSF, Complete™ EDTA-free Protease Inhibitor Cocktail (Sigma-Aldrich), and 50 μL were taken for the input lanes. After centrifugation for 10 min at 4000 rpm, the supernatant (cytosolic fraction) and pellet (nuclear fraction) were collected. Nuclei were then resuspended in nuclear incubation buffer (150 mM HEPES, pH 7.9; 1.5 mM MgCl₂; 150 mM KOAc; 10% glycerol; and protease inhibitors) and digested with 0.15 unit/μl benzonase (Sigma-Aldrich) according to Aigun et al (Aygun et al, 2008). The samples were cleared by centrifugation at $20,000 \times g$ for 20 min, and proteins were collected from the nuclear extract. An identical procedure was used to purify isogenic cells lacking bait protein expression (null control).

Nuclear extracts were pre-cleared with an inert resin Protein A Agarose (Thermo Fisher Pierce) for 2 h at 4 °C with rotation, then separately applied to ANTI-FLAG M2 Affinity beads (Sigma-Aldrich) (≈25 μl of packed beads) and incubated for 2 h at 4 °C. After binding the protein complexes, the beads were washed extensively with wash buffer [1 mL 2 M Tris-HCl pH 7.4, 25 mL Glycerol 99.5%, 150 μL 1 M MgCl₂, 40 μL 0.5 M EDTA pH 8, 9.99 mL 3 M KCl, 200 μL 100 mM PMSF, 70 μL 14.3 M bME, and protease inhibitors]. Finally, the purified protein complexes were eluted with FLAG elution buffer [1 mL 2 M Tris-HCl pH 7.4, 25 mL Glycerol 99.5%, 150 μL 1 M MgCl₂, 40 μL 0.5 M EDTA pH 8, 9.99 mL 3 M NaCl, 400 μg/ml 3× FLAG peptide, 200 μL 100 mM PMSF, and protease inhibitors] or/and with LDS loading buffer. Eluates were resolved in 4–12% bis-Tris gradient PAGE (NuPAGE, Life Technologies) and Western blotting was performed using antibodies raised to HA-epitope (Sigma Alrich). When the protein bands were excised for mass spectrometry, the gels were stained with InstantBlue (Expedeon).

## Sample preparation and proteomic analysis by MS/MS

### Sample preparation

The Co-IP-derived gels were combined and underwent in-gel digestion as described previously (Moio et al, 2011). Briefly, the gel slices were washed with 100 μl of 1:1 acetonitrile (ACN):50 mM Ambic followed twice by 100 μl of 100% ACN, then reconstituted with 100 μl 10 mM DTT and alkylated with 100 μl 55 mM Iodoacetamide. Final dehydration was performed using 100 μl of 100% acetonitrile. Each wash was carried out for 20 min at 37 °C with shaking at 1400 rpm. The gel slices were dried in a speed-vac for 5 min. Gel slices were pre-incubated with 20 μl of pre-activated trypsin (Promega France, Charbonnières) at room temperature for

15–20 min, 50 mM ammonium bicarbonate was added to coat the gel, and incubated at 37 °C for 16 h. Peptide-containing supernatants were collected, and the gel slices were washed twice with 50 µl of 0.1% formic acid in 70% ACN for 15 min at 37 °C. The collected supernatants were dried on a high-speed fan until completely dry, then resuspended in 10 µl of water with 0.1% formic acid (v/v) for mass spectrometric analysis.

### MS/MS analysis

Protein identification were performed using an Orbitrap Q-Exactive mass spectrometer (Proteomic Platform, Gustave Roussy) supplied with an Easy-nLC-100 chromatography system (Thermo Scientific) coupled to an Orbitrap Q-Exactive analyser (Thermo Scientific) and an Easy-spray column source for samples ionization (Thermo Scientific). The gradient used for protein identification and/or MS2 targeting was as follows: 0 to 120 min: 5% to 35% solution B; 120 to 140 min: 35 to 50% solution B; 140 to 145 min: 50 to 90% solution B; 145 to 150 min: 90% solution B, where solution A: 100% water with 0.1% formic acid and solution B: 100% acetonitrile and 0.1% formic acid. The MaxQuant package was used to identify proteins using the human genome FASTA database. The software was adjusted to a confidence level of 10 ppm for MS and 0.02 Da for MS/MS. The $p$ value used for peptide confidence identification was $p \leq 0.01$. The obtained proteins were then analyzed using Perseus tools (Tyanova et al, 2016).

### Mass spectrometry data processing

Mass spectrometry data processing. The data obtained from the mass spectrometry analysis were then filtered according to the following parameters: (a) at least two unique peptides must match the sequence of the candidate protein and (b) the proteins must show a significant ($p \leq 0.05$) increase in LFQ intensity in the DiPRO1 fraction compared with the control fraction. Common false positives, such as keratins, keratin-associated proteins, and serum albumin were excluded from the analysis. In addition, the CRAPome database [http://www.crapome.org, (Mellacheruvu et al, 2013)] was employed to exclude the most likely false positive hits.

### Validation of DiPRO1-protein binding

The DiPRO1-interactions were confirmed with co-immunoprecipitation (Co-IP) using specific antibodies and Protein A Agarose (Thermo Fisher Pierce). Nuclear extracts were isolated as described in the Immunoaffinity purification section. Specific antibodies (5 µg) were separately incubated overnight at 4 °C with 500 µg of protein with constant rotation. Protein A agarose beads were added (100 µl of 50% slurry in lysis buffer) and incubation was performed for 2 h at 4 °C with rotation. The mixture was washed three times and centrifuged for 3 min at $500 \times g$ at 4 °C. Precipitated proteins were eluted by boiling in NuPAGE LDS sample buffer and analyzed by Western blotting using appropriate antibodies. Interaction with DiPRO1 was included as a positive control. To this end, DiPRO1 was co-immunoprecipitated from cell lines expressing DiPRO1 using the anti-FLAG antibody. Protein complexes were analyzed by Western blotting using HRP-conjugated anti-HA antibodies.

## Protein isolation and immunoblotting

Sample pellets were diluted in 150–300 µl of 1× phosphate-buffered saline with a protease inhibitor cocktail (Sigma-Aldrich) and

sonicated. NuPAGE® LDS sample buffer (Life Technologies) and 5 mM DTT were added and the samples were incubated for 30 min at 50 °C. Western blotting was performed according to a standard procedure using pre-cast gel cassettes (Life Technologies) and NuPAGE® MOPS SDS run buffer (Thermo Fisher Scientific). The Pierce G2 Fast Blotter device (Thermo Fisher Scientific) was exploited for the semi-dry transfer of protein gels. SeeBlue Plus2 protein standard (Life Technologies) was used to detect the size of the protein being analyzed. The detection solution was HRP Immobilon Western Substrate (Merck Millipore). The analysis was performed using ANTI-FLAG® M2-Peroxidase (Sigma-Aldrich, 1/5000), anti-HA-peroxidase (Roche, Sigma-Aldrich, 1/1000), anti-Actin (Sigma-Aldrich, 1/3000) and anti-rabbit secondary antibodies (Sigma-Aldrich) in 1/1000 dilution. Fold change was calculated after quantification using ImageJ software.

## ChIP-seq

To associate the binding site information with gene expression, we performed the analyses as outlined below. First, the ChIP-seq processed datasets of DiPRO1/ZNF555 from reference (Imbeault et al, 2017, Data ref: Imbeault et al, 2017) (NCBI's GEO accession number GSM2466593) were genome-wide annotated using the HOMER package (Heinz et al, 2010) and the PAVIS tool (Huang et al, 2013). The target region was located 10 Kb upstream of the TSS and 5 Kb downstream of the TES. The annotation of repeats was realized using the EaSeq "Colocalization" tool and the UCSC "rmsk" track. In the ChIP experiment, 293T cells were induced to express a cDNA encoding human DiPRO1 by lentiviral transduction. For expression-linkage correlation analysis, our series of gene expression experiments ($p < 0.05$) reported here were used in combination with ChIP-seq to define a set of genes that are enriched in direct transcriptional targets. Motif enrichment and similarity analysis were performed using Homer's tool and the MEME suite tool (Bailey et al, 2009): MAST, FIMO (Grant et al, 2011), Tomtom (Gupta et al, 2007). The ChIP-seq data were extracted from GSM2466593 for DiPRO1, jaspar.genereg.net for SIX1 MA1118.1 (Spitz et al, 1998; Cheneby et al, 2018, Data ref: Cheneby et al, 2018) and GSE105311 for ZSCAN4 (Consortium 2012, Data ref: Consortium 2012). Cis-regulatory regions were determined using GREAT software (default settings for cis-regions: 5.0 kb constitutive upstream and 1.0 kb downstream of TSS regions and up to 1 Mb of extension) (McLean et al, 2010). The ChIP-seq data were remapped to human RefSeq hg38 using the NCBI genome "Remap" tool. Transcription factor binding regions and chromatin state were annotated using the ENCODE TxnFactorChIPE3 (Wang et al, 2013, Data ref: Wang et al, 2013) and BroadChromHMM tracks (Ernst and Kellis 2010, Ernst et al, 2011, Data ref: Ernst et al, 2011). The repeats were annotated using the RepeatMasker track (Smit et al, 1996-2010, Data ref: Smit et al, 1996-2010) from UCSC Genome Browser.

## MIRA-seq

### DNA extraction and library preparation

Cells were washed in PBS and mixed with 500 µl of ice-cold MES lysis buffer (Keeney and Kleckner 1996), incubated for 5 min, and centrifuged at 10,000 rpm for 5 min. The supernatant was collected and then diluted in 300 µl of LDS extraction buffer

(10% glycerol, 0.14 M Tris-Base, 0.14 M Tris-HCl, 2% LDS, 500 nM EDTA. The samples were briefly sonicated to reduce viscosity and then centrifuged. The resulting supernatant was mixed with an equal volume of phenol:chloroform:isoamyl alcohol (25:24:1 [vol/vol]). The aqueous phase was retained by centrifugation, and then the dissolved DNA was precipitated with 0.1 volume of 2.5 M NaCl and a 2.5-fold volume of ice-cold ethanol. The samples were then centrifuged at $15,000 \times g$ for 15 min at 4 °C. Finally, the DNA pellet was washed with precooled 70% ethanol and resuspended in 50 μL of Milli-Q water. The concentration of extracted DNA was measured using the Qubit instrument and the Qubit dsDNA HS Assay Kit (Thermo Fisher Scientific). After extraction, genomic DNA was sonicated (300–700 bp size) and MIRA pull-down was performed using the Methyl Collector Ultra Kit (Active Motif), which uses an MBD2b/MBD3L1 protein complex, according to the manufacturer's instructions starting with 1 μg of DNA and using high salt binding conditions. Extracted genomic DNA (gDNA) was sonicated in a Covaris S220 (LGC Genomics GmbH) to an average fragment size of 180 bp. Fragment length distribution was assessed by microelectrophoresis using the Bioanalyzer 2100 (Agilent). Enriched DNA and input DNA fragments were end-repaired, extended with an "A" base at the 3′ end, ligated with pair-indexed adapters (NEXTflex, Bioo Scientific) using the Bravo platform (Agilent), and PCR amplified for ten cycles. Final libraries were purified with AMPure XP beads (Beckman Coulter), pooled in equal concentrations, and subjected to pair-end sequencing (100 cycles: $2 \times 50$) on the Novaseq-6000 sequencer (Illumina) at the Genomic platform (Gustave Roussy, Villejuif, France).

### Bioinformatics analysis

Using the Galaxy platform (Afgan et al, 2016), reads were groomed and cleaned by clipping (ClipAdapter) and trimming (TrimGalor) and the quality of NGC data was evaluated using the FastQC tool. GC content per sequence was distributed normally for input samples and showed a ~6–10% higher shift for MIRA-seq reads. The Bowtie2 algorithm was used to map the pared end reads against the hg38 genome index version 2013. The resulting BAM formatted files were filtered using bitwise flag-based alignment records to obtain the output of uniquely mapped reads corresponding to unique chromosome regions, which were used for downstream analysis. Bedtools were used to convert BAM to BED. The following steps of the analysis were performed using the free interactive software EaSeq (Lerdrup et al, 2016). Easeq browser was used for tracking and visualization of genomic regions. Peak searching was performed with the default procedure, adaptive local thresholding (ALT). The settings were window size 200 bp, $p$ value $\leq 10^{-5}$, FDR $10^{-5}$, and Log2FC 2. Peaks were merged within a 500 bp size. CpG sites were downloaded from http://methylqa.sourceforge.net/. The human genome (version hg38) and repeat-masked CGI (hk-CGI) and unmasked CGI tracks were downloaded from the UCSC genome database (Gardiner-Garden and Frommer 1987). The rep-CGI regions correspond to the unmasked CGI excluding hk-CGI. CGI meets the following criteria: GC content of 50% or more, length greater than 200 bp, ratio greater than 0.6 between the observed number of CG dinucleotides and the expected number based on the number of Gs and Cs in the segment. The genomic distribution of CGIs was analyzed by the annotation tool with the hg38 genome, and regions from 501 to 10,000 bp upstream of transcription start sites (TSSs) were considered as promoter regions, 500 bp upstream and downstream of TSSs were considered as TSS regions. CGI repeats were annotated using the "Colocalization" tool with RepeatMasker track from the UCSC Genome Browser.

## In vivo study design

All procedures on animals were approved by the CEEA26 Ethics Committee, and the Ministry of Agriculture (approval number: APAFIS#8550-201701160922201) and were performed within the guidelines of humane care of laboratory animals established by the European Community (Directive 2010/63/UE). Upon arrival, animals were housed in groups of five and allowed 7 days to acclimatize under conventional housing conditions (water and food ad libitum, 12-h light-dark cycle, controlled temperature).

The antitumor activity of DiPRO1 inhibition was evaluated using PEI- nanocomposites containing siRNA oligos or a vector expressing a DiPRO1-targeting shRNA compared with a non-targeting siRNA. Nucleic acids (10 μg) were complexed with a linear polyethylenimine transfection agent (in vivo-jetPEI ®; Polyplus Transfection) at an N/P ratio of 8 (N/P ratio is the number of nitrogen residues in jetPEI ® per nucleic acid phosphate) and resuspended in 50 μL of 5% (wt/vol) glucose. The shDiPRO1 conformed to shDiPRO1-1 (see section Plasmids and Constructs). The modified siRNA-based sequences were as follows:

siDiPRO1:
Sense: 5′-GCCAUCAUCUUUACCAAUAdTdTdTdTdT-3′
Antisense: 5′-UAUUGGUAAAGAUGAUGGCdAdAdAdAdA-3′
Scramble control siCtl (Kedinger et al, 2013)
Sense: 5′-AUGUCUACUGGCAGUCCUGdTdTdTdTdT-3′
Antisense: 5′-CAGGACUGCCAGUAGACAUdAdAdAdAdA-3′
siDiPRO1_Cy5
Sense 5′-[CY5]GCCAUCAUCUUUACCAAUAdTdTdTdTdT-3′
Antisense: 5′-UAUUGGUAAAGAUGAUGGCdAdAdAdAdA-3′
d is deoxyribonucleotide

For subcutaneous (s.c.) xenograft models, $6 \times 10^6$ Ewing sarcoma (A673) cells were injected into the right flank of 6-week-old females (Hsd:Athymic Nude-Foxn1$^{nu}$, Envigo, 6903F). Animals with established s.c. tumors were randomized to treatment groups based on tumor size ($n = 7$ or 5), receiving either nanocomposites of (i) siDiPRO1/jetPEI®, (ii) shDiPRO1/jetPEI®, or (iii) scrambled siRNA controls (Ctl), with the exclusion criterion of tumor volume (V < 40 mm³ or V > 250 mm³). In accordance with the "3R's" of animal research, we did not use the second scramble control (shRNA), considering the lack of effect in vitro as identical to scrambled siRNA. The nanocomposites were administered intra-tumorally in glucose solution at 0.5 or 1.0 mg/kg/injection twice or three times per week. The mice were monitored regularly for changes in tumor size and weight. Animals were sacrificed according to the animal ethic endpoints or at the end of the experiments. Satellite mice were treated equally to the treatment groups and sacrificed for histochemical and tumor cell uptake analysis using fluorescent nanocomposites coupled to Cy5. Fluorescence emission was monitored at 24 and 72 h using the 3D scanner of the IVIS® Spectrum in vivo imaging system (PerkinElmer) and analyzed by Living Image Version 4.5.2 software

after a single administration at 0.5 mg/kg/injection. The length and width of the tumor were measured with calipers, and the Antitumor activity was assessed on the basis of tumor growth delay (TGD) in median time for tumor volumes to be doubled (Demidenko 2010), and tumor growth inhibition at day 10 (TGId10, %) according to (Tsukihara et al, 2015). Toxicity was defined as body weight loss >20%, or mortality. Data analysis was performed using GraphPad Prism and TumGrowth (Enot et al, 2018) software. Statistical tests and the number of biological replicates (n) for each experiment are indicated in the figure legends. No blinding was done.

## In vivo tumor cell internalization of amino acids/PEI particles

In vivo tumor cell uptake was measured in mice with siDiPRO1 coupled to the fluorescent red dye $Cy^5$ (Sigma-Aldrich) delivered by intratumoral injection at 0.5 mg/kg. Fluorescent images were followed 24 and 72 h post-injection with the 3D scanner of the in vivo imaging system (IVIS® 200 Spectrum CT, PerkinElmer) and analyzed by the Living Image Software Version 4.5.2.18424. In addition, tumors were extracted, and fluorescence was monitored ex vivo.

## Immunohistochemical analysis of xenografts

Tumors were fixed in 4% paraformaldehyde, embedded in paraffin, and sections (4 µm) were stained with hematoxylin-eosin and safranin (HES). For detection of in situ cell death (Roche), sections were processed for proteolytic digestion with proteinase K (Roche) and incubated with TUNEL solution. Staining was visualized using the permanent red kit (Dako) and examined using a Zeiss Axiophot microscope. For immunohistochemistry, sections were incubated after heat-induced antigen retrieval with rat anti-mouse CD34 antibody (1/20) (Hycult technologies), rabbit monoclonal anti-Ki67 (1/200) (Neomarkers; LabVision), anti-cleaved caspase-3 (1/100) (Cell Signaling), visualized using the polink anti-rat kit (GBI Labs) and the peroxidase/diaminobenzidine Rabbit PowerVision kit (ImmunoVision Technologies), respectively. Slides were acquired with a Virtual Slides microscope VS120-SL (Olympus, Tokyo, Japan), 20X air objective (0.75 NA), and visualized using OLYMPUS OlyVIA software. TUNEL-stained cells were detected using the Definiens Tissue Studio software nuclei detection algorithm (Definiens AG, Munich, Germany). The analysis was performed on manually selected regions of interest by a qualified pathologist. Briefly, stained nuclei were automatically detected according to their spectral staining properties in the regions of interest. Thresholds on the area of the nuclei, set at 10 and 60 µm², were used to discard objects that were too small or too large from the analysis. The algorithm exports for each region of interest the number of stained cells and the total area in µm². The quantitative analysis represents the % of positive cells in the tumor tissue sections compared to the total number of tumor cells. Five to eleven fields were chosen for counting. ImmunoRatio ImageJ plugin was used to quantify Caspase-3, Ki-67, and CD34-positive cells. The average count in each region was used for statistical analysis by t-test. Necrotic fields were excluded. Results are means ± SEM of at least three independent experiments. Statistic tests and the number of biological replicates (n) per experiment are outlined in figure legends.

## Analysis of somatic mutations and copy number variations

MAPPYACTS samples were sequenced using the Illumina NextSeq 500 or Hiseq 2000/2500/4000 platforms [UMO1] to produce a library of 75-bp or 100-bp reads. Three different kits were used to perform the Exome capture: SureSelect XT human All exon CRE version 1 or 2 and Twist Human Core Exome Enrichment System. Sequencing libraries quality was estimated with fastqc and fastqscreen. Reads were mapped with BWA (v0.7.17 with parameters: -M -A 2 -E) onto the Human reference genome assembly hg19/GRCh37 120. SNVs (single nucleotide variations) and small indels were called using GATK3 121 (Indel Realigner, Base Recalibrator), samtools (fixmate, markdup, mpileup), and Varscan (v2.3.9) from paired normal/tumors bam files. Variant annotation and the functional prediction was performed with ANNOVAR 122 using public database releases on 2019.11.07 from the 1000 genomes project, Exome Aggregation Consortium, NHLBI-ESP project, Kaviar. Somatic mutations were filtered according to their enrichment in the tumor samples compared to the paired normal samples using a Fisher test and p value <0.001. To estimate the mutational spectrum, we excluded questionable somatic variants observed in less than 3 reads, or with an allele frequency lower than 0.05, or described in 1000 genomes and EXAC databases with a frequency higher than 0.05%. Non exonic variant, not covered by the Exome capture, were also excluded. Additionally, we performed allele-specific copy number profiling using the Sequenza algorithm (Favero et al, 2015). Focusing on the genomic region of DiPRO1/ZNF555 in our cohort study, segments with start or end in, or start before and end after, were considered to be related to the ZNF555 gene. CNV (copy number variation) was defined as gain: CN > ploidy; loss: CN < ploidy; LOH (Loss Of Heterozygosity): deletion of one allele; LOH copy neutral (cn): deletion of one allele, with a gain of the second allele and ultimately a copy number equal to the sample ploidy; Homozygous deletion: no DNA copy. Amplification was defined starting from 7 CN for a ploidy of 2. For 3 ≤ ploidy ≥ 9, the DNA copy cutoff was set according to the expression: ploidy x 2 + 1; and for 10 ≥ ploidy, the DNA copy cutoff was set according to ploidy + 11. For the correlation between gene expression and copy number, gene-wise CNV was regarded as −3 = homozygous deletion; −2 = hemizygous loss LOH; 1 = hemizygous loss without allele deletion; 0 = neutral/cn; 1 = gain without allele deletion; 2 = gain LOH; 3 = amplification. Z scores of gene expression levels were computed using tumor samples that are diploid for the corresponding gene. For each sample, the Z score = (x-u)/o (Shao et al, 2019), where u, o represent the mean and standard deviation of the DiPRO1 gene expression among diploid samples for the DiPRO1 gene, respectively; x represents the DiPRO1 gene expression in a sample with CNV. As illustrated in Fig. EV4B, the variation in DiPRO1 expression level is moderate, and thus, differentially expressed DiPRO1 was considered upregulated when Z scores ˃0.5 or downregulated when Z scores <−0.5. Spearman's correlation coefficient (r) test and nonlinear regression analysis were implemented to define the relationship between differential DiPRO1 gene expression and CNV using GraphPad Prism.

## scRNAseq data analysis

Publicly available scRNAseq datasets (GSE218974) (Danielli et al, 2023, Data ref: Danielli et al, 2023) derived from PDX primary

**The paper explained**

**Problem**

The specific consequences of pediatric cancer treatment are associated with exposure to chemotherapy and radiation during the rapid and dramatic physiological changes that can lead to specific tissue or organ damage, or alterations in growth and development. Two-thirds of survivors experience late effects, which may be severe or life-threatening. Therefore, there is a significant medical need for new, less-toxic therapeutic approaches, especially in the context of pediatric cancer treatment.

**Results**

In the study, the function of DiPRO1 (differentiation and proliferation protein) is explored in normal skeletal muscle cells and pediatric cancer cells. We have demonstrated a novel vulnerability that can be exploited through a nanomedicine approach (si/shRNA) in Rhabdomyosarcoma (RMS) and Ewing's sarcoma, suggesting a new target for therapeutic intervention. Furthermore, in addition to RMS and Ewing's sarcoma, gene sequencing data of primary recurrent or refractory patient tumors demonstrate elevated DiPRO1 expression in medulloblastoma, glioma, and neuroblastoma. This provides evidence supporting its clinical relevance for mesenchymal cancer treatment, with potential extension to neurogenic cancers.

**Impact**

Despite the ever-growing progress, epigenetic medicines still need improvement due to the lack of locus specificity, leading to off-target effects. Progress in nanotechnology has facilitated the development of nanomedicines based on inorganic nanoparticles for the transport and delivery of small RNA and DNA molecules. These nanomedicines can act with high precision and efficacy to silence target genes. Thus, DiPRO1-inhibiting nucleic acid-based nanomedicines could be an attractive target, contributing to the development of precision pediatric medicine. Moreover, considering that DiPRO1 depletion in normal cells does not induce cell death, the selective action represents a potentially less-toxic approach for treating young patients.

RMS cultures ($n = 3$ eRMS and $n = 3$ aRMS) were reanalyzed using Seurat R toolkit. We normalized, feature-selected, and integrated the individual data samples using the SCTransform function, followed by the FindIntegrationAnchors function. The expression density analysis was performed using scCustomize R package. The coexpression analysis was performed using stat_cor(method = "pearson") R function.

## Functional analysis

Gene ontology term enrichment was analyzed using Cytoscape software with the ClueGO and the CluePedia plug-ins (apps.cytoscape.org/apps/cluego) (Bindea et al, 2009) to decipher gene ontology annotation networks and functionally clustered pathways. Benjamini-corrected term values of $p < 0.05$ were considered statistically significant. The two-tailed hypergeometric test and Bonferroni step-down correction method were used. Group values of $p < 0.05$ were considered statistically significant. The parameters were as follows: Kappa score = 0.4, H. sapiens [9606], genes in GO_BiologicalProcess-EBI-QuickGO-GOA. GO annotation groups "GO_Muscle" and "GO_CellCycle" were downloaded from the Gene Ontology annotation database. Regulator prediction was performed using the iRegulon V1.3 (Janky et al, 2014). Differential transcript sets were analyzed

using Ingenuity Pathway Analysis (QIAGEN Inc, version 107193442). Significantly overrepresented pathways ($p < 0.05$) were identified with a right-tailed Fisher's Exact test that calculates an overlap $p$ value. The activation status of each pathway was also predicted using z-score calculation. This metric assesses the activation (positive z-score) or repression (negative one) of each term. Positional gene enrichment analysis was performed using the PGE tool (De Preter et al, 2008). Molecular signatures dataBase (MSigDB) and gene set enrichment (GSEA) software were employed to analyze the gene set according to their hallmarks and target genes (Subramanian et al, 2005; Liberzon et al, 2015, Data ref: Liberzon et al, 2015). The Venn diagrams were made using Venny 2.1 (bioinfogp.cnb.csic.es/tools/venny/) and Jvenn tools (Bardou et al, 2014).

## For more information

CpG Islands, Unmasked CpG, RepeatMasker, TxnFactrChIPE3 and BroadChromHMM, Candidate Cis-Regulatory Elements (cCREs) tracks of UCSC Genome Browser ENCODE: https://hgdownload.soe.ucsc.edu
CpG sites: http://methylqa.sourceforge.net/
Galaxy Community Hub: https://github.com/galaxyproject/galaxy
EaSeq: https://easeq.net
CRAPome database: http://www.crapome.org
Perseus: https://maxquant.net/perseus/
Homer: http://homer.ucsd.edu/homer
COMPARTMENTS: http://compartments.jensenlab.org/
VarElect NGS Phenotyper: https://varelect.genecards.org
MEME suite: https://meme-suite.org/meme
GREAT: http://great.stanford.edu/public/html
ImageJ: https://imagej.net/
Cytoscape: https://github.com/cytoscape/cytoscape
TumGrowth: https://github.com/tonedivad/TumGrowth
DEBrowser: https://github.com/UMMS-Biocore/debrowser
PAVIS: https://maayanlab.cloud/datasets2tools/landing/tool/PAVIS
Annovar: http://www.openbioinformatics.org/annovar
Cosmic: https://cancer.sanger.ac.uk/cosmic
Kaviar: http://db.systemsbiology.net/kaviar/
ECR Browser: https://ecrbrowser.dcode.org
Ensembl: www.ensembl.org
The MAPPYACTS clinical trial data request form https://redcap.gustaveroussy.fr/redcap/surveys/?s=DYDTLPE4AM

## Data availability

Transcriptome data: EMBL-EBI ArrayExpress E-MTAB-10822 https://www.ebi.ac.uk/biostudies/arrayexpress/studies/E-MTAB-10822. MIRA-seq data: EMBL-EBI ArrayExpress E-MTAB-10825 https://www.ebi.ac.uk/biostudies/arrayexpress/studies/E-MTAB-10825. and EMBL-EBI ArrayExpress E-MTAB-13689. https://www.ebi.ac.uk/biostudies/arrayexpress/studies/E-MTAB-13689?key=e206a0c1-4001-4834-bb96-55a1ec9dd9cc. Protein interaction AP-MS data: PRIDE PXD027542. UCSC browser public session of DiPRO1 differentially methylated CGI using MIRA-seq analysis: https://genome.ucsc.edu/cgi-bin/hgTracks?hgS_doOtherUser=submit&hgS_otherUserName=dr_finder&hgS_otherUserSessionName=metCGI_ZNF55%2FDiPRO1_2023.

The source data of this paper are collected in the following database record: biostudies:S-SCDT-10_1038-S44321-024-00097-z.

## Peer review information

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

## Acknowledgements

Dedicated to the talented scientists, Dr. Andrey Maksimenko, Harrison School of Pharmacy, Auburn University, USA, and Dr. Vaslily Ogryzko, INSERM, CNRS, Gustave Roussy, France, who contributed significantly to this work and passed away unexpectedly. We are grateful to Prof. Valisly OGRYZKO for his contribution and helpful advice on the development of the experimental strategy. We thank Dr. Andrey MAKSIMENKO (Harrison School of Pharmacy, Auburn University, USA) and Dr. Estelle DAUDIGEOS (Gustave Roussy, Villejuif, France) for their expert assistance in the design of the nanomedicine and in vivo experiments. We also thank the proteomics facilities for the qualified validation of the proteomic data (proteomic platform of the Institut Gustave Roussy, Villejuif; 3P5 of the Université Paris-Descartes) and François GILLETTE (Hays) for the computer assistance. We thank Juliette HAMROUNE, a member of the GENOM'IC facility of Cochin Institute) for qualified sequencing and upstream analysis. This study used patient data collected as part of the MAPPYACTS trial supported by the PHRC-K14-175 grant from the Institut National du Cancer, the MAPY201501241 grant from the ARC Foundation, and the Association Imagine pour Margo. We thank the Encode Consortium and the Encode production laboratories that generated the datasets. We thank the MyoBank AFM-Institut de Myologie (Paris, BB-0033-00012), Dr. Vincent MOULY (Sorbonne University, Inserm, Institut de Myologie, U974, Centre de Recherche en Myologie, Paris) for providing myoblasts. We would like to express our gratitude to Laurent BERNARD and Mathieu POIROT (DRH, CNRS Délégation Paris-Villejuif) for their administrative support and to Corinne GAULTON for the copy editing. We are grateful to the Federation Enfants, Cancers et Sante (FES) and the Societe Française de Lutte Contre les Cancers et les Leucemies de l'Enfant et de l'Adolescent (SFCE) for the support, which allowed to advance the project significantly (W381005059) to IP, BG). This work was also supported, Plan Cancer/INSERM, Environnement et cancer (N°ENV201416) to IP, VO, Gustave Roussy Transfert, Proof of concept (X75390) to IP and the Taxe d'Apprentissage (Institute Gustave Roussy) to JR/IP. BG was also supported by the 'Parrainage médecin-chercheur' of Gustave Roussy.

## Author contributions

**Jeremy Rich**: Conceptualization; Data curation; Software; Formal analysis; Validation; Investigation; Visualization; Methodology; Project administration. **Melanie Bennaroch**: Validation; Investigation; Visualization; Methodology. **Laura Notel**: Investigation; Methodology. **Polina P**atalakh: Investigation; Methodology. **Julien Alberola**: Investigation; Methodology. **Fayez Issa**: Resources; Methodology. **Paule Opolon**: Investigation; Methodology. **Olivia Bawa**: Data curation; Software; Formal analysis; Methodology. **Windy Rondof**: Data curation; Software; Formal analysis; Investigation. **Antonin Marchais**: Data curation; Software; Formal analysis. **Philippe Dessen**: Data curation; Software; Formal analysis. **Guillaume Meurice**: Data curation; Software; Formal analysis. **Morgane Le-Gall**: Data curation; Software; Formal analysis. **Melanie Polrot**: Data curation; Investigation; Methodology. **Karine Ser-Le Roux**: Investigation; Methodology. **Kamel Mamchaoui**: Resources. **Nathalie Droin**:

Formal analysis; Methodology. **Hana Raslova**: Investigation; Methodology; Writing—review and editing. **Pascal Maire**: Conceptualization; Investigation; Methodology; Writing—review and editing. **Birgit Geoerger**: Resources; Funding acquisition; Writing—original draft; Project administration; Writing—review and editing. **Iryna Pirozhkova**: Conceptualization; Resources; Data curation; Software; Formal analysis; Supervision; Funding acquisition; Validation; Investigation; Visualization; Methodology; Writing—original draft; Project administration; Writing—review and editing.

Source data underlying figure panels in this paper may have individual authorship assigned. Where available, figure panel/source data authorship is listed in the following database record: biostudies:S-SCDT-10_1038-S44321-024-00097-z.

## Disclosure and competing interests statement

The authors declare no competing interests.

# Expanded View Figures

**Figure EV1.  DiPRO1 regulates myogenic genes of human myoblasts.**

(A–C) Myoblasts were stably transduced with a retroviral vector expressing the DiPRO1 ORF (pDiPRO1, $n = 6$) and compared with their parental counterparts (Ctl, $n = 6$). (A) Heatmap revealing differential expression of a set of genes in control (blue) and pDiPRO1 (red) samples. Upregulation of DiPRO1 gene expression (B) stimulates expression of PAX-family genes (C). (D) Cell death analysis of shDiPRO1 myoblasts. The cells transduced with lentiviral vector expressing shDiPRO1 or nontargeting shCtl, were stained with propidium iodide (PI) and analyzed by flow cytometry. The percentage of dead cells according to DNA content was compared to control cells, proportions (%) ± SD, $n = 3$ corresponding to three independent experiments. (E–G) Human myoblasts were transduced with a lentiviral vector expressing a DiPRO1-targeting shRNA (shDiPRO1, $n = 7$) and compared to myoblasts transduced with the equivalent vector expressing a nontargeting shRNA (shCTL, $n = 6$). The analysis was performed one week post-transduction. (E) Heatmap revealing changes in gene expression profile related to DiPRO1 knockdown in myoblasts. (F) DiPRO1 expression was significantly inhibited by shDiPRO1 (red) relative to shCtl (blue). (G) Clustered functional network of 185 upregulated (UP) and 136 downregulated (DW) genes encompassing muscle-related functions, negatively correlated with DiPRO1 expression in myoblats. Muscle GO terms are represented by nodes. Two networks were confronted. Green clusters correspond to upregulated myogenic genes, while lilac clusters correspond to downregulated genes in shDiPRO1 versus shCtl. Cytoscape software with ClueGO plug-in was used, kappa score = 0.4. (H) PCA analysis of differentially expressed genes separates pDiPRO1 ($n = 6$) and control according to the first principal component and shDiPRO1 ($n = 7$) and control ($n = 6$) according to the second principal component. (I) DiPRO1 upregulation in RMS and Ewing's sarcoma (ES) cell lines. RNA-seq data were extracted from the DepMap portal (Broad Institute). The whisker plot shows median (Q2) ± Q3-Q1 ± 1.5x IQR of ZNF555 expression by cancer type or in all cancer types (ALL). *** ($P < 0.001$) indicates a significant difference with ALL. (J) Stable knockdown of RMS cells was achieved by transduction of lentiviral vectors producing nontargeting shRNA (shCtl) or DiPRO1-targeting shRNAs (shDiPRO1). Cell cycle analysis of TE671/RMS cells lacking DiPRO1 was performed 48 and 72 h after transduction with vectors expressing DiPRO1-targeting shRNA. Transduced RMS cells were fixed in ethanol and stained with propidium iodide (PI). DNA content and percentage of cells in each cell cycle phase were determined by flow cytometry. The results represent two independent experiments. (K) RMS and myoblast (Myo) cells transfected with shCtl and shDiPRO1 were assayed for caspase-3 activity. The results were normalized to the protein quantity and appropriate shCtl was referenced as 100%. All conditions represent three or six transfections (RMS $n = 6$, Myo $n = 3$), proportions (%) ± SD. (L) DiPRO1 induces RMS cell proliferation. RMS cells were stably transduced with a retroviral vector expressing the DiPRO1 ORF (pDiPRO1) and compared with their parental counterparts (Ctl). The proliferation rate was estimated for DiPRO1-overexpressing RMS (pDiPRO1) and their control (Ctl) by Counting Kit 8 (Sigma-Aldrich). Cell number was determined using a titration curve performed at different cell dilutions from 0 to 25,000 cells/well. Cells were then seeded at $2.0 \times 10^3$, $5.0 \times 10^3$, $1.0 \times 10^4$, and $1.5 \times 10^4$ cells/well and absorbance was measured at 460 nm after 72 h of proliferation. Viable cells were compared to the initial number of cells. Data were expressed as mean % relative to initial cell number ± SD and represent four independent experiments ($n = 4$). The titration curve for RMS-Ctl is presented in Fig. 2E. Data information: In (A–C, E–H), Global transcriptome analysis of microarray was implemented using total mRNA. The Limma R package was used for statistical analysis. In (B, C, F, I), boxplots represent median (line) ± dispersion, box length IQR = Q3-Q1, whisker length = 1.5 * IQR. In (D, I–L), Welch's two-sample $t$-test. In (B–D, I, K, L), *,**, and *** indicate significant differences with the corresponding control, $P < 0.05$, 0.01, and 0.001, respectively. Source data are available online for this figure.

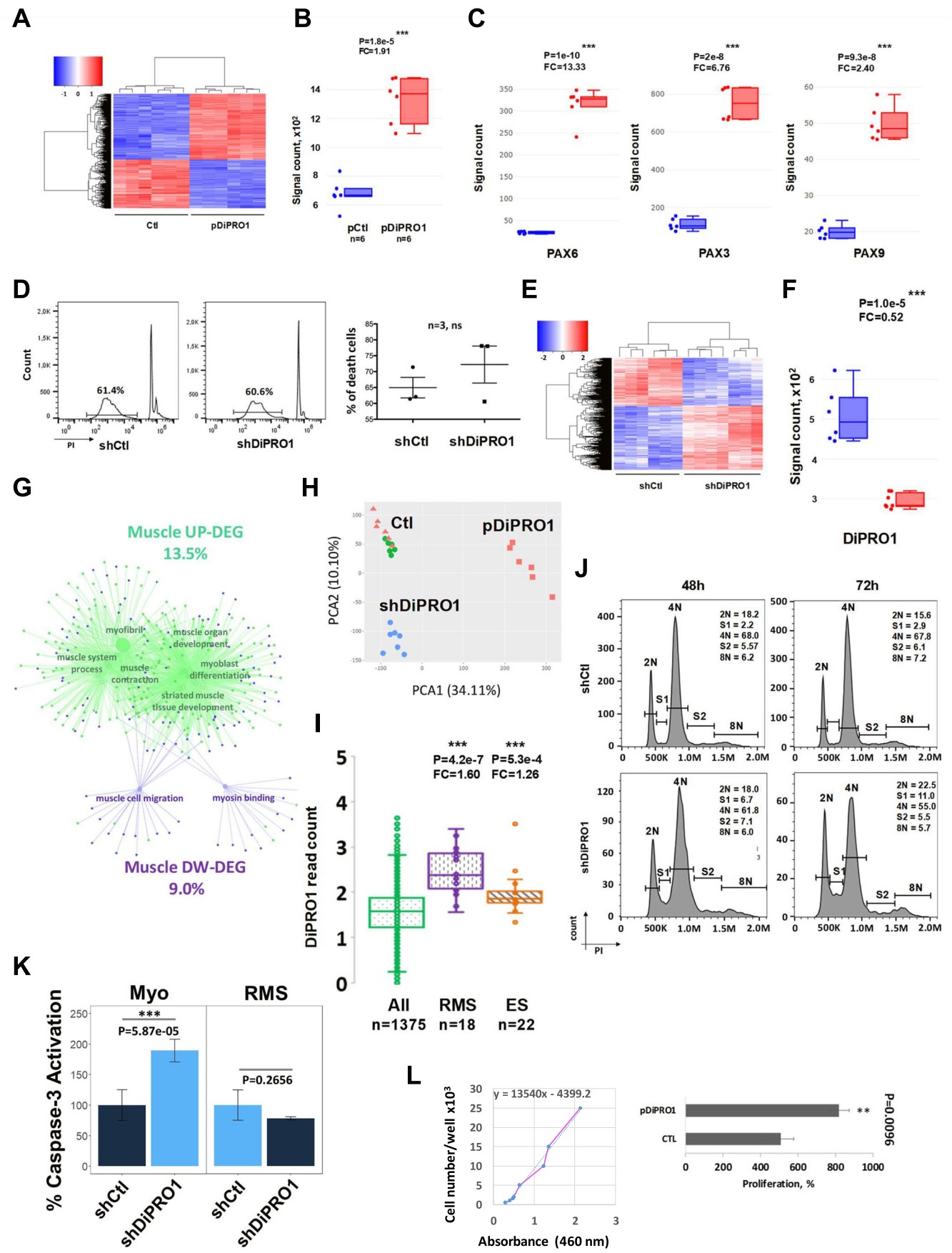

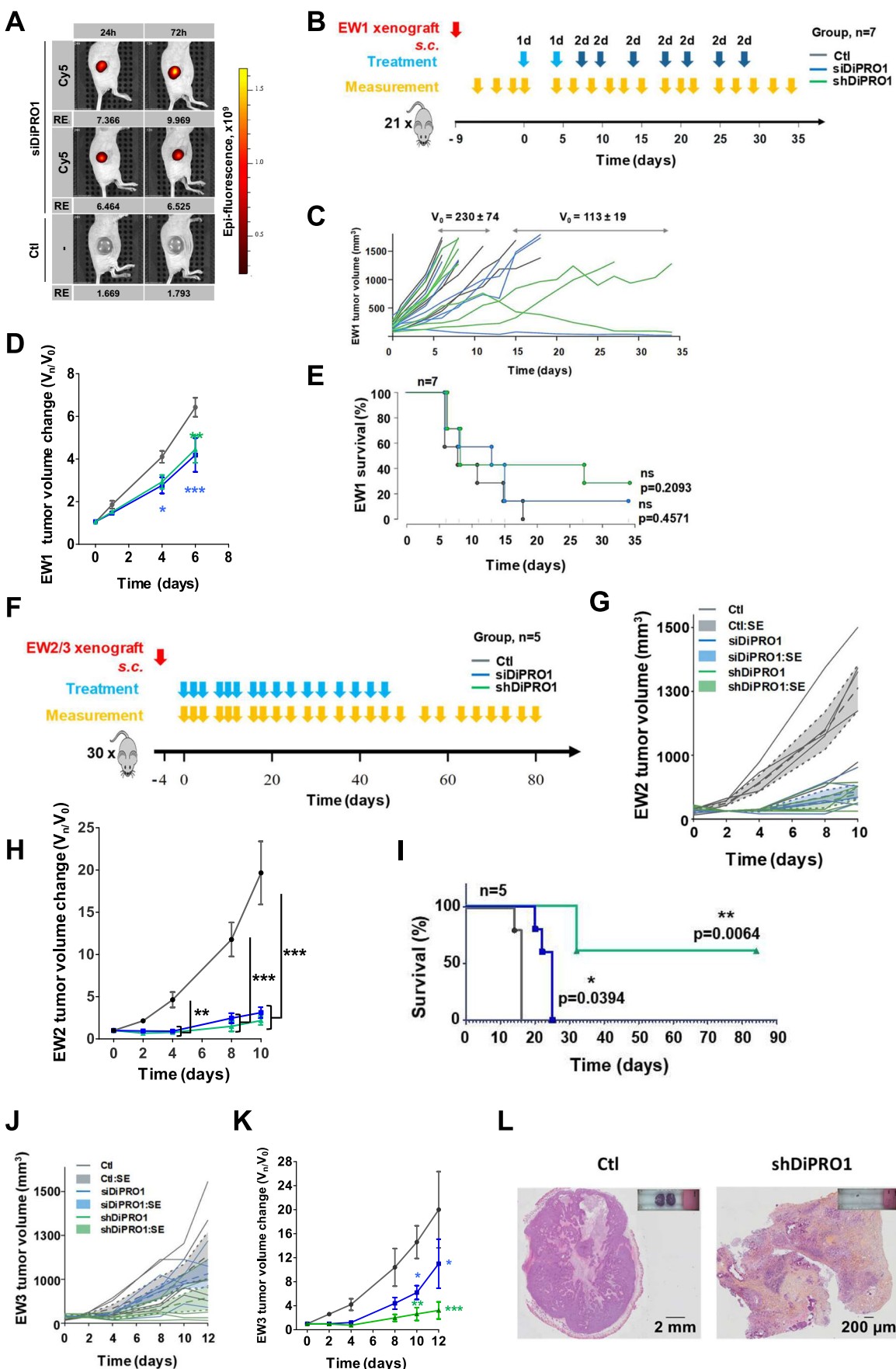

◄ **Figure EV2. DiPRO1 inhibitors compromise tumor growth in Ewing's sarcoma xenograft models.**

Nude mice received s.c. inoculations of A673 Ewing's sarcoma (EW) cells and were treated with siDiPRO1/jetPEI®, shDiPRO1/jetPEI®, and siCtl/jetPEI® scramble (Ctl) nanocomposites at 0.5 or 1 mg/kg/d. Three independent experiments were performed. (A) Tumor cell uptake of siDiPRO1/jetPEI®/Cy5 nanocomposites. Specific anti-DiPRO1 siRNA complexes coupled to Cy5 were delivered at a single dose (0.5 mg/kg/) by intratumoral injection into two nude mice bearing Ewing sarcoma tumors. Internalization was monitored in live mice 24 and 72 h post-treatment and radiant efficacy (RE) was measured. Control mice were treated with an equimolar dose of nontargeting siCtl/jetPEI® free of Cy5. (B–E) EW1 xenograft model. (B) Schematic showing the timing of tumor injection and treatment with DiPRO1 inhibitors. Mice were treated for 25 days at a dose of 0.5 mg/kg/inj (1d) twice-weekly for the first week, and then the dose was doubled (2d) for the rest of the treatment. (C) Growth curves of individual EW1 tumors (V, mm3). (D) Tumor progression was compared between the treated and control groups ($n = 7$). (E) Kaplan–Meier survival curves for treated mice. Overall survival was followed for 34 days ($n = 7$). The log-rank test (Mantel-Cox) was applied for each pairwise comparison, proportion (%) ± SEM. The $p$ value was adjusted by Holm method. (F–L) EW2 and EW3 xenograft models. (F) Schematic showing the timing of tumor injection and treatment with DiPRO1 inhibitors. Mice were treated with siDiPRO1/jetPEI®, shDiPRO1/jetPEI®, and scrambled siRNA/jetPEI® (Ctl) nanocomposites for 80 days at a dose of 0.5 mg/kg/d (1d). The intensive three-dose-per-week regimen was used for the first 2 weeks, followed by a two-dose-per-week regimen until the end of treatment. (G) Growth curves of individual EW2 tumors (V, mm3). (H) EW2 tumor progression was compared between the treated and control groups ($n = 5$). (I) Kaplan–Meier survival curves for treated mice bearing EW2 tumor xenografts. Overall survival was followed for 84 days. The log-rank test (Mantel-Cox) for pairwise comparison with control was applied for analysis, proportion (%) ± SEM. The $p$ value was adjusted by the Holm method. (J) Growth curves of individual EW3 tumors (V, mm3). (K) EW3 tumor progression was compared between treatment and control groups ($n = 5$). (L) Representative HES staining of tumor sections from a scrambled control and an Ewing's sarcoma tumor (EW2) treated with shDiPRO1/jetPEI® shows the absence of viable tumor cells in the treated residual tumor 84 days after xenografting. Data information: In (D, H, K), a two-way ANOVA test was used for statistical analysis, $V_n/V_0$ ratio ± SEM. In (D, E, H, I, K), *$P < 0.05$, **$P < 0.01$, ***$P < 0.001$. Source data are available online for this figure.

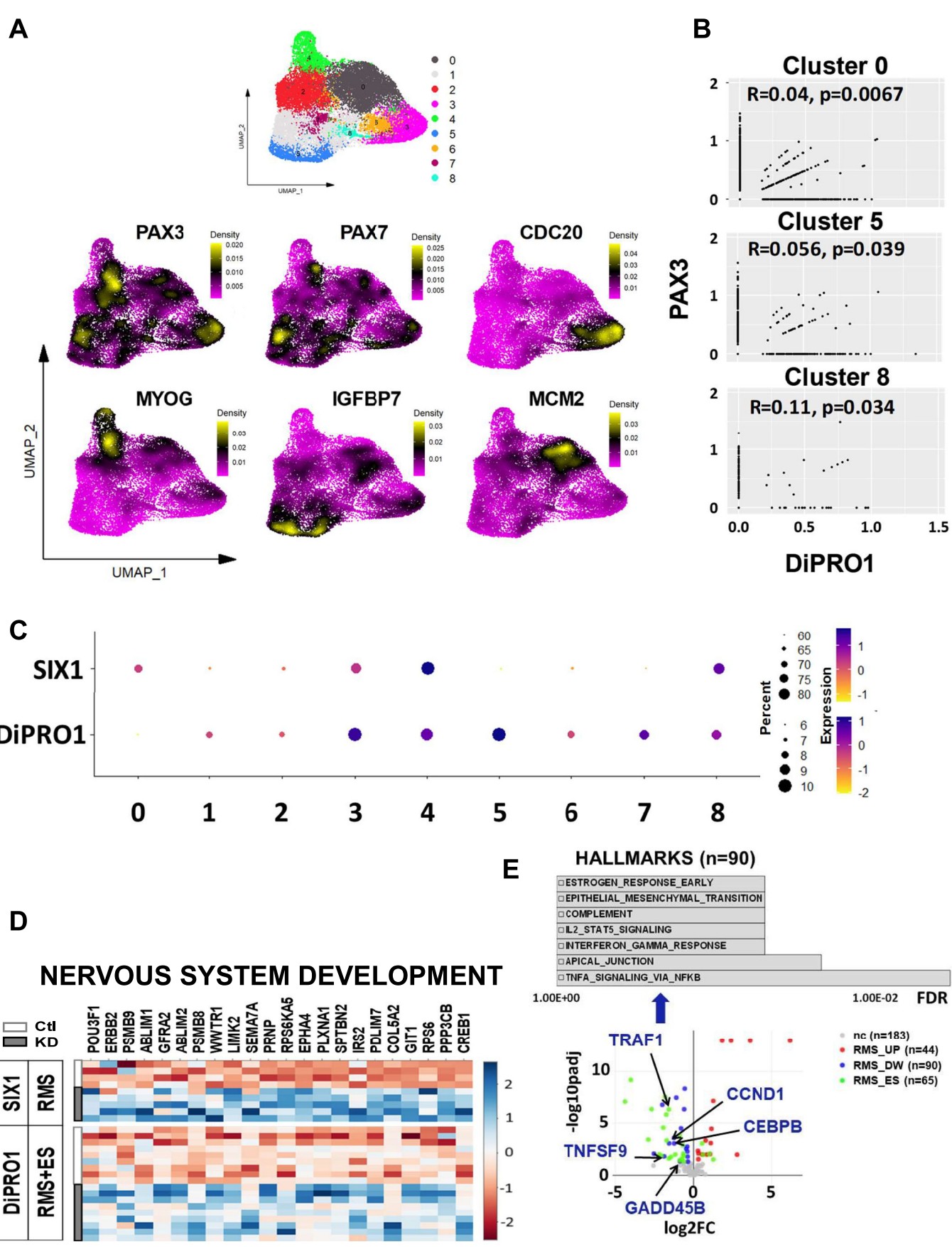

◄ **Figure EV3. Expression profiling of DiPRO1, SIX1, and their target genes in PDX tumors and tumor biopsies of RMS and Ewing sarcoma patients.**

(A–C) scRNAseq datasets (GSE218974) derived from PDX primary RMS cultures ($n = 3$ eRMS and $n = 3$ aRMS) were reanalyzed using Seurat R toolkit. The individual data were normalized, feature-selected, and integrated into the individual data samples using the SCTransform function. (A) Integrated UMAP plot showing RMS cell populations expressing lineage-specific marker genes. Expression density analysis was performed using the R package scCustomize. (B) Coexpression of DiPRO1 and PAX3 genes in muscle/mesenchymal progenitors and S-phase cells using stat_cor(method = "pearson") R function. Pearson correlation coefficient (R) and *p* value represents the statistical significance of the linear relationship between two gene expression, $n = 6$. (C) Dot plot showing DiPRO1 and SIX1 gene expression in different RMS cell populations. (D) Heatmap of common expression gene set of the nervous system development KEGG pathway across DiPRO1 KD ($n = 9$) and SIX1 KD ($n = 4$) samples and appropriate controls of RMS and Ewing sarcoma (ES) cells. (E) Volcano plot representing differently upregulated (red) and downregulated (blue) DiPRO1 *cis*-target genes in RMS ($n = 37$) compared to other pediatric tumors (Other, $n = 265$), marked in blue. Common DEGs with Ewing sarcoma (ES, $n = 25$) are highlighted in green. Hallmark enrichment analysis of downregulated DEGs ($n = 90$) (blue arrow) in RMS was performed using the MSigDB database (FDR <0.05). Data information: In (E), RNA-seq data of primary tumor biopsies were processed using DEBrowser, DESeq2 (TMM normalization, local fit, LRT test), *p*Vadj <0.05.

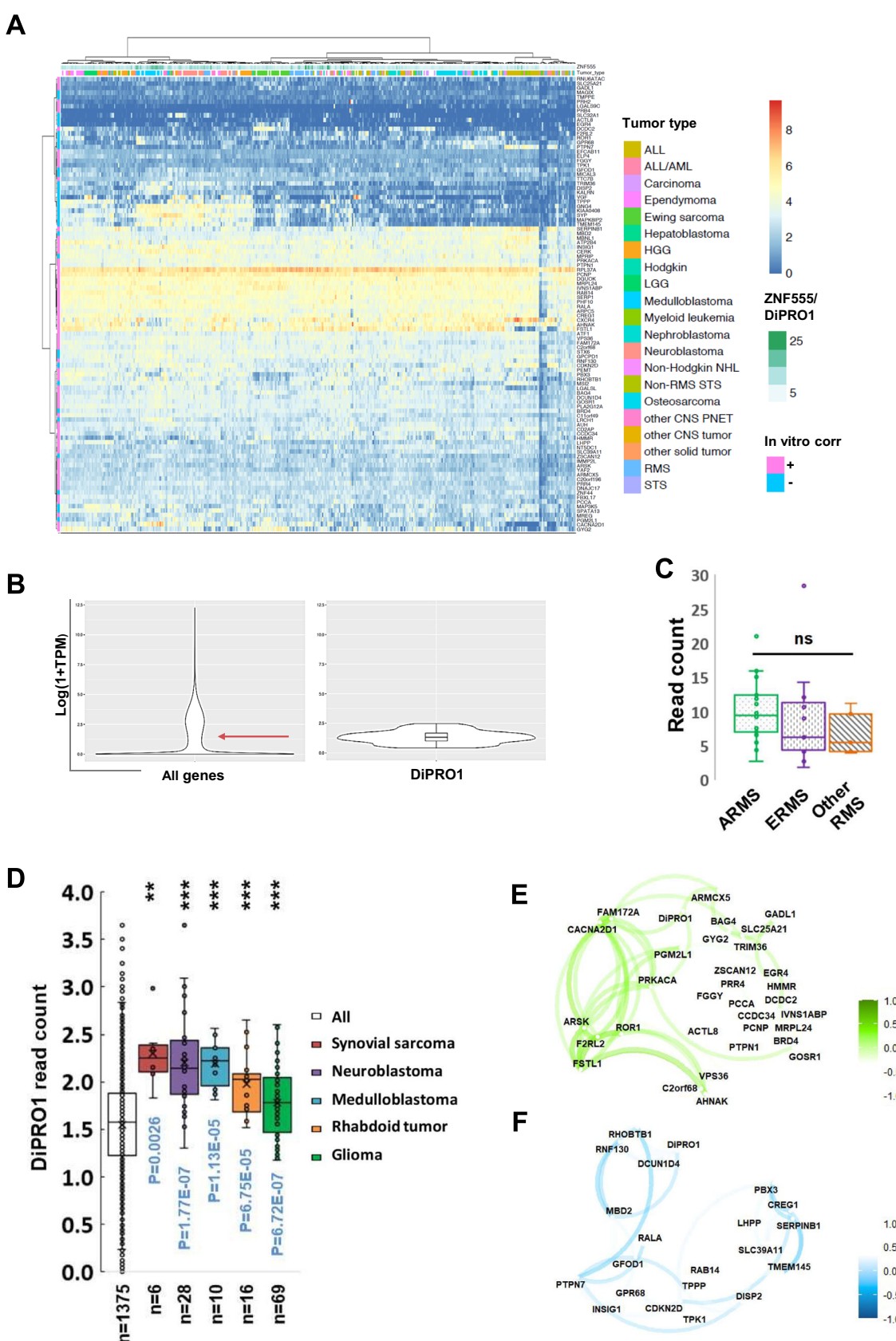

◀ **Figure EV4. Clinical relevance of the DiPRO1 gene and its downstream targets in cancer.**

(A) Unsupervised clustering analysis shows upregulation of the ZNF555/DiPRO1 gene in several tumor types. Genes positively (+) and negatively (−) correlated with DiPRO1 expression in in vitro DiPRO1 knockdown experiments are shown on the left side. (B) The variation in DiPRO1 expression levels among other genes in pediatric tumors ($n = 327$) Violin plots show the distribution of gene expression for all genes (left) and specifically for DiPRO1 (right). The median of the DiPRO1 expression is denoted by the red arrow. Additionally, the box plot presents a median (Q2) ± Q3-Q1 ± 1.5x IQR. (C) Boxplots of DiPRO1 expression in RMS subtypes; ARMS: alveolar RMS ($n = 21$), ERMS: embryonic RMS ($n = 11$) and other RMS ($n = 5$). The data are presented as median (Q2) ± Q3-Q1 ± 1.5x IQR. Statistical analysis using the $t$-test showed no significant (ns) difference between RMS subtypes ($P > 0.1591$). (D, E) RNA-seq data of cancer cell lines were retrieved from the DepMap portal (Broad Institute). (D) The whisker plot displays DiPRO1 expression across different cancer types: synovial sarcoma ($n = 6$), neuroblastoma ($n = 28$), medulloblastoma ($n = 10$), rhabdoid tumor ($n = 16$), and glioma ($n = 69$), as well as in all cancer types combined ($n = 1375$) from Dataset EV2. The results depict mean (cross) and median (Q2) ± Q3-Q1 ± 1.5x IQR. $t$-test was used for analysis. **$P < 0.01$, ***$P < 0.001$. $P$ values are shown between samples of the indicated cancer type and all cancer samples. Network plot of positive (E) and negative (F) expression correlation of DiPRO1 downstream targets across cancer cell lines ($n = 1375$). Variables that are more highly correlated appear closer together and are linked by paths (green for positive and blue for negative). The affected DEGs in RMS and Ewing's sarcoma tumor samples were analysed across cancer cell lines. Data information: In (A–E), RNA-seq data were processed using DEBrowser, the R package DESeq2 (TMM normalization, parametric adjustment, LRT test), FC >1.4, $pV < 0.05$. In (C, D), $t$-test was used. In (A, E, F), analysis was performed using the Corrr R package (default parameters).

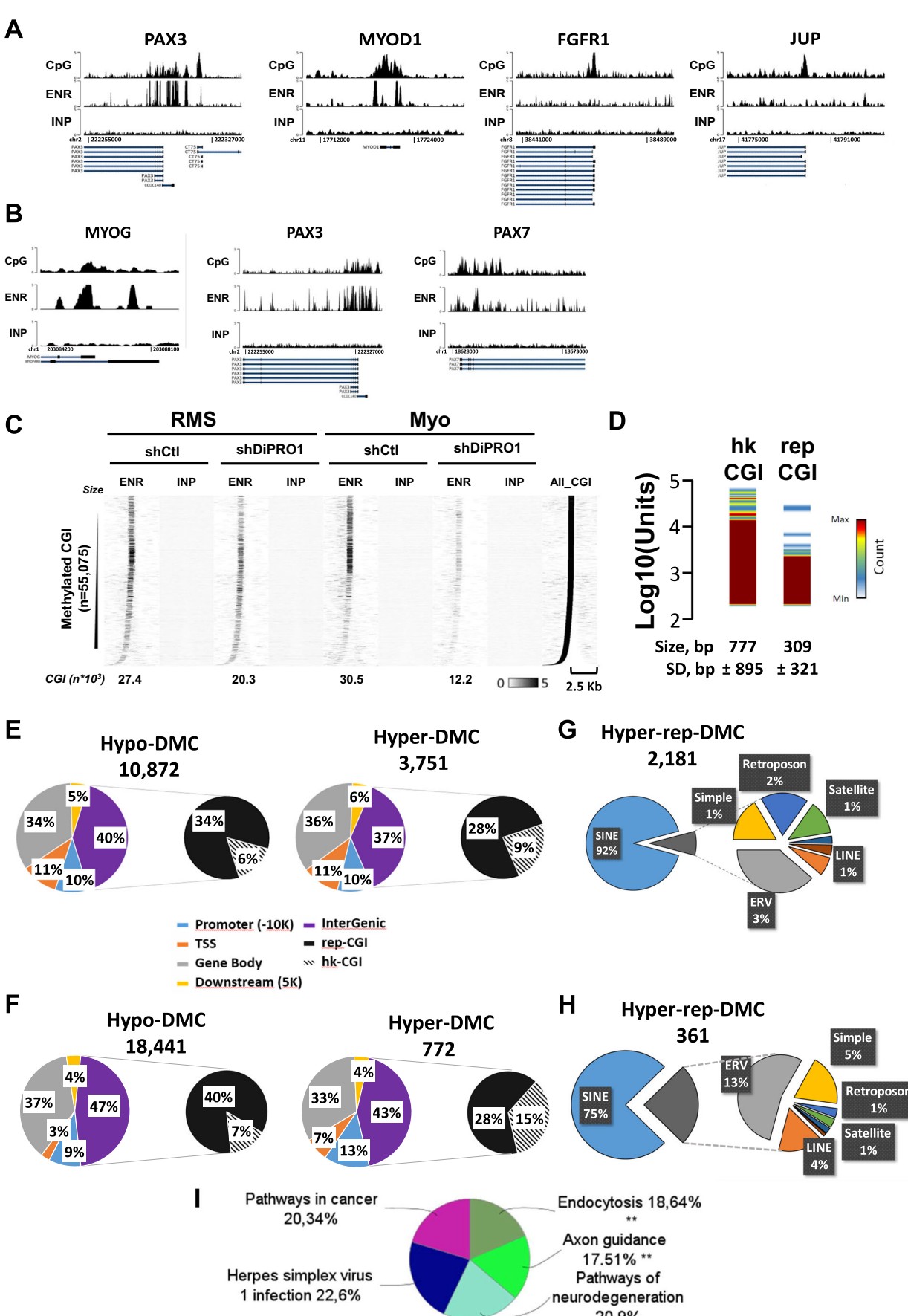

◀ **Figure EV5.  Methylation changes in different CGI populations related to DiPRO1 inhibition in myoblasts and RMS cells.**

Methylation by MIRA-seq was analyzed in DNA samples from TE671 RMS cells and human myoblasts (Myo). Cells expressing a DiPRO1-targeting shRNA (shDiPRO1) were compared to control cells expressing a nontargeting shRNA (shCtl). Signals from methylation-enriched DNA (ENR) were normalized to the unenriched input DNA (INP). CGI, CpG islands according to Gardiner-Garden and Frommer criteria; DMC, differentially methylated CGI; hk-DMC, housekeeping DMC w/o repetitive elements; rep-DMC, DMC with repetitive elements. Analysis was performed using EaSeq software. (**A**) Tracking of signals on both strands corresponding to methylation signals within CGI regions of PAX3, MYOD1, FRGR1, and JUP genes in RMS control cells. The Y-axis represents signal intensity; the X-axis represents hg38 genomic coordinates. (**B**) Heatmaps displaying methylation levels in size-sorted CGIs in shDiPRO1 and shCtl and corresponding inputs. The bar reflects the signal intensity. Y-axis: DNA fragments per 1 M reads per 1 K. X-axis: 2.5 Kb surrounding each region. (**C**) Density plot of the log-transformed size of hk- and rep-CGI in the human genome. (**D**) Tracking of signals on both strands corresponding to methylation signals within CGI regions of MYOG, PAX3, and PAX7 genes in control myoblasts. The Y-axis represents signal intensity; the X-axis represents hg38 genomic coordinates. (**B**) Genomic distribution of hypo- (top) and hypermethylated (bottom) DMC is shown as % of DMC total number. (**E, F**) Genomic distributions of hypo- (left) and hypermethylated (right) DMC are shown as % of DMC total number, in RMS cells (**E**) and myoblasts (**F**). (**G, H**) DNA repeat class distribution within DiPRO1-linked hypermethylated DMC regions in RMS cells (**G**) and myoblasts (**H**). (**I**) KEGG-based functional analysis of genes associated with hypomethylated DMC regions of RMS cells under DiPRO1 KD. *P* value of the group corrected with Bonferroni step-down, kappa score = 0.4. ClueGO, a Cytoscape plug-in was used.

