## [Peer Review File · EMBO Molecular Medicine]

DiPRO1 distinctly reprograms muscle and mesenchymal cancer cells

Jeremy RICH, Melanie BENNAROCHE, Laura NOTEL, Polina PATALAKH, Julien ALBEROLA, Fayez ISSA, Paule Opolon, Olivia BAWA, Windy RONDOP, Antonin MARCHAIS, Philippe Dessen, Guillaume Meurice, Morgane Le Gall, Melanie POLROT, Karine SER-LE ROUX, Kamel Mamchaoui, Nathalie DROIN, Hana RASLOVA, Pascal Maire, Birgit Georger, and Iryna PIROZHKOVA

Corresponding author: Iryna PIROZHKOVA (iryna.pirozhkova@inserm.fr)

Review Timeline:

Submission Date:	10th Jan 23
Editorial Decision:	10th Feb 23
Revision Received:	18th Jan 24
Editorial Decision:	11th Mar 24
Revision Received:	24th May 24
Editorial Decision:	11th Jun 24
Revision Received:	13th Jun 24
Accepted:	18th Jun 24

Editor: Lise Roth

Transaction Report:

10th Feb 2023

Dear Dr. Pirozhkova,

Thank you for submitting your work to EMBO Molecular Medicine. We have now heard back from the referees who agreed to evaluate your manuscript. As you will see below, the reviewers raise substantial concerns on your work, which unfortunately preclude its publication in EMM in its current form.

The reviewers find that the question addressed by the study is of potential interest, but they also raise a number of partially overlapping concerns which overall point to a lack of validation of several aspects of the manuscript.

Addressing the reviewers concerns in full will be necessary for further considering the manuscript in our journal, except from point #1 from referee #2 (PDX/patient explants/humanized mouse model). Indeed, we acknowledge that these experiments might require a lot of additional time and work and are not required for the data to support the conclusions. If you decide not to perform these experiments, we would nevertheless welcome a discussion on this point. Moreover, please note that genomics and proteomics data should be deposited in public repositories.

Revising the manuscript according to the referees' recommendations appears to require a lot of additional work and experimentation, and given the potential interest of your findings, we are ready to extend the deadline to 6 months with the understanding that acceptance of the manuscript would entail a second round of review.

We require:

- 1) A .docx formatted version of the manuscript text (including legends for main figures, EV figures and tables). Please make sure that the changes are highlighted to be clearly visible.
- 2) Individual production quality figure files as .eps, .tif, .jpg (one file per figure). For guidance, download the 'Figure Guide PDF' (<https://www.embopress.org/page/journal/17574684/authorguide#figureformat>).
- 3) At EMBO Press we ask authors to provide source data for the main figures. Our source data coordinator will contact you to discuss which figure panels we would need source data for and will also provide you with helpful tips on how to upload and organize the files.
- 4) A .docx formatted letter INCLUDING the reviewers' reports and your detailed point-by-point responses to their comments. As part of the EMBO Press transparent editorial process, the point-by-point response is part of the Review Process File (RPF), which will be published alongside your paper.
- 5) A complete author checklist, which you can download from our author guidelines (<https://www.embopress.org/page/journal/17574684/authorguide#submissionofrevisions>). Please insert information in the checklist that is also reflected in the manuscript. The completed author checklist will also be part of the RPF.
- 6) It is mandatory to include a 'Data Availability' section after the Materials and Methods. Before submitting your revision, primary datasets produced in this study need to be deposited in an appropriate public database, and the accession numbers and database listed under 'Data Availability'. Please remember to provide a reviewer password if the datasets are not yet public (see <https://www.embopress.org/page/journal/17574684/authorguide#dataavailability>). Note that the Data Availability Section is restricted to new primary data that are part of this study.
- 7) For data quantification: please specify the name of the statistical test used to generate error bars and P values, the number (n) of independent experiments (specify technical or biological replicates) underlying each data point and the test used to calculate p-values in each figure legend. The figure legends should contain a basic description of n, P and the test applied. Graphs must include a description of the bars and the error bars (s.d., s.e.m.). Please provide exact p values.
- 8) Our journal encourages inclusion of *data citations in the reference list* to directly cite datasets that were re-used and obtained from public databases. Data citations in the article text are distinct from normal bibliographical citations and should directly link to the database records from which the data can be accessed. In the main text, data citations are formatted as

follows: "Data ref: Smith et al, 2001" or "Data ref: NCBI Sequence Read Archive PRJNA342805, 2017". In the Reference list, data citations must be labeled with "[DATASET]". A data reference must provide the database name, accession number/identifiers and a resolvable link to the landing page from which the data can be accessed at the end of the reference. Further instructions are available at .

9) We replaced Supplementary Information with Expanded View (EV) Figures and Tables that are collapsible/expandable online. A maximum of 5 EV Figures can be typeset. EV Figures should be cited as 'Figure EV1, Figure EV2" etc... in the text and their respective legends should be included in the main text after the legends of regular figures.

10) The paper explained: EMBO Molecular Medicine articles are accompanied by a summary of the articles to emphasize the major findings in the paper and their medical implications for the non-specialist reader. Please provide a draft summary of your article highlighting

11) For more information: There is space at the end of each article to list relevant web links for further consultation by our readers. Could you identify some relevant ones and provide such information as well? Some examples are patient associations, relevant databases, OMIM/proteins/genes links, author's websites, etc...

12) Author contributions: CRediT has replaced the traditional author contributions section because it offers a systematic machine readable author contributions format that allows for more effective research assessment. Please remove the Authors Contributions from the manuscript and use the free text boxes beneath each contributing author's name in our system to add specific details on the author's contribution. More information is available in our guide to authors.

13) Disclosure statement and competing interests: We updated our journal's competing interests policy in January 2022 and request authors to consider both actual and perceived competing interests. Please review the policy <https://www.embopress.org/competing-interests> and update your competing interests if necessary.

14) Every published paper now includes a 'Synopsis' to further enhance discoverability. Synopses are displayed on the journal webpage and are freely accessible to all readers. They include a short stand first (maximum of 300 characters, including space) as well as 2-5 one-sentences bullet points that summarizes the paper. Please write the bullet points to summarize the key NEW findings. They should be designed to be complementary to the abstract - i.e. not repeat the same text. We encourage inclusion of key acronyms and quantitative information (maximum of 30 words / bullet point). Please use the passive voice. Please attach these in a separate file or send them by email, we will incorporate them accordingly.

15) As part of the EMBO Publications transparent editorial process initiative (see our Editorial at <http://embomolmed.embopress.org/content/2/9/329>), EMBO Molecular Medicine will publish online a Review Process File (RPF) to accompany accepted manuscripts.

In the event of acceptance, this file will be published in conjunction with your paper and will include the anonymous referee reports, your point-by-point response and all pertinent correspondence relating to the manuscript. Let us know whether you agree with the publication of the RPF and as here, if you want to remove or not any figures from it prior to publication. Please note that the Authors checklist will be published at the end of the RPF.

I look forward to receiving your revised manuscript.

Yours sincerely,

Lise Roth

Please note: When submitting your revision you will be prompted to enter your funding and payment information. This will allow Wiley to send you a quote for the article processing charge (APC) in case of acceptance. This quote takes into account any reduction or fee waivers that you may be eligible for. Authors do not need to pay any fees before their manuscript is accepted and transferred to the publisher.

EMBO Press participates in many Publish and Read agreements that allow authors to publish Open Access with reduced/no publication charges. Check your eligibility: <https://authorservices.wiley.com/author-resources/Journal-Authors/open-access/affiliation-policies-payments/index.html>

**** Reviewer's comments ****

Referee #1 (Remarks for Author):

In this manuscript, Rich et al. present an interesting set of data regarding the role of DIPRO1 in both myogenesis and mesenchymal cancers. Using different approaches, their data support the view that DIPRO1 may be involved in regulation of cell state by modifying the expression pattern of genes via epigenetic modifications. Although their results and proposed therapeutic approach are innovative, the accumulation of data from different approaches and models may lead to confounding effects that could result in misinterpretations of the role of DIPRO1 in sarcoma tumorigenesis. Revisions are then needed to clarify these points.

1. Fig1-2: Chip-seq analyses have to be performed in myoblasts to strengthen the correlation with DE genes in myoblasts overexpressing or silenced for DIPRO1, and the subsequent identification of DIPRO1 specific targets.
2. Fig3: The rhabdomyosarcoma lines used are not the most consensual. The results obtained deserve to be validated in other lines such as RD (ERMS) or RH30 (ARMS) for example.
3. Fig3: Authors claim that DIPRO1 KD induces RMS cell apoptosis, based mostly on changes in the expression pattern of some effectors of the pathway. However, since these arguments are only correlative, they should instead define the impact of down regulation of DIPRO1 on caspase-3 activity, or show how ZVAD can block this cell death.
4. Fig3: Changes in apoptotic genes correspond to gain of both anti and pro-apoptotic genes. Progeny or Ingenuity scores should be applied to infer score of activation of the pathway.
5. Fig 2-3 :What is the overlap in DE genes in myoblasts and cancer cells models ? This could improve the identification of DIPRO1 specific targets, since the number of DE genes is huge and could lead to confounding effects.
6. Fig 4: What is the relevance of this therapeutic approach in an RMS model ?
7. Fig 5: The rationale of exploring MappyActs pan-cancers relapse cohort rather than publicly available transcriptomic data on RMS and EWS is not clear. What is the expression level of DIPRO1 in RMS compared to normal muscles ? What is the prognostic value of this gene in EWS/RMS cohorts that have already been published ? Is there a correlation between DIPRO1 and some of the target genes presented here ? Along the same line, what is the expression profile of DIPRO1 in the recently published single-cell analyses of RMS samples/models ? Is its expression restricted to mesenchymal-like progenitors cluster ?
8. Fig 5: The robustness and relevance of exploring the so-called downstream genes network is not that clear/convincing. Intersection between CHIPseq and RNAseq analyses performed in all overexpressing/downregulated models should be used to define this network and to strengthen the conclusions.
9. The link between SOX2/SIX1 and DIPRO1 across models is at least confusing, and perhaps even over interpreted. Authors should provide more evidences that there is a synergistic action of DIPRO1 and SOX2 or SIX1 on target genes, resulting respectively from epigenetic and transcriptional regulation, by showing for example that impact of DIPRO1 overexpression on the expression of these genes is blocked by downregulation of SIX1 or SOX2.
10. The fact that only 14.6% of DMCs correlate with changes in genes expression in RMS cells seems quite low. The authors should comment on this and perhaps provide some statistical insights. Same comment for the 12.4% of promoter regions bound by SIX1 and DMCs in DIPRO1 KD.

Minor points

1. Authors report their previous finding upon the role of DIPRO1 on ANT1 regulation. Regarding the pro-apoptotic role of ANT1, and its previously established role in RMS, how do the authors reconcile their finding that DIPRO1 down regulation induces apoptosis?
2. Some English/typo misspellings.

Referee #2 (Remarks for Author):

The authors show evidence that DiPRO1 is a transcriptional and epigenetic regulator that compromises normal and malignant cell fate in a selective manner supported by analysis of publicly available patient datasets and some functional data generated. The authors suggest that DiPRO1 is a positive regulator of proliferation and a negative regulator of differentiation of human myoblasts whereas it is a positive regulator of proliferation and prevents apoptosis in cancer cells. Importantly from a translational point of view and potential therapeutic strategy, the authors show that DiPRO1 inhibition is toxic to RMS and Ewing sarcoma cancer cells, but not normal. The therapeutic concept is applicable to paediatric cancers of mesenchymal origin which has impact on high unmet clinical needs and is novel in terms of target (DiPRO1) but also providing some in vivo Poc in terms of delivery.

Points for revision:

- Animal mouse studies not suitable (no orthologues) - and therefore model used is immunodeficient, any alternatives considered? (patient explants? organ/tumour on a chip? humanised mouse model?)
- Overexpression of DiPRO1 in vitro: is the recombinant protein overexpressed maintaining native conformation and cellular localisation (nuclear)?
- Depletion sh vs OE: would it be possible to generate mutants (to disrupt interaction with binding partners extensively referred to across the manuscript) and perform rescue experiments?
- Comparison with fibroblasts (could DiPRO1 have a role in myofibroblast differentiation that occurs in a tumour microenvironment giving rise to cancer associated fibroblasts? (commonly generated in vitro by treatment with TGFb)
- in vitro functional experiments don't address/match the suggested caspase activation for cell death induction in cancer cells (to mirror/correlate with the IHC for in vivo PoC), staining (IF or even flow cytometry to complement cell cycle analysis performed) would address this.
- In vivo PoC: what's the KD efficiency in si vs sh? Is it correlating with higher benefit observed with sh? (transient vs stable kd?) can IHC include DiPRO1 staining? (if not possible due to lack of Ab, RNAscope? Spatial transcriptomics?)
- As a theory/role in repression/de-repression of endogenous retrovirus, would be useful to explore a link with STING pathway (sensing of nucleic acid) which triggers an innate immune response via IFN I signalling, currently being explored within cancer immunotherapies strategies, experimentally testable looking at STING activation markers/IFN responsive genes (ISGs)
- typo in "Animals with established s.c. tumors were randomly assigned to treatment groups (n=7 or 5) receiving either nanocomposites of (i) shDiPRO1/jetPEI®, (ii) shDiPRO1/jetPEI® or (iii) scramble siRNA controls (Ctl)" where sh 2x instead of siDiPRO1?
- In fig2A/B: actin as loading control for WB shows differences between ctrl and DiPRO1 - is the graph taking into account normalisation to actin as well (any other loading control most suitable or total protein normalisation more accurate?)

Referee #3 (Comments on Novelty/Model System for Author):

The cell lines used in this study are standard ones. However, instead of knockdown approaches, knockout of DIPRO1 could have enhanced the study further.

Referee #3 (Remarks for Author):

The manuscript by Rich et al characterizes the human ZNF555 (DIPRO1) gene in myoblast differentiation and two mesenchymal sarcomas rhabdomyosarcoma and Ewing's sarcoma. Previously, the same group identified DIPRO1 as a transcriptional regulator of the Adenine Nucleotide Translocator 1 (ANT1) gene in facio-scapulohumeral muscular dystrophy. In this manuscript, the Authors initially perform analyses to shed light on how DIPRO1 is important in myogenesis using gain- and loss- of function approaches. They find that DIPRO1 shares consensus binding motif with SIX1, an upstream myogenic regulator. Extending their study, the Authors characterize DIPRO1 in rhabdomyosarcoma and Ewing's sarcoma, two mesenchymal tumors where DIPRO1 levels are high. Knockdown of DIPRO1 in these tumors lead to increased cell death and reduced proliferation in vitro and in xenograft mouse models in vivo. The Authors find that CpG island methylation is reduced upon DIPRO1 knockdown, specifically affecting pro-apoptotic genes, which according to them leads to the enhanced apoptosis seen upon loss of DIPRO1 function in tumor cells.

The manuscript is heavy on global genomic and transcriptomic analyses which is nice, providing a broader outlook to the findings. Having said that, very little data is provided by way of validation using specific targets, which would have enhanced the

manuscript considerably, providing conclusive proof and also making it simpler to comprehend. Switching back and forth between normal myogenesis, different types of sarcoma and later on to brain tumors takes away somewhat from the work, instead of focusing on any single aspect. However, this is an elaborate body of work, which has translational potential and can be considered for publication if the Authors address the following comments satisfactorily.

Major comments

1. Since the Authors claim that DIPRO1 and SIX1 share binding motifs, is it that the two proteins interact and co-occupy binding sites? This should be tested. The Authors should confirm their claims by testing out SIX1 target gene regulation using 2-3 known SIX1 target genes, in the presence of SIX1, DIPRO1 and both together.
2. The Authors mention that overexpression of DIPRO1 leads to increased proliferation and reduced differentiation in myoblasts, whereas depletion of DIPRO1 promotes differentiation. These results are difficult to appreciate without a clear picture of the endogenous expression dynamics of DIPRO1 during myogenic differentiation. Is it expressed in the muscle stem cells and what is its contribution to their normal differentiation?
3. The Authors should show endogenous DIPRO1 expression in rhabdomyosarcoma and Ewing's sarcoma tumor cells compared to other sarcoma cells. While the data from the DepMap portal is a good starting point, validation has to be done to make a convincing case. It is puzzling that knockdown of DIPRO1 causes differentiation in myoblasts and apoptosis in tumor cells. How do the Authors explain this? Differentiation markers should be tested in rhabdomyosarcoma and Ewing's sarcoma cells depleted for DIPRO1 and apoptotic markers in myoblasts depleted for DIPRO1.
4. What happens when DIPRO1 is overexpressed in rhabdomyosarcoma and Ewing's sarcoma tumor cells? Do the cells change morphology and become more proliferative?
5. How is the nanomedicine administered to the tumor bearing mice? Why is the tumor volume data only pursued until day 12 (Figure 4C)? Is it that the control tumors reach maximum permitted tumor volume by then? Markers of myogenic differentiation should be shown for control and DIPRO1 depleted tumors. Compared to the in vitro data, the effect on cell death in the in vivo tumor xenografts seems modest, making it unclear how the decrease in tumor volume occurs.
6. The data provided on a possible correlation between DIPRO1 and SOX2 in different tumor types such as rhabdomyosarcomas, Ewing's sarcoma, medulloblastoma and neuroblastoma are interesting, but preliminary and speculative. Without further validation, it is not clear how this is relevant to this manuscript.
7. The mechanistic basis of the role of DIPRO1 needs to be elucidated better. What are the functions of the identified DIPRO1 interactors TIF1B, UHRF1 and MCM6? What roles do these proteins play in myogenic differentiation and sarcoma tumors?
8. How is DIPRO1 regulating CpG island methylation? Does it interact with and recruit DNA methyl transferases? What regulates the specificity of hypomethylation upon DIPRO1 knockdown, such that CpG islands upstream of pro-apoptotic genes are specifically affected? What happens to CpG islands upstream of proliferation and differentiation related genes in tumor cells and myoblasts upon DIPRO1 knockdown?
9. The degree of overlap between SIX1 and DIPRO1 with respect to methylated regions and target genes are ~10%. Since knockdown of both genes might have similar phenotypic effects, sharing 10% target genes is not surprising and need not necessarily suggest shared binding and targets. Can the Authors comment on this?

Minor comments

10. Why is there no panel B in Figure 1?
11. What is the "other" category which accounts for more than 46% of DiPRO1 ChIP peaks in Figure 1C? How does the genomic distribution of ChIP peaks compare to that of another (control) zinc finger gene?
12. How does the repeat DNA distribution in DiPRO1 ChIP peak regions in Figure 1D compare to genome wide repeat DNA distribution? In other words, is there any bias in repeat DNA distribution with respect to DiPRO1 ChIP peak regions compared to the abundance of repeat DNA distribution in the entire genome?
13. How were the chromatin states identified in Figure 1E?
14. Why are only about 100 targets shared between DiPRO1 and SIX1, if their consensus binding motifs are similar as shown in Figure 1F? Why is the Six1 mouse motif shown with respect to DiPRO1 in Figure 1F?
15. How does the SIX1 single cell type specificity compare to that of DiPRO1 shown in Figure 1G?
16. In Figure 2A-G, what are the controls? Are they also transduced with the empty retroviral vector? If not, it is important to do that to ensure identical treatment of the control cells.
17. In Figure 2I, how are the length and width measurements made, since the cultured myofibers are not linear? This does not seem to be a reliable measure; a more standard approach would be to measure the sarcomere length and width in the myofibers.
18. In Figure 3B, why does control shRNA treatment result in about 50% of cell death in TE671 cells?
19. The cell death seen in Figure 3C seems strange, since no live cells are visible upon DiPRO1 shRNA treatment; is this representative?
20. In Figure 6G, what genes are tested?

We appreciated the constructive criticism of the Referees. We have addressed each of their concerns as outlined below.

Consistent with the referees' main recommendations, substantial improvements have been made to the manuscript. Three other scientists, including Pr Pascal MAIRE, an expert in the field of SIX genes and myogenesis, actively contributed to the experimental and advisory aspects. Methylation sequencing experiments were carried out after DiPRO1 knockout in myoblast cells to delineate differential effects between normal and cancerous muscle cells. Additionally, we included EMSA and Luciferase reporter experiments and single-cell RNA (scRNA-seq) sequencing analysis of 6 PDX primary cell cultures of rhabdomyosarcoma. That enabled us to (i) validate the *in silico* prediction of common targets for SIX1 and DiPRO1 and (ii) offer mechanistic insights into DiPRO1's role in inflammation activation in muscle cancer cells. We mainly focused on the crosstalk between SIX1 and DiPRO1, supported by various experimental data, and consequently removed the suggested link between SOX2 and DiPRO1 from the Results and Discussion.

However, due to time, cost, and resource constraints, we were unable to perform the labor-intensive experiments involving additional PDX models, patient explants, or humanized mouse models. The primary aim of this work was to establish a proof of concept. To adhere to ethical considerations related to animal welfare (3R principles), we minimized animal experiments by employing *in vitro* models as substitutes. To enhance the robustness of our findings, we integrated the scRNA-seq analysis of six patient-derived xenograft tumors.

The detailed answers (**A**) to the referees' comments (**R**) are addressed below.

Referee #1 (Remarks for Author):

In this manuscript, Rich et al. present an interesting set of data regarding the role of DIPRO1 in both myogenesis and mesenchymal cancers. Using different approaches, their data support the view that DIPRO1 may be involved in regulation of cell state by modifying the expression pattern of genes via epigenetic modifications. Although their results and proposed therapeutic approach are innovative, the accumulation of data from different approaches and models may lead to confounding effects that could result in misinterpretations of the role of DIPRO1 in sarcoma tumorigenesis. Revisions are then needed to clarify these points.

R1. *Fig1-2: Chip-seq analyses have to be performed in myoblasts to strengthen the correlation with DE genes in myoblasts overexpressing or silenced for DIPRO1, and the subsequent identification of DIPRO1 specific targets.*

A1: ChIP-seq experiments proved challenging due to antibody compatibility issues. However, to enhance the correlation with differentially expressed genes (DE genes) in myoblasts with silencing of DiPRO1, and subsequently identify specific epigenetic targets of DiPRO1, we conducted additional MIRA-seq experiments (methylation CpG island recovery assay) in myoblasts. We juxtaposed this data with results obtained from muscle cancer RMS cells, providing a more unique perspective for this transcription factor. Additionally, we validated certain direct targets of DiPRO1 through EMSA and Luciferase reporter assays.

R2. *Fig3: The rhabdomyosarcoma lines used are not the most consensual. The results obtained deserve to be validated in other lines such as RD (ERMS) or RH30 (ARMS) for example. which ones did we use ? there are indeed a multitude of different RMS cell lines*

A2: As detailed in the Materials section, we employed two rhabdomyosarcoma (RMS) cell lines of different subtypes: ARMS JR cells (COG, RRID: CVCL_RT33) and ERMS TE671 (RRID: CVCL_1756), along with two Ewing sarcoma cell lines A673 (RRID: CVCL_0080) and EW7 (RRID: CVCL_1217) cells. The TE671/RD cell line is genetically identical to the human rhabdomyosarcoma cell line RD (doi: 10.3389/fonc.2013.00183. Upon thorough verification through STR genotyping (Mycrosynth, Switzerland), it was confirmed to be JR ARMS), initially thought to be RH30. Both RMS cell lines have been employed in preclinical studies according to published literature. Additionally, the impact of DiPRO1 knockdown was corroborated in Ewing sarcoma cell lines, and the findings were consistent with those observed in RMS, affirming the reproducibility of the effect.

R3. *Fig3: Authors claim that DIPRO1 KD induces RMS cell apoptosis, based mostly on changes in the expression pattern of some effectors of the pathway. However, since these arguments are only correlative, they should instead define the impact of down regulation of DIPRO1 on caspase-3 activity, or show how ZVAD can block this cell death. you have no functional assays done, correct?*

A3: The results of in vitro expression analyses were corroborated by analysis of *ex vivo* tumour sections after inhibition by shDiPRO (Fig. 4G-H). In particular, we observed a significant increase in immunostaining for cleaved caspase-3, detected by antibody directed against endogenous levels of the large fragment (17/19 kDa) of activated caspase-3 resulting from cleavage adjacent to Asp175. This increase was correlated with an increase in TUNEL staining. This analysis was carried out by a qualified pathologist from the Pathology and Cytology Unit at Gustave Roussy cancer campus. Furthermore, a new comparative functional analysis of DiPRO1 knockdown in cancer and normal muscle cells, utilizing Ingenuity scores, has been conducted. The findings reveal a reduction in the senescence pathway and the activation of apoptotic pathways, specifically in cancer cells, aligning with the cell death quantified by FACS.

R4. *Fig3: Changes in apoptotic genes correspond to gain of both anti and pro-apoptotic genes. Progeny or Ingenuity scores should be applied to infer score of activation of the pathway.*

A4: We appreciate this valuable feedback, which has been taken into account. A further analysis of Canonical pathways using Ingenuity scores has been carried out, and the results have been integrated into the revised version (Fig. 3F).

R5. *Fig 2-3: What is the overlap in DE genes in myoblasts and cancer cells models? This could improve the identification of DIPRO1 specific targets, since the number of DE genes is huge and could lead to confounding effects.*

A5: There is a 20.5% overlap in upregulated DEGs and a 14.8% overlap in downregulated DEGs between mesenchymal cancer cells and myoblasts in response to DiPRO1 knockdown, suggesting a distinct whole transcriptome signature in a cell-state-dependent manner. We added the Fig. EV1K to the revised version. The Ingenuity Canonical Pathway analysis was performed to score the pathways that differentially enriched, reflecting the distinct effect of DiPRO1 depletion in non-malignant and malignant cells (Fig. 3F). This aligns with our earlier analysis using Cytoscape ClueGo and Hallmarks Molecular Signatures Database (MSigDB), as well as Gene Set Enrichment Analysis (GSEA) software.

R6. *Fig 4: What is the relevance of this therapeutic approach in an RMS model ?*

A6: Based on the RNA sequencing data from relapsed tumors in the patient cohort, a conspicuous upregulation of DiPRO1 is evident, suggesting its potential as both a prognostic marker and a therapeutic target. Furthermore, the single-cell RNA sequencing (scRNA-seq) data from patient-derived xenograft (PDX) tumors reveals enhanced DiPRO1 expression, particularly in cycling and muscle/mesenchymal-like progenitors. Notably, literature recommendations underscore the significance of targeting these cell populations for therapeutic development. It is noteworthy that, adhering to the 3R reduction principles, we opted to utilize one mouse model. We confirmed the effects of DiPRO1 knockdown across various *in vitro* mesenchymal cancer cell lines and validated these findings in a highly aggressive mesenchymal cancer animal model (growth rate within a two-week period).

R7. *Fig 5: The rationale of exploring MappyActs pan-cancers relapse cohort rather than publicly available transcriptomic data on RMS and EWS is not clear. What is the expression level of DIPRO1 in RMS compared to normal muscles ? What is the prognostic value of this gene in EWS/RMS cohorts that have already been published ? Is there a correlation between DIPRO1 and some of the target genes presented here ? Along the same line, what is the expression profile of DIPRO1 in the recently published single-cell analyses of RMS samples/models ? Is its expression restricted to mesenchymal-like progenitors cluster ?*

Assume you could do the expression also over other public cohorts ?

A7: The expression data were analyzed using MAPPYACTS as the initial step. This dataset is a pan-cancer cohort specifically consisting of recurrent or refractory tumors, representing a population with the highest medical need for new treatment strategies. Moreover, it is our own dataset with facilitated access and assured annotations. Certainly, additional databases should be explored, along with investigating the prognostic value of DiPRO1 and its target genes. The further exploration of this gene should also be carried out using recent data from single-cell analyses of RMS, which could be a next step in our preclinical research.

In response to this remark, the analysis of single-cell RNA sequencing (scRNA-seq) data from PDX tumors (new Fig. 5A-C, new Fig. EV3A-C) has been integrated. Our results demonstrated that, in addition to cycling cell subpopulations, DiPRO1 expression exhibited enrichment in mesenchymal/muscle progenitors and differentiated tumor cells. Previous studies have associated cycling cells with a worse patient prognosis, and newly discovered insights suggest that mesenchymal muscle-like stem cells play a role in driving relapse, opening avenues for therapeutic targeting. This implies that DiPRO1 could potentially serve as a biomarker to predict the prognosis of RMS. Consequently, it was rational to examine DiPRO1 expression specifically in relapsed tumors that did not exhibit a complete response to therapy.

In response to the referee's request, additional results have been included demonstrating that the expression level of endogenous DiPRO1 in RMS cells is comparable to that in proliferating myoblasts and higher than that observed in differentiating myotubes. Similar observations were made when comparing cell populations within primary PDX RMS tumors. Differentiated MYOG-expressing cells exhibited lower expression levels of DiPRO1 compared to mesenchymal/muscle progenitors (new Fig. 5A-B, new Fig. EV3C).

R8. *Fig 5: The robustness and relevance of exploring the so-called downstream genes network is not that clear/convincing. Intersection between CHIPseq and RNAseq analyses performed in all overexpressing/downregulated models should be used to define this network and to strengthen*

the conclusions.

A8: As mentioned in answer A1, additional experiments (MIRA-seq in myoblasts) and an intersectional analysis between cell-specific methylation and transcriptomic signatures have been carried out. This allowed us to delineate the disparate functions of DiPRO1 in muscle cells under normal and malignant state. Furthermore, we validated specific direct targets of DiPRO1 using EMSA and Luciferase reporter assays. Our investigation also confirmed the direct binding of DiPRO1 to noteworthy targets, including those shared with the SIX1 myogenic regulator.

R9. *The link between SOX2/SIX1 and DIPRO1 across models is at least confusing, and perhaps even over interpreted. Authors should provide more evidences that there is a synergistic action of DIPRO1 and SOX2 or SIX1 on target genes, resulting respectively from epigenetic and transcriptional regulation, by showing for example that impact of DIPRO1 overexpression on the expression of these genes is blocked by downregulation of SIX1 or SOX2.*

A9: We acknowledge the suggestion and have opted to concentrate on the interplay between SIX1 and DiPRO1, substantiated by supplementary experimental data in the revised version. As a result, the proposed association between SOX2 and DiPRO1 has been omitted from the Results and Discussion. The distinct expression of SIX1 and DiPRO1 in individual cancer cells has been additionally demonstrated via scRNA-seq analysis.

The earlier prediction has been confirmed in muscle cells via a Luciferase reporter assay, demonstrating the dose-dependent effect of DiPRO1 and SIX1 gene co-regulation. The findings have been integrated into the revised version (Fig. 8 D-H).

R10. *The fact that only 14.6% of DMCs correlate with changes in genes expression in RMS cells seems quite low. The authors should comment on this and perhaps provide some statistical insights. Same comment for the 12.4% of promoter regions bound by SIX1 and DMCs in DIPRO1 KD.*

A10. Stringent conditions were chosen, with a focus on the conventional correlation of CGI methylation associated with gene repression, while the possibility of methylation versus gene upregulation, which might also occur, was not considered. This omission could explain the low number of differentially expressed genes (DMG). Concerning the SIX1 and DiPRO1 DMCs, 12.4% are aligned with the approximately 16% overlap of SIX1 and DiPRO1 direct targets. Similar observations (10% overlap of direct targets) have been reported for ZNF516 and CtBP1, as demonstrated by Lifang Li et al. in Nature Communications (2017), illustrating the ZNF516 targeting complex with CtBP1 and their shared targets. These comments have been included in the revised version.

Minor points

R1. *Authors report their previous finding upon the role of DIPRO1 on ANT1 regulation. Regarding the pro-apoptotic role of ANT1, and its previously established role in RMS, how do the authors reconcile their finding that DIPRO1 down regulation induces apoptosis?*

A1: Our previously published results, as demonstrated by Luciferase reporter assays, revealed that downregulation of ANT1 occurred with the depletion of DiPRO1 in the presence of the 4qA enhancer, juxtaposed to the ANT1 promoter in FSHD dystrophy. Notably, there was no observable effect of DiPRO1 depletion on the ANT1 promoter alone. It was suggested that the regulation of the ANT1 gene by DiPRO1 might occur through cis-regulatory elements. Simultaneously, evidence supporting the transcriptional repressor role of DiPRO1 is presented in the current manuscript. However, specific enhancer/repressor elements in close proximity to

ANT1 in RMS cells were not investigated. Furthermore, this particular gene did not exhibit expression changes in shDiPRO1 RMS. In accordance with our findings, DiPRO1 demonstrated both activator and repressor activities. Thus, the up-regulation of certain pro-apoptotic genes by DiPRO1 is not ruled out, as depicted in Fig. 3G.

R2. *Some English/typo misspellings.*

A2: checked

Referee #2 (Remarks for Author):

The authors show evidence that DiPRO1 is a transcriptional and epigenetic regulator that compromises normal and malignant cell fate in a selective manner supported by analysis of publicly available patient datasets and some functional data generated. The authors suggest that DiPRO1 is a positive regulator of proliferation and a negative regulator of differentiation of human myoblasts whereas it is a positive regulator of proliferation and prevents apoptosis in cancer cells. Importantly from a translational point of view and potential therapeutic strategy, the authors show that DiPRO1 inhibition is toxic to RMS and Ewing sarcoma cancer cells, but not normal. The therapeutic concept is applicable to paediatric cancers of mesenchymal origin which has impact on high unmet clinical needs and is novel in terms of target (DiPRO1) but also providing some *in vivo* Poc in terms of delivery.

Points for revision:

R1. *Animal mouse studies not suitable (no orthologues) - and therefore model used is immunodeficient, any alternatives considered? (patient explants? organ/tumour on a chip? humanised mouse model?)*

would be very time consuming, organoids a la mode but even less close than PDX

A1: Due to time, cost, and resource constraints, we were unable to perform the labor-intensive experiments involving additional PDX models, patient explants, or humanized mouse models. The primary aim of this work was to establish a proof of concept. To adhere to ethical considerations related to animal welfare (3R principles), the minimization of animal experiments was prioritized by employing *in vitro* models as substitutes. The effects of DiPRO1 knockdown were confirmed across various *in vitro* mesenchymal cancer cell lines, and these findings were validated in a highly aggressive mesenchymal cancer animal model (growth rate within a two-week period). To enhance the robustness of our results, scRNA-seq analysis data from six patient-derived xenograft tumors have been integrated. This analysis revealed an enrichment of DiPRO1 expression in cycling cells and mesenchymal/muscle progenitors, the latter being implicated in driving relapse. This finding suggests targeting DiPRO1 could be a potential therapeutic strategy.

R2. *Overexpression of DiPRO1 in vitro: is the recombinant protein overexpressed maintaining native conformation and cellular localisation (nuclear)?*

A2: A specific study on this matter was not performed. According to literature (B. Gasset, et al, Microb Cell Fact, 2008), one of the main challenges in recombinant protein production is the difficulty of foreign polypeptides to attain their native conformation in heterologous host cells. In our experiments, human DiPRO1 protein was expressed in human host cells. Based on predictions and consistent with the cellular compartment expression patterns of a large cohort of

Zinc finger transcription factors, DiPRO1 is primarily expressed in the nucleus. Misfolding can lead to protein degradation. Employing multiple approaches, we are confident in the preservation of the native structure of DiPRO1. The EMSA assay with DiPRO1 protein extract under native gel conditions demonstrated selective binding of DiPRO1 to its targets, and the observed protein size by Immunoblotting corresponded to the predicted size.

R3. *Depletion sh vs OE: would it be possible to generate mutants (to disrupt interaction with binding partners extensively referred to across the manuscript) and perform rescue experiments?*

A3: That's an intriguing aspect. We attempted to induce the expression of a mutant in the Zinc-finger domain, which interacts with DNA, but unfortunately, we were unable to obtain the expressed protein. This failure could be attributed to misfolding and subsequent degradation, as pointed out in your comment R2. In terms of the implications suggested in this manuscript, inhibiting DiPRO1, in our perspective, could be achieved more efficiently by using small biological RNAs as nanomedicines. However, for fundamental research aimed at more comprehensively characterizing the protein and its domains, exploring this aspect could be valuable for future investigations.

R4. *Comparison with fibroblasts (could DiPRO1 have a role in myofibroblast differentiation that occurs in a tumour microenvironment giving rise to cancer associated fibroblasts? (commonly generated in vitro by treatment with TGFb)*

A4: In our previous publication (Kim L, et al, NAR 2015), we demonstrated that DiPRO1 expression was notably lower in fibroblasts compared to myoblasts. The scRNA-seq analysis of PDX-derived tumors revealed the highest enrichment of DiPRO1 gene expression in mesenchymal/muscle progenitors. The up-regulation of fibroblast markers such as DCN, COL1A1, PDGFRA was not observed in RMS cells following DiPRO1 knockdown. Consequently, it is challenging for us to envision a significant impact of DiPRO1 in this cell type. However, we cannot rule out this possibility as we did not investigate fibroblast differentiation.

R5. *in vitro functional experiments don't address/match the suggested caspase activation for cell death induction in cancer cells (to mirror/correlate with the IHC for in vivo PoC), staining (IF or even flow cytometry to complement cell cycle analysis performed) would address this.*

A5: The results of *in vitro* expression analyses were corroborated by analysis of *ex vivo* tumour sections after inhibition by shDiPRO (Fig. 4G-H). In particular, we observed a significant increase in immunostaining for cleaved caspase-3, detected by antibody directed against endogenous levels of the large fragment (17/19 kDa) of activated caspase-3 resulting from cleavage adjacent to Asp175. This increase was correlated with an increase in TUNEL staining. This analysis was carried out by a qualified pathologist from the Pathology and Cytology Unit at Gustave Roussy cancer campus. Furthermore, a new comparative functional analysis of DiPRO1 knockdown in cancer and normal muscle cells, utilizing Ingenuity scores, has been conducted. The findings reveal a reduction in the senescence pathway and the activation of apoptotic pathways, specifically in cancer cells, aligning with the cell death and cell cycle analysis quantified by FACS (Fig. 3B and Fig. EV1J).

R6. *In vivo PoC: what's the KD efficiency in si vs sh? Is it correlating with higher benefit observed with sh? (transient vs stable kd?) can IHC include DiPRO1 staining? (if not possible due to lack of Ab, RNAscope? Spatial transcriptomics?)*

In Fig. 4A, the *in vivo* and *ex vivo* scanning of Cy5-labelled siDiPRO1 demonstrated the internalization of nanomedicines in the tumors. The inhibition of the DiPRO1 gene was verified using mRNA extracted from excised tumors, showing approximately 45% and 55% inhibition of DiPRO1 gene expression (Fig. 4B). This resulted in the inhibition of tumor growth or total tumor cell death. A more drastic effect *in vitro* was observed, resulting in the complete death of cancer cells, which correlated with an 80% inhibition of DiPRO1. The effects of DiPRO1 knockdown were reproducible *in vitro* across various mesenchymal cancer cell lines, and these findings were validated in a highly aggressive mesenchymal cancer animal model (with the maximum growth rate detected within a two-week period). Therefore, we are confident about the observed effects *in vitro* and *in vivo*.

RNA-scope is a time-consuming and costly methodology that has so far remained semi-quantitative. scRNA-seq is based on gene transcript "clouds" that detect only 10% of DiPRO1 gene expression (the highest level) (Fig. EV3C), and is not yet optimized for weakly expressed transcription factors. Spatial transcriptomics has an even lower detection sensitivity than scRNA-seq and will hardly detect DiPRO1 inhibition. Consequently, real-time qPCR, the most suitable method for our experiments, was applied to quantify DiPRO1 inhibition.

R7. *As a theory/role in repression/de-repression of endogenous retrovirus, would be useful to explore a link with STING pathway (sensing of nucleic acid) which triggers an innate immune response via IFN I signalling, currently being explored within cancer immunotherapies strategies, experimentally testable looking at STING activation markers/IFN responsive genes (ISGs)*

A7: We agree with the referee's suggestion. Indeed, differential analysis via Ingenuity Canonical Pathways has demonstrated activation of IFN1/2 signaling by DiPRO1 knockdown in mesenchymal cancer cells and myoblasts (new Fig. 3F).

R8. *typo in " Animals with established s.c. tumors were randomly assigned to treatment groups (n=7 or 5) receiving either nanocomposites of (i) shDiPRO1/jetPEI®, (ii) shDiPRO1/jetPEI® or (iii) scramble siRNA controls (Ctl)" where sh 2x instead of siDiPRO1?*

A8: Thank you. The necessary corrections have been implemented.

R9. *In fig2A/B: actin as loading control for WB shows differences between ctrl and DiPRO1 - is the graph taking into account normalisation to actin as well (any other loading control most suitable or total protein normalisation more accurate?)*

A9: Figure 2A depicts the mRNA expression of exogenous DiPRO1, while Figure 2B illustrates the protein expression of exogenous DiPRO1. DiPRO1 expression is clearly observable even in the smaller protein extract.

Referee #3 (Remarks for Author):

The manuscript by Rich et al characterizes the human ZNF555 (DIPRO1) gene in myoblast differentiation and two mesenchymal sarcomas rhabdomyosarcoma and Ewing's sarcoma. Previously, the same group identified DIPRO1 as a transcriptional regulator of the Adenine Nucleotide Translocator 1 (ANT1) gene in facio-scapulohumeral muscular dystrophy. In this manuscript, the Authors initially perform analyses to shed light on how DIPRO1 is important in

myogenesis using gain- and loss- of function approaches. They find that DIPRO1 shares consensus binding motif with SIX1, an upstream myogenic regulator. Extending their study, the Authors characterize DIPRO1 in rhabdomyosarcoma and Ewing's sarcoma, two mesenchymal tumors where DIPRO1 levels are high. Knockdown of DIPRO1 in these tumors lead to increased cell death and reduced proliferation in vitro and in xenograft mouse models in vivo. The Authors find that CpG island methylation is reduced upon DIPRO1 knockdown, specifically affecting pro-apoptotic genes, which according to them leads to the enhanced apoptosis seen upon loss of DIPRO1 function in tumor cells.

The manuscript is heavy on global genomic and transcriptomic analyses which is nice, providing a broader outlook to the findings. Having said that, very little data is provided by way of validation using specific targets, which would have enhanced the manuscript considerably, providing conclusive proof and also making it simpler to comprehend. Switching back and forth between normal myogenesis, different types of sarcoma and later on to brain tumors takes away somewhat from the work, instead of focusing on any single aspect. However, this is an elaborate body of work, which has translational potential and can be considered for publication if the Authors address the following comments satisfactorily.

Major comments

R1. *Since the Authors claim that DIPRO1 and SIX1 share binding motifs, is it that the two proteins interact and co-occupy binding sites? This should be tested. The Authors should confirm their claims by testing out SIX1 target gene regulation using 2-3 known SIX1 target genes, in the presence of SIX1, DIPRO1 and both together.*

A1: We appreciate the suggestion. Consequently, we performed additional experiments and included Figures 8D-F, demonstrating DiPRO1 binding to SIX1 targets through EMSA. Additionally, we conducted Luciferase reporter assays, revealing the dose-dependent effect of DiPRO1 and SIX1 co-expression, further supporting our hypotheses.

R2. *The Authors mention that overexpression of DIPRO1 leads to increased proliferation and reduced differentiation in myoblasts, whereas depletion of DIPRO1 promotes differentiation. These results are difficult to appreciate without a clear picture of the endogenous expression dynamics of DIPRO1 during myogenic differentiation. Is it expressed in the muscle stem cells and what is its contribution to their normal differentiation?*

A2: We acknowledge the referee's feedback and have included an analysis of endogenous DiPRO1 expression in proliferative and differentiated myoblasts and RMS cells (new Fig. 2M). Additionally, scRNA-seq data in RMS cells reveals higher expression in proliferative cells compared to differentiated cells expressing MYOG (Fig. 5B). This aligns with our DiPRO1 overexpression/downregulation models.

R3. *The Authors should show endogenous DIPRO1 expression in rhabdomyosarcoma and Ewing's sarcoma tumor cells compared to other sarcoma cells. While the data from the DepMap portal is a good starting point, validation has to be done to make a convincing case. It is puzzling that knockdown of DIPRO1 causes differentiation in myoblasts and apoptosis in tumor cells. How do the Authors explain this? Differentiation markers should be tested in rhabdomyosarcoma and Ewing's sarcoma cells depleted for DIPRO1 and apoptotic markers in myoblasts depleted for DIPRO1.*

A3: The comparison of myogenic markers between myoblasts, rhabdomyosarcoma, and Ewing's sarcoma cells was demonstrated in Fig. 8C. To strengthen our findings on apoptosis enrichment

events in DiPRO1-depleted RMS cells, a comparative analysis of apoptotic pathways was performed using Ingenuity Canonical Pathway analysis. This analysis considered the impact of DiPRO1 knockdown in mesenchymal cancer cells versus myoblasts (new Fig. 3F-G).

R4. *What happens when DIPRO1 is overexpressed in rhabdomyosarcoma and Ewing's sarcoma tumor cells? Do the cells change morphology and become more proliferative?*

A4: The overexpression of DiPRO1 led to an enhanced proliferation rate of RMS cells without apparent morphological changes. Supplementary Fig. EV1L has been added to the revised version. Although transcriptomic data were available (E-MTAB-10822, EMBL-EBI database), we preferred to highlight the beneficial effects of DiPRO1 knockdown. While these data are not presented in the article due to limitations in figure number and text volume, we have included a comment in the text.

R5. *How is the nanomedicine administered to the tumor bearing mice? Why is the tumor volume data only pursued until day 12 (Figure 4C)? Is it that the control tumors reach maximum permitted tumor volume by then? Markers of myogenic differentiation should be shown for control and DIPRO1 depleted tumors. Compared to the in vitro data, The effect on cell death in the in vivo tumor xenografts seems modest, making it unclear how the decrease in tumor volume occurs. was this experiment with the ulcerations ?*

A5: The nanocomposites were administered intratumorally in glucose solution at 0.5 or 1.0 mg/kg/injection twice or three-times per week. The information on the intratumoral administration has been added to the Fig. 4 legend. Two experiments (1 and 2) were performed until the control tumors reach maximum permitted tumor volume. In the 3rd experiments visible ulcerations were observed independent of treatment, therefore all experimental animals were sacrificed and data only pursued until day 12. The A673 Ewing cell line results in very rapid growing xenograft tumors as illustrated by the tumor doubling time of 1.8 and 1.5 days respectively in the two different experiments EW2 and EW3 in Table 4E. The antitumor activity encountered was indeed mainly a tumor growth delay, which was however significant as compared to control tumors and which at least in part should be due to the very rapid tumor growth of the xenograft model. Indeed, several of rapidly growing xenograft models develop necrotic features that seem to be related to the cell type and probably also their capacity to induce spontaneous apoptotic or necrotic cell death. The A673 Ewing cells do not express as many myogenic markers (Fig. 8C). Given that this tumor is not myogenic but mesenchymal in origin, we believed that focusing mainly on proliferation and apoptosis markers would provide more valuable information in these experiments.

R6. *The data provided on a possible correlation between DIPRO1 and SOX2 in different tumor types such as rhabdomyosarcomas, Ewing's sarcoma, medulloblastoma and neuroblastoma are interesting, but preliminary and speculative. Without further validation, it is not clear how this is relevant to this manuscript. would need validation indeed*

A6: We agree with this suggestion and focused primarily on the interaction between SIX1 and DiPRO1, supported by additional experimental data in the revised version (Fig. 8D-F). Consequently, we have omitted the proposed association between SOX2 and DiPRO1 in the results and discussion.

R7. *The mechanistic basis of the role of DIPRO1 needs to be elucidated better. What are the functions of the identified DIPRO1 interactors TIF1B, UHRF1 and MCM6? What roles do these proteins play in myogenic differentiation and sarcoma tumors?*

A7: We sincerely appreciate the valuable remark provided and have integrated it into the discussion section.

Our findings indicate that DiPRO1 directly interacts with TIF1B, UHRF1, and MCM complex proteins. The KAP1/TIF1B protein is present alongside MyoD and Mef2 at numerous muscle genes, where it acts as a scaffold for recruiting coactivators and corepressors (Singh, Cassano et al. 2015). UHRF1 is activated in satellite cells and regulates their proliferation and differentiation (Sakai, Sawada et al. 2022). In myoblast cells, the MCM complex is involved in rendering chromatin operational for DNA replication, a crucial function for myoblasts emerging from quiescence and undergoing proliferation. MyoD directly targets the promoters of MCM complex members 2 and 7, inducing transcription from MCM (Zhang, Sha et al. 2010). Upregulation of MCM genes has also been demonstrated to be triggered by MEF2A, inducing myoblast proliferation and inhibiting myogenic differentiation (Wang, Yang et al. 2018).

The expression of UHRF1 increases in mesenchymal sarcoma tumors compared to non-sarcomas (Liu, Yu et al. 2020), promoting cell invasion and correlating with recurrence and overall survival (Sannino, Marchetto et al. 2017). TRIM28/TIF1B gene expression exhibits strong positive associations with sarcoma tumors (Shang, Wu et al. 2023) and serves as a prognostic predictor of immunotherapy resistance, regulating stemness, proliferation, and migration of cancer cells (Yang, Tan et al. 2023). TIF1B plays a crucial role in herpes virus-associated tumors, including EBV, Kaposi's sarcoma-associated herpesvirus, and human cytomegalovirus (Randolph, Hyder et al. 2022). The MCM core hexamer complex (MCM2-7) plays an essential role in DNA replication and cell cycle progression, being ubiquitously expressed in proliferating normal and cancer cells (Maiorano, Lutzmann et al. 2006). According to our findings, MCM 2/3/4/6 members interact with the DiPRO1 protein. These MCM members have been shown to be highly expressed in sarcoma tumors, proposing them as potential biomarkers for the survival prognosis of sarcoma patients (Zhou, Wang et al. 2021).

R8. *How is DIPRO1 regulating CpG island methylation? Does it interact with and recruit DNA methyl transferases? What regulates the specificity of hypomethylation upon DIPRO1 knockdown, such that CpG islands upstream of pro-apoptotic genes are specifically affected? What happens to CpG islands upstream of proliferation and differentiation related genes in tumor cells and myoblasts upon DIPRO1 knockdown?*

A8: First of all, we highly appreciate the valuable comment from the referee, finding it immensely valuable for deepening the comparative study and understanding the distinct effects of DiPRO1 knockdown. Supplementary experiments have been performed to investigate CGI methylation in control and DiPRO KD myoblasts, aligning differentially methylated CGIs and DEGs in normal and cancer cells. Consequently, the corresponding chapter was revised and titled "Distinct DiPRO1-mediated CGI methylation regulation in cancer and normal muscle cells." Our results demonstrate that DiPRO1 predominantly influences the methylation of the inflammation pathway in RMS and myogenesis in myoblasts, highlighting a unique pattern between cancer and normal muscle cells.

It has been proposed that KRAB-ZFPs selectively bind to cognate cis-regulatory elements and the corepressor TIF2B/KAP-1 to targeted loci (Schultz, Ayyanathan et al. 2002). TIF2B functions as a scaffold coordinating specific gene silencing, targeting specific promoters via the KZNP UHRF1 is a 5mC reader and plays a role in the control of cell-type-specific transcripts

through the maintenance of DNA methylation (Bostick, Kim et al. 2007). UHRF2, a paralog of UHRF1, directly binds to 5hmC. ZNF618 binds DNA adjacent to 5hmC and facilitates UHRF2 deposition, thus regulating UHRF2 chromatin localization (Liu, Zhang et al. 2016). Both UHRF2 and TIF2B can recruit DNMT1 (Liu, Gao et al. 2013, Cheng, Kuo et al. 2014). In our experiments, DiPRO1 depletion led to CGI demethylation in specific regions, suggesting that DiPRO1 could recruit TIF2B and contribute to UHRF1 5mC binding in DiPRO1-targeted loci. This in turn could lead to DNMT1 loading and chromatin silencing. Importantly, the interaction between KAP1 and TIF2B has been reported in several studies (Kim, Tanaka et al. 2018, Verdikt, Bendoumou et al. 2022), supporting our hypothesis.

R9. *The degree of overlap between SIX1 and DiPRO1 with respect to methylated regions and target genes are ~10%. Since knockdown of both genes might have similar phenotypic effects, sharing 10% target genes is not surprising and need not necessarily suggest shared binding and targets. Can the Authors comment on this?*

A9: Concerning the DMCs associated with SIX1 and DiPRO1, approximately 12.4% correlate with the observed overlap of approximately 16% between the direct targets of SIX1 and DiPRO1. Similar observations of a 10% overlap in direct targets have been documented for ZNF516 and CtBP1, as evidenced by Lifang Li et al. in Nature Communications (2017), illustrating the ZNF516 targeting complex with CtBP1. We have included these comments into the revised version. Moreover, we demonstrated the binding of DiPRO1 to multiple MEF3 motifs, which are SIX1 binding sites present in the promoters of myogenic genes. This was achieved through EMSA and Luciferase reporter assays, confirming our earlier findings.

Minor comments

R10. *Why is there no panel B in Figure 1?*

A10: We regret the labeling confusion, and the panels of Figure 1 have been adjusted accordingly.

R11. *What is the "other" category which accounts for more than 46% of DiPRO1 ChIP peaks in Figure 1C? How does the genomic distribution of ChIP peaks compare to that of another (control) zinc finger gene?*

A11: We appreciate the reviewer's observation. The category name has been updated, and "other" regions now correspond to "InterGenic" regions.

R12. *How does the repeat DNA distribution in DiPRO1 ChIP peak regions in Figure 1D compare to genome wide repeat DNA distribution? In other words, is there any bias in repeat DNA distribution with respect to DiPRO1 ChIP peak regions compared to the abundance of repeat DNA distribution in the entire genome?*

A12: We appreciate the reviewer's comment. A percentage analysis of repeat classes in the genome has been performed. This enabled us to demonstrate the enrichment of DiPRO1 binding to satellite and retroposon repeats (SVAs) relative to portion of distribution of these repeats in the human genome. This analysis has been added to Figure 1C.

R13. *How were the chromatin states identified in Figure 1E?*

A13: The ChIP seq regions were annotated using the ENCODE BroadChromHMM track. This was described in the Methods section "ChIP-seq"

R14. *Why are only about 100 targets shared between DiPRO1 and SIX1, if their consensus binding motifs are similar as shown in Figure 1F? Why is the Six1 mouse motif shown with respect to DiPRO1 in Figure 1F?*

A14: By crossing DiPRO1- and SIX1-ChIP seq data linked to the motifs shown in Figure 1E (previously Fig. 1F), we identified 98 common targets. This information was detailed in the legend for Dataset EV1. We have also added this information into the Methods section. The mouse Six1 motif has been omitted, and the MEF3 motif, a part of the SIX1 motif, has been added (Fig. 1E)

R15. *How does the SIX1 single cell type specificity compare to that of DiPRO1 shown in Figure 1G?*

A15: In the revised version, analysis of scRNA-seq data in RMS PDX tumors has been performed (Fig. 8G-H and Fig. EV3C). We compared the expression of DiPRO1 and SIX1. Both genes were enriched in cycling (KI67 marker) and differentiated (MYOG marker) tumor cells, without co-expression.

R16. *In Figure 2A-G, what are the controls? Are they also transduced with the empty retroviral vector? If not, it is important to do that to ensure identical treatment of the control cells.*

A16: As described in the Methods section, the pOZ_FHHY vector was used as a control

R17. *In Figure 2I, how are the length and width measurements made, since the cultured myofibers are not linear? This does not seem to be a reliable measure; a more standard approach would be to measure the sarcomere length and width in the myofibers.*

A17: Measurements were made using the ImageJ image analysis program, suitable for non-linear cell length measurements.

R18. *In Figure 3B, why does control shRNA treatment result in about 50% of cell death in TE671 cells?*

A18: Typically, during FACS analysis, the medium containing dead cells should be replaced to add trypsin for transferring adherent cells for analysis. However, in these experiments, we combined floating dead cells with adherent living cells before FACS measurement. The trypsin treatment might have contributed to cell death as well. Despite this, we are confident in the FACS results, considering a significant difference in cell death between shCtl and shDiPRO1 treated cells.

R19. *The cell death seen in Figure 3C seems strange, since no live cells are visible upon DiPRO1 shRNA treatment; is this representative?*

A19: These experiments were carefully performed several times in various cancer cell lines, using at least three distinct shRNA sequences targeting DiPRO1. The results presented are representative of the consistency of the results obtained. Other cancer cell lines of various origins were also tested (data not shown). The observed effect on cell death was considerably less pronounced in several cell lines, or even absent, compared to RMS and Ewing sarcomas.

R20. *In Figure 6G, what genes are tested?*

A20: The trans-acting activity of the DiPRO1 protein was tested. pDiPRO1 was co-transfected with a luciferase reporter construct containing the enhancer and promoter sequences. The activity of luciferase gene was measured. Additionally, in the revised version, we verified the *trans-*

acting activity of DiPRO1 alone or together with SIX1, on the MEF3 motif binding SIX (Fig. 8F).

11th Mar 2024

Dear Dr. Pirozhkova,

Thank you for the submission of your revised manuscript to EMBO Molecular Medicine, and please accept my apologies for the delay in getting back to you. As mentioned in a previous email, referee #1 initially asked for review time extension, but unfortunately did not return a report. Since this referee was critical at initial evaluation, referee #3 kindly agreed to assess your responses to referee #1's concerns.

As you will see from the reports below and the attached comments, referees #2 and #3 are satisfied with the revision. However, referee #3 noted that referee #1's concerns have not been entirely and satisfactorily addressed.

Therefore, we would like to invite further revisions of the study, in which we would ask you to address the following:

- R1: please clarify your answer
- R2: please provide experimental evidence or references showing that the cell lines are alveolar RMS.
- The remaining queries should be addressed experimentally or in the discussion.

As EMBO Press usually encourages one single round of revisions, please be aware that this will be the last chance for you to address these issues.

The revised manuscript will once again be subjected to review, and we cannot guarantee a positive outcome at this stage.

Moreover, please address the following editorial requests:

1/ Manuscript text:

- Please provide a .docx formatted version of the manuscript text (including legends for main figures, EV figures and tables). Please remove the red font and only keep in track changes mode any new modification.
- Please provide up to 5 keywords.
- Please remove Data not shown (p. 17). As per our guidelines on "Unpublished Data" the journal does not permit citation of "Data not shown". All data referred to in the paper should be displayed in the main or Expanded View figures.
- Data Availability: please make sure URLs are provided. Please note that the datasets must be public before publication of the manuscript.
- Acknowledgements: please note that funders listed in the submission system should be indicated in the manuscript as well. The "Dedication" section should be merged with the Acknowledgements.
- Please rename the conflicts of interest to "Disclosure statement and competing interests": We ask authors to consider both actual and perceived competing interests. Please review the policy <https://www.embopress.org/competing-interests> and update your competing interests if necessary.
- References: please list max. 10 authors before et al.
- Please note that the data citations (Data ref: DepMap portal, Broad Institute, Data ref: Kolmykov, Yevshin et al. 2021, Data ref: JASPAR collections 2020 and, Data ref: Consortium 2012) does not refer to deposited experimental data, but refers to journal article.

2/ Figures:

- Please provide exact p values, not a range, in the figures or in their legends.
- Please correct the Dataset EV4 legend and title in the tab.
- Please correct the movie nomenclature to "Movie EV1". The legend should be zipped with the movie file.
- Appendix: please add page numbers to the table of content, remove the red font and upload as a pdf.
- Please make sure that all figures/figure panels are referenced in the text, and in chronological order (callout currently missing for Fig. 8D; Fig EV2D is called out before Fig EV1 E-F; Fig EV4 C-D is called out before Fig EV2 E,I; there is a callout for a Fig EV8F but this does not exist). The callout for the video needs correcting to "Movie EV1".
- Please note that re-use of image is permitted but should be indicated in the figure legend (Figure 3C and Appendix Figure S3 A, B, C, D).
- Please address the queries from our data editors:
 1. Please note that a separate 'Data Information' section is required in the legends of figures 2a, d-e, g-j, m; 3d-g, i; 4b-d, f, h; 5d-f; 7f-g; EV 1b-d, f, i, l; EV 2d, h; EV 4a-f.
 2. Please note that the error bar information for figures 4b-c, f-h is mislabeled as 4c, f: box plots. This needs to be rectified.
 3. The error bar and asterisk related information in the legend for figure EV 2h is incorrectly labelled as EV 2d. This needs to be rectified.
 4. Please note that the legend for figure EV 3c is incorrectly labelled as EV 3b. This needs to be rectified.

5. Please define the annotated p values * in the legend of figure 3a-b as appropriate.
6. Please indicate the statistical test used for data analysis in the legends of figures 2k; 3d-f, i; 4d, h; 5c; 7f-g; 8b-c, h; EV 1f, i; EV 3b.
7. Please note that in figure 5f there is a mismatch between the annotated p values in the figure legend and the annotated p values in the figure file that should be corrected.
8. Please note that the box plots need to be defined in terms of minima, maxima, centre, bounds of box and whiskers, and percentile in the legends of figures EV 1b-c, f, EV 4b.
9. Please note that the box plots need to be defined in terms of minima, maxima, bounds of box and whiskers, and percentile in the legends of figures 3e, i; 5f; 6g; EV 4c.
10. Please note that the box plot needs to be defined in terms of minima, maxima, bounds of box and percentile in the legend of figure EV 4d.
11. Please note that information related to n is missing in the legends of figure 4h; EV 4b-c.
12. Please note that n=2 in figure 4f.
13. Although 'n' is provided, please describe the nature of entity for 'n' in the legends of figure 5f; EV 4d.
14. Please note that the error bars are not defined in the legends of figures 3a-b.
15. Please note that the red arrows are not defined in the legend of figure 2c. This needs to be rectified.

3/ Source Data: Thank you for providing Source Data. Please upload them as 1 file per figure.

4/ Synopsis:

Thank you for providing a synopsis image. Please resize it as a png/tiff/jpeg file 550px wide x 300-600px high.

Please also provide a synopsis text that should include a short stand first (maximum of 300 characters, including space) as well as 2-5 one-sentences bullet points that summarizes the paper.

5/ As part of the EMBO Publications transparent editorial process initiative (see our Editorial at <http://embomolmed.embopress.org/content/2/9/329>), EMBO Molecular Medicine will publish online a Review Process File (RPF) to accompany accepted manuscripts.

In the event of acceptance, this file will be published in conjunction with your paper and will include the anonymous referee reports, your point-by-point response and all pertinent correspondence relating to the manuscript. Let us know whether you agree with the publication of the RPF and as here, if you want to remove or not any figures from it prior to publication. Please note that the Authors checklist will be published at the end of the RPF.

I look forward to receiving your revised manuscript.

Yours sincerely,

Lise Roth

Lise Roth, PhD
Senior Scientific Editor
EMBO Molecular Medicine

***** Reviewer's comments *****

Referee #2 (Remarks for Author):

Since last review authors made a significant effort to address the reviewers comments and suggestions, improving the manuscript considerably. Of note the inclusion of scRNAseq datasets (on 6 patients PDX) in this version, showing reverse translation and increasing clinical interest/impact and context to the more basic research being explored, which was very well received.

Referee #3 (Comments on Novelty/Model System for Author):

The Authors have addressed most of the queries raised satisfactorily.

Referee #3 (Remarks for Author):

The Authors have done a commendable job in revising the manuscript and in my view, the manuscript may be accepted for

publication.

Referee 1 comments Assessment following revision

R1. Fig1-2: Chip-seq analyses have to be performed in myoblasts to strengthen the correlation with DE genes in myoblasts overexpressing or silenced for *DIPRO1*, and the subsequent identification of *DIPRO1* specific targets.

The Authors mention that the ChIP-seq experiments proved challenging due to compatibility issues. This is difficult to understand since they have done similar ChIP-seq experiments on myogenic cells in an earlier paper: Kim et al Nucleic Acids Research 2015, titled “ZNF555 protein binds to transcriptional activator site of 4qA allele and *ANTI1*: potential implication in Facioscapulohumeral dystrophy”. The Authors mention that MIRA-seq experiments, EMSA and luciferase assays have been done to address this point but it is not clear in the response where this new data is provided and what the conclusions are. Overall, the response provided is vague and does not address the query raised.

R2. Fig3: The rhabdomyosarcoma lines used are not the most consensual. The results obtained deserve to be validated in other lines such as RD (ERMS) or RH30 (ARMS) for example. which ones did we use ? there are indeed a multitude of different RMS cell lines

The Authors have used the TE671 and JR cell lines. While TE671 has been shown to be similar to the RD embryonal RMS cell line, the JR cell line is not a commonly used one. Are the Authors referring to the JR-1 cell line, which is an embryonal RMS cell line and not an alveolar RMS one (refer Hinson et al, Frontiers in Oncology 2013, “Human rhabdomyosarcoma cell lines for rhabdomyosarcoma research: utility and pitfalls”)? Clear details on this cell line, its origin and published literature needs to be confirmed before it can be considered as an alveolar RMS cell line. As suggested, a cell line such as SJRH30 is standard for alveolar RMS and would have cleared doubts in this regard. This is a major concern with respect to all interpretations made regarding alveolar RMS in this manuscript.

R3. Fig3: Authors claim that *DIPRO1* KD induces RMS cell apoptosis, based mostly on changes in the expression pattern of some effectors of the pathway. However, since these arguments are only correlative, they should instead define the impact of down regulation of *DIPRO1* on caspase-3 activity, or show how ZVAD can block this cell death. you have no functional assays done, correct?

Caspase3 activity assays using DEVD substrates or experiments to inhibit Caspase3 activity would have addressed this query effectively.

R4. Fig3: Changes in apoptotic genes correspond to gain of both anti and pro-apoptotic genes. Progeny or Ingenuity scores should be applied to infer score of activation of the pathway.

This query has been satisfactorily addressed.

R5. Fig 2-3: What is the overlap in DE genes in myoblasts and cancer cells models? This could improve the identification of *DIPRO1* specific targets, since the number of DE genes is huge and could lead to confounding effects.

This query has been satisfactorily addressed.

R6. Fig 4: What is the relevance of this therapeutic approach in an RMS model ?

The shRNA/siRNA based approach is adequate but the query is with respect to RMS, since the Authors have only carried out in vivo experiments for an Ewing's sarcoma model.

R7. Fig 5: The rationale of exploring MappyActs pan-cancers relapse cohort rather than publicly available transcriptomic data on RMS and EWS is not clear. What is the expression level of *DIPRO1* in RMS compared to normal muscles ? What is the prognostic value of this gene in EWS/RMS cohorts that have already been published ? Is there a correlation between *DIPRO1* and some of the target genes presented here ? Along the same line, what is the expression profile of *DIPRO1* in the recently published single-cell analyses of RMS samples/models ? Is its expression restricted to mesenchymal-like progenitors cluster ? Assume you could do the expression also over other public cohorts ?

The Authors have performed additional analyses and added new data to address this comment.

R8. Fig 5: The robustness and relevance of exploring the so-called downstream genes network is not that clear/convincing. Intersection between *CHIPseq* and *RNAseq* analyses performed in all overexpressing/downregulated models should be used to define this network and to strengthen the conclusions.

The Authors again mention that *MIRA-seq* experiments, *EMSA* and luciferase assays have been done to address this point but it is not clear in the response where this new data is provided and what the conclusions are.

R9. The link between *SOX2/SIX1* and *DIPRO1* across models is at least confusing, and perhaps even over interpreted. Authors should provide more evidences that there is a synergistic action of *DIPRO1* and *SOX2* or *SIX1* on target genes, resulting respectively from epigenetic and transcriptional regulation, by showing for example that impact of *DIPRO1* overexpression on the expression of these genes is blocked by downregulation of *SIX1* or *SOX2*.

This query has been satisfactorily addressed.

R10. The fact that only 14.6% of *DMCs* correlate with changes in genes expression in RMS cells seems quite low. The authors should comment on this and perhaps provide some statistical insights. Same comment for the 12.4% of promoter regions bound by *SIX1* and *DMCs* in *DIPRO1 KD*.

This query has been satisfactorily addressed.

Minor points

R1. Authors report their previous finding upon the role of *DIPRO1* on *ANT1* regulation. Regarding the pro-apoptotic role of *ANT1*, and its previously established role in RMS, how do the authors reconcile their finding that *DIPRO1* down regulation induces apoptosis ?

This query has been satisfactorily addressed.

R2. Some English/typo misspellings.

***** Reviewer's comments *****

Referee #2 (Remarks for Author):

Since last review authors made a significant effort to address the reviewers comments and suggestions, improving the manuscript considerably. Of note the inclusion of scRNAseq datasets (on 6 patients PDX) in this version, showing reverse translation and increasing clinical interest/impact and context to the more basic research being explored, which was very well received.

Referee #3 (Comments on Novelty/Model System for Author):

The Authors have addressed most of the queries raised satisfactorily.

Referee #3 (Remarks for Author):

The Authors have done a commendable job in revising the manuscript and in my view, the manuscript may be accepted for publication.

We are grateful to the referees for their positive feedback and for the time they devoted to our manuscript. We appreciate the helpful comments and suggestions. Following the main recommendations of the referees 1 and 3, further improvements have been made to the manuscript. The detailed answers (A) to the referees' comments (R) are presented below.

Referee 1 comments Assessment following revision

R1. Fig1-2: Chip-seq analyses have to be performed in myoblasts to strengthen the correlation with DE genes in myoblasts overexpressing or silenced for DiPRO1, and the subsequent identification of DiPRO1 specific targets.

The Authors mention that the ChIP-seq experiments proved challenging due to compatibility issues. This is difficult to understand since they have done similar ChIP-seq experiments on myogenic cells in an earlier paper: Kim et al Nucleic Acids Research 2015, titled "ZNF555 protein binds to transcriptional activator site of 4qA allele and ANT1: potential implication in Facioscapulohumeral dystrophy". The Authors mention that MIRA-seq experiments, EMSA and luciferase assays have been done to address this point but it is not clear in the response where this new data is provided and what the conclusions are.

Overall, the response provided is vague and does not address the query raised.

A1. We agree with the referees' recommendation; however, the anti-ZNF555 antibody batch (AB4, Sigma-Aldrich) was replaced, and subsequent control experiments to assess antibody specificity by western blot in functional DiPRO1 KD and overexpression experiments, as well as by EMSA experiments were unsuccessful. Despite attempts with other anti-ZNF555 antibodies from Abcam and Santa-Cruz, satisfactory control results were not obtained. We reported this information to our suppliers. The data of DiPRO1 ChIPseq in HEK cells were previously published by Didier TRONO's team. Replication of ChIP-seq experiments in different cell lines can help confirm the robustness and generalizability of results, and should be consistent across different cell lines, with the exception of cell-specific regulations. To assess the myogenic specificity of DiPRO1 direct targets, in our experiments we demonstrated the affinity of ZNF555 to bind the MEF3 consensus (TCAGGTTTC) (Santolini, Sakakibara et al. 2016), which is presented within the target genes of myogenic regulatory factors (Liu, Chu et al. 2010), highlighting its crucial role in myogenesis. This is consistent with the transcriptomic analysis performed in DiPRO1 overexpression/reduction models. This sheds further light on muscle cell-specific binding targets, which can be directly regulated by DiPRO1. In our study, we mainly focused on dynamic regulatory networks in

response to DiPRO1 KD, integrating gene expression and DNA methylation in relation to DiPRO1 protein partners, thus revealing novel properties of this protein. We retrieved published ChIP-seq data to enhance our understanding of the established DiPRO1 network.

R2. Fig3: The rhabdomyosarcoma lines used are not the most consensual. The results obtained deserve to be validated in other lines such as RD (ERMS) or RH30 (ARMS) for example. which ones did we use ? there are indeed a multitude of different RMS cell lines

The Authors have used the TE671 and JR cell lines. While TE671 has been shown to be similar to the RD embryonal RMS cell line, the JR cell line is not a commonly used one. Are the Authors referring to the JR-1 cell line, which is an embryonal RMS cell line and not an alveolar RMS one (refer Hinson et al, Frontiers in Oncology 2013, "Human rhabdomyosarcoma cell lines for rhabdomyosarcoma research: utility and pitfalls")? Clear details on this cell line, its origin and published literature needs to be confirmed before it can be considered as an alveolar RMS cell line. As suggested, a cell line such as SJRH30 is standard for alveolar RMS and would have cleared doubts in this regard. This is a major concern with respect to all interpretations made regarding alveolar RMS in this manuscript.

A2. We used the JR cell line, identified as an alveolar RMS, as documented in the Cellosaurus database (CVCL_RT33). All cell lines underwent genotyping verification. Profiling of the human cell lines was performed using highly polymorphic short tandem repeat loci (STRs). STR loci were amplified using the PowerPlex® 16 HS System (Promega). Fragment analysis was performed on an ABI3730xl instrument (Life Technologies), and the resulting data were analyzed using GeneMarker HID software (Softgenetics) or other software provided by the Childhood Cancer Repository Cell Line and Xenograft STR Database (Texas). This cell line is well-known and extensively characterized, with several publications referencing its use, including the most recent one focusing on immunotherapy development (DOI: 10.3390/ijms24032601)

R3. Fig3: Authors claim that DIPRO1 KD induces RMS cell apoptosis, based mostly on changes in the expression pattern of some effectors of the pathway. However, since these arguments are only correlative, they should instead define the impact of down regulation of DIPRO1 on caspase-3 activity, or show how ZVAD can block this cell death. you have no functional assays done, correct?

Caspase3 activity assays using DEVD substrates or experiments to inhibit Caspase3 activity would have addressed this query effectively

A3. The additional experiments *in vitro* have been performed. Caspase-3 activation was shown in RMS cells under DiPRO1 KD (new Fig. EV1K), consistent with the results obtained *in vivo*.

R6. Fig 4: What is the relevance of this therapeutic approach in an RMS model ?

The shRNA/siRNA based approach is adequate but the query is with respect to RMS, since the Authors have only carried out in vivo experiments for an Ewing's sarcoma model.

A6. There are many different types of mesenchymal cancer, including osteosarcoma, chondrosarcoma, liposarcoma and leiomyosarcoma. The main aim of our *in vivo* study was to demonstrate the relevance of DiPRO1 targeting for mesenchymal cancers. The decision to focus on Ewing sarcoma models for our *in vivo* experiments was based on several factors, including scientific rationale and resource constraints. Although rhabdomyosarcoma (RMS) is indeed relevant in the context of mesenchymal tumors, our choice to prioritize Ewing sarcoma was motivated by its well-established representation as a mesenchymal tumor type. Ewing sarcoma is a widely studied mesenchymal tumor, and its molecular pathways often overlap with those of RMS. The second point was to extend the effect of DiPRO1 KD from RMS (extensively studied in our work) to other mesenchymal cancers. Furthermore, given the scope and objectives of our study, focusing on a single tumor model allowed us to allocate our resources efficiently and ensure the

robustness of our experimental approach. In fact, the experiments were carried out three times, optimizing the dose and schedule of administration.

While we acknowledge the importance of investigating DiPRO1 targeting in RMS models, we believe that our findings in Ewing sarcoma provide a solid foundation for further exploration of DiPRO1 as a therapeutic target for mesenchymal tumors, including RMS. Future preclinical studies could certainly extend our findings to RMS and different mesenchymal tumor types.

With this in mind, the comment has been added to the Discussion chapter.

R8. Fig 5: The robustness and relevance of exploring the so-called downstream genes network is not that clear/convincing. Intersection between CHIPseq and RNAseq analyses performed in all overexpressing/downregulated models should be used to define this network and to strengthen the conclusions.

The Authors again mention that MIRA-seq experiments, EMSA and luciferase assays have been done to address this point but it is not clear in the response where this new data is provided and what the conclusions are

A8. An supplementary analysis was carried out, corresponding to Figure EV3E and Dataset EV4D.

11th Jun 2024

Dear Dr. Pirozhkova,

Thank you for the submission of your revised manuscript to EMBO Molecular Medicine, which was sent back to referee #3. As you will see below, this referee is satisfied with the revisions, and I will therefore be able to accept your manuscript pending minor editorial revisions:

- Please remove the red font and only keep in track changes mode any new modification.
- Materials and Methods
 - o Antibodies: please indicate the dilutions/concentrations used
 - o Cell culture: please indicate whether the cells were authenticated and tested for mycoplasma contamination
 - o Patients: please include the full sentence that the experiments conformed to the principles set out in the WMA Declaration of Helsinki and the Department of Health and Human Services Belmont Report
 - o Animals: please provide the housing and husbandry conditions.
 - o Statistics: please provide a statement on blinding, randomization, sample size and inclusion/exclusion criteria.
- Data Availability: Please note that this section is restricted to datasets generated in this study. Please correct accordingly.
- Fig. EV8F is referenced in the text but does not exist. Please correct.
- Checklist: please fill in all the subsections in "Experimental study design and statistics". Please also complete the "Ethics" section (human samples)
- Thank you for providing a For More Information section. Please provide the links as a bullet points list (i.e.:
 - o UCSC Genome Browser ENCODE: <https://hgdownload.soe.ucsc.edu>
 - o CpG sites: <http://methyIqa.sourceforge.net/>
 - o ...)
- Source Data:
 - o Please note that I could not open the SD for Figure 2B, 3H, 6B and 6F.
 - o Please provide SD for Figure 5F.
 - o Please provide explanation for the Source Data that have not been provided.

- I slightly edited your synopsis text, please let me know if you agree with the following or amend as you see fit:

Inhibiting DiPRO1 promotes myogenesis in myoblasts yet leads to apoptosis in rhabdomyosarcoma and Ewing sarcoma cells. Targeting DiPRO1 with si/shDiPRO1 nanomedicines represent a potential treatment avenue for mesenchymal cancers.

- DiPRO1, in conjunction with TIF1B and UHRF1, participates in the epigenetic silencing of cellular responses to viruses by maintaining the repression of retrotransposable repeats (RE).
- DiPRO1 plays a role in cell fate determination as a modulator of TNF- α via NF-kappaB signaling.
- DiPRO1 myogenic regulatory functions extend to SIX1 binding sites.

- Please let us know if you agree with the publication of the Review Process File in conjunction with your paper (it will include the anonymous referee reports, your point-by-point response and all pertinent correspondence relating to the manuscript). Let us know whether you agree with the publication of the RPF and as here, if you want to remove or not any figures from it prior to publication.

I look forward to receiving your revised manuscript.

With kind regards,

Lise Roth

**** Reviewer's comments ****

Referee #3 (Comments on Novelty/Model System for Author):

The cellular models used are adequate.

Referee #3 (Remarks for Author):

The Authors have addressed all the queries and the manuscript may now be accepted for publication.

The authors addressed the minor formatting issues.

18th Jun 2024

Dear Dr. PIROZHKOVA,

Thank you for submitting your revised files. I am pleased to inform you that your manuscript is accepted for publication and is now being sent to our publisher to be included in the next available issue of EMBO Molecular Medicine.

Please note that I have inserted the mention that no blinding was done in the main manuscript file and in the author checklist. Please get back to us immediately if you do not agree with these changes.

Please also note that I could not access the following link in the 'For more information' section:

EaSeq: <https://easeq.net>

Kindly double-check that the link is functional, and correct if needed at proof stage.

With kind regards,

Lise
